# Sharp Statistical Limits and Algorithm for Attributed Graph Alignment

## Abstract

This paper investigates the problem of recovering hidden vertex mappings between two correlated weighted graphs with both edge structure and node features. While most existing studies on graph alignment focus solely on edge information, many practical scenarios also provide node features in addition to graph topology. To address this setting, we introduce the featured correlated Gaussian Wigner model, in which the graphs are correlated through a latent vertex permutation, and the associated features are also correlated under the same permutation. We establish the optimal information-theoretic thresholds for recovering the latent vertex mappings. Furthermore, we propose QPAlign, a fast algorithm leveraging quadratic programming relaxation to the Birkhoff polytope, and validate its effectiveness on both synthetic and real datasets.

## 1 Introduction

Graph alignment is a fundamental problem in network science and machine learning, with applications in many areas. For example, in computer vision, 3-D shapes can be represented as graphs and a significant problem for pattern recognition and image processing is determining whether two graphs represent the same object under rotations (Berg et al., 2005; Mateus et al., 2008); in natural language processing, each sentence can be represented as a graph and the ontology problem refers to uncovering the correlation between different knowledge graphs that are in different languages (Haghighi et al., 2005); in computational biology, proteins can be regarded as vertices and the interactions between them can be formulated as weighted edges (Singh et al., 2008; Vogelstein et al., 2015).

Since real-world scenarios often present challenges due to the noise in real data, many studies focused on random graph models to serve as a pivotal step, including graph alignment problem in Erdős-Rényi model (Wu et al., 2022; Ding & Du, 2023b; Huang et al., 2024), Gaussian Wigner model (Fan et al., 2023a; Araya et al., 2024; Ding & Li, 2024), stochastic block model (Onaran et al., 2016; Lyzinski, 2018; Chai & Rácz, 2024), and graphon model (Zhang, 2018). However, previous works on random graph alignment mainly focus on models that rely solely on topological information. In real scenarios, however, feature information often plays a crucial role. For instance, in the ACM–DBLP dataset, node features such as paper titles or author names are essential for identifying corresponding entities across the two graphs (Tang et al., 2023; Bommakanti et al., 2024). This motivates the study of alignment models that incorporate both structural and feature information, beyond purely topology-based settings.

While existing work on attributed graph alignment mostly builds on correlated Erdős-Rényi or community-based model with binary edges and node attributes (Zhang et al., 2024; Yang & Chung, 2024; 2025), many real-world networks are both weighted and attributed. For example, in gene co-expression networks (Zhang & Horvath, 2005), edge weights encode co-expression strength while genes carry functional annotations, and in social networks (Leskovec et al., 2010), edges record rating or interaction strengths while nodes have profile or content features. To bridge this gap, we investigate the following Featured correlated Gaussian Wigner model.

**Definition 1** (Featured correlated Gaussian Wigner model). *Let $G_1$ and $G_2$ be two weighted random graphs with vertex sets $V(G_1), V(G_2)$ such that $|V(G_1)| = |V(G_2)| = n$. Let $\pi^*$ denote a latent bijective mapping from $V(G_1)$ to $V(G_2)$. We say that a pair of graphs $(G_1, G_2)$ follows featured correlated Gaussian Wigner model $\mathcal{G}(n, d, \rho, r)$ if*

1. *each pair of weighted edges $\beta_{uv}(G_1)$ and $\beta_{\pi^*(u)\pi^*(v)}(G_2)$ for any $u, v \in V(G_1)$ are correlated standard normals with correlation coefficient $\rho \in (0, 1)$;*

2. *each pair of features $(\boldsymbol{x}_u, \boldsymbol{y}_{\pi^*(u)})$ for any $u \in V(G_1)$ follows multivariate normal distribution $\mathcal{N}(0, \Sigma_d)$ with $\Sigma_d = \begin{bmatrix} I_d & rI_d \\ rI_d & I_d \end{bmatrix}$, where the dimension $d \in \mathbb{N}$ and the correlation $r \in (0, 1)$. Moreover, we assume that the features are independent with the weighted edges.*

We assume features are standardized so that each coordinate has unit variance and is independent, which justifies the identity matrices in $\Sigma_d$. Edge weights are likewise centered and variance-normalized, with correlations $\rho$ and $r$ capturing structural and feature dependence, respectively. Furthermore, we assume $\rho, r \in (0, 1)$ without loss of generality, since the model is invariant under flipping the signs of all edge weights or node features in $G_2$, and hence negative correlations can be reduced to positive ones. Indeed, this model bridges two significant extremes, it reduces to correlated Gaussian Wigner model (Ding et al., 2021) when $r = 0$ and correlated Gaussian database model (Dai et al., 2019a) when $\rho = 0$. Given $G_1$ and $G_2$ under $\mathcal{G}(n, d, \rho, r)$, our goal is to recover the latent vertex mapping $\pi^*$. More specifically, given two permutations $\pi^*, \hat{\pi} : V(G_1) \mapsto V(G_2)$, denote the fraction of their overlap by $\text{overlap}(\pi^*, \hat{\pi}) = \frac{1}{n} |\{v \in V(G_1) : \pi^*(v) = \hat{\pi}(v)\}|$. To quantify the performance of an estimator $\hat{\pi}$, we say an estimator $\hat{\pi}(G_1, G_2)$ achieves

- *partial recovery*, if $\text{overlap}(\hat{\pi}, \pi^*) \geq \delta$ for some $\delta \in (0, 1)$;
- *exact recovery*, if $\text{overlap}(\hat{\pi}, \pi^*) = 1$.

## 1.1 MAIN RESULTS

In this subsection, we present our main results on information-theoretic thresholds. Let $\mathcal{S}_n$ denote the set of bijective mappings $\pi : V(G_1) \mapsto V(G_2)$. Our goal is to determine the correlation required for successful recovery of $\pi^*$. Next, we introduce our main theorems.

**Theorem 1** (Partial Recovery). *If $d = \omega(\log n)$ and $n \log(\frac{1}{1-\rho^2}) + 2d \log(\frac{1}{1-r^2}) \geq (4 + \epsilon) \log n$ for some constant $\epsilon > 0$, then there exists an estimator $\hat{\pi}$ such that, for any fixed constant $0 < \delta < 1$ and $\pi^* \in \mathcal{S}_n$, we have*

$$\mathbb{P}\left[\text{overlap}(\hat{\pi}, \pi^*) \geq \delta\right] = 1 - o(1).$$

*Conversely, for any $0 < \delta < 1$, if $n \log(\frac{1}{1-\rho^2}) + 2d \log(\frac{1}{1-r^2}) \leq c \log n$ for some constant $c$, then for any estimator $\hat{\pi}$,*

$$\mathbb{P}\left[\text{overlap}(\hat{\pi}, \pi^*) < \delta\right] \geq 1 - \frac{c}{4\delta},$$

*where $\pi^*$ is uniformly distributed over $\mathcal{S}_n$.*

The upper bound holds uniformly over all $\pi^* \in \mathcal{S}_n$, while the lower bound is obtained by analyzing the Bayes risk under the uniform prior on $\pi^*$, which is least favorable in our permutation-symmetric model and therefore leads to the same threshold for the minimax risk. Consequently, the threshold is valid for both minimax and Bayesian risks. As for the assumption $d = \omega(\log n)$, it is standard in Gaussian database alignment: identifying a vertex among $n$ candidates requires $\Theta(\log n)$ bits, while each feature coordinate contributes only $O(1)$ bits of discriminative information, so the total feature dimension must eventually dominate $\log n$ (see, e.g., Dai et al. (2019a)). Theorem 1 characterizes the optimal rate for the information-theoretic threshold in the partial recovery regime. In particular, the special case $\delta = 1$ corresponds to exact recovery. For obtaining a sharper constant in this regime, we have the following theorem.

**Theorem 2** (Exact Recovery). *If $d = \omega(\log n)$ and $n \log(\frac{1}{1-\rho^2}) + d \log(\frac{1}{1-r^2}) \geq (4 + \epsilon) \log n$ for some constant $\epsilon > 0$, then there exists an estimator $\hat{\pi}$ such that, for any fixed constant $0 < \delta < 1$ and $\pi^* \in \mathcal{S}_n$, we have*

$$\mathbb{P}\left[\text{overlap}(\hat{\pi}, \pi^*) = 1\right] = 1 - o(1).$$

*Conversely, if $r^2 \geq \frac{40}{d}$ and $n \log(\frac{1}{1-\rho^2}) + d \log(\frac{1}{1-r^2}) + 4 \log d \leq (4 - \epsilon) \log n$ for some constant $\epsilon > 0$, then for any estimator $\hat{\pi}$,*

$$\mathbb{P}\left[\text{overlap}(\hat{\pi}, \pi^*) \neq 1\right] = 1 - o(1),$$

*where $\pi^*$ is uniformly distributed over $\mathcal{S}_n$.*

The technical condition $r^2 \geq 40/d$ is only imposed to sharpen the leading constant in this regime: Theorem 1 already yields the optimal rate without this assumption under the special case $\delta = 1$, while under $d = n^{o(1)}$ and $d = \omega(\log n)$ such a lower bound on the feature signal ensures that our exact recovery threshold attains the optimal constant. In comparison with the special case $\delta = 1$ in Theorem 1, Theorem 2 establishes a sharper information-theoretic threshold for exact recovery under certain conditions on $d$. Indeed, the difference between the $2d$-term in partial recovery and the $d$-term in exact recovery arises because exact recovery requires distinguishing much smaller distances between candidate alignments, which in turn necessitates stronger correlation and thus a stronger condition.

For the special case $r = 0$, (Wu et al., 2022) shows that in the correlated Gaussian Wigner model, there is a phase transition from possible exact recovery to impossible recovery as the quantity $\frac{n \log(1/(1-\rho^2))}{\log n}$ changes from $4 + \epsilon$ to $4 - \epsilon$. For another special case $\rho = 0$, (Dai et al., 2019a) demonstrates that in the Gaussian database model exact recovery is possible when $d \log(1/(1-\rho^2)) \geq (4+\epsilon) \log n$, and impossible when $d \log(1/(1-\rho^2)) \leq (4-\epsilon) \log n$, under the same condition on the feature dimension $d$. In our new model $\mathcal{G}(n, d, \rho, r)$, we show that the optimal threshold is determined by the two components of the model: structure and features. Specifically, the term $n \log(1/(1-\rho^2))$ captures the contribution of structural information, while $d \log(1/(1-r^2))$ captures the contribution of feature information. Indeed, when $n \log(1/(1-\rho^2)) = C_1 \log n$ and $d \log(1/(1-r^2)) = C_2 \log n$ with $C_1, C_2 < 4$, and $C_1 + C_2 > 4$, there exists an estimator $\hat{\pi}$ that achieves exact recovery only when both edge and feature information are available. This demonstrates that our approach goes beyond the performance attainable by relying on either structural or feature information alone.

## 1.2 RELATED WORK

**Attributed graph alignment** In the information-theoretic perspective, Zhang et al. (2024) proposed the attributed Erdős-Rényi pair model, where the edges and the features follows Bernoulli distribution under a latent bijective mapping, and it derived the information-theoretic thresholds for recovering the latent mappings in both possibility and impossibility regimes. Yang & Chung (2024) proposed the correlated Gaussian-attributed Erdős-Rényi model and derived the optimal information-theoretic thresholds for exact recovery by analyzing the $k$-core estimator. Both work proposed random graph model with additional feature and found that the graph matching becomes feasible in a wider regime through the information of attributed nodes. There are also many algorithms on attributed in attributed graph alignment, including methods based on subgraph counting (Du et al., 2017; Liu et al., 2019; Wang et al., 2025), spectral methods (Zhang & Tong, 2016), optimal transport (Tang et al., 2023), and neighborhood statistics (Wang et al., 2024).

**Other graph models** Many information-theoretic properties of the correlated Gaussian Wigner model and correlated Erdős-Rényi model have been extensively investigated (Cullina & Kiyavash, 2016; 2017; Ganassali et al., 2021; Wu et al., 2022; 2023; Ding & Du, 2023b;a; Hall & Massoulié, 2023; Huang et al., 2024; Du, 2025; Huang & Yang, 2025), along with a rich line of algorithmic developments (Babai et al., 1980; Bollobás, 1982; Barak et al., 2019; Dai et al., 2019b; Ganassali & Massoulié, 2020; Ding et al., 2021; Mao et al., 2021; Piccioli et al., 2022; Mao et al., 2023a;c; Fan et al., 2023a;b; Ding & Li, 2023; 2024; Araya et al., 2024; Ganassali et al., 2024; Muratori & Semerjian, 2024; Du et al., 2025). However, the marginal distributions inherent in these models makes it different from graph models in practical applications. Therefore, it is crucial to explore more general graph models, such as graphon model (Wolfe & Olhede, 2013; Gao et al., 2015), inhomogeneous graph model (Rácz & Sridhar, 2023; Song et al., 2023; Ding et al., 2025), geometric random graph model (Wang et al., 2022; Bangachev & Bresler, 2024; Gong & Li, 2024; Sentenac et al., 2025), planted cycle model (Mao et al., 2023b; 2024), and multiple graph model (Ameen & Hajek, 2024; 2025).

## 2 POSSIBILITY RESULTS

We first introduce our estimator. Given two graphs $(G_1, G_2) \sim \mathcal{G}(n, d, \rho, r)$ under the featured correlated Gaussian Wigner model, our goal is design an estimator $\hat{\pi}(G_1, G_2)$ to recover the latent bijective mapping $\pi^* : V(G_1) \mapsto V(G_2)$. Let $\mathcal{P}$ denote the joint distribution of $(G_1, G_2)$, $P(\cdot, \cdot)$

denote the distribution of two correlated standard normals with correlation $\rho$, and $f(\cdot, \cdot)$ denote the multivariate normal distribution $\mathcal{N}(0, \Sigma_d)$ with $\Sigma_d = \begin{bmatrix} I_d & r I_d \\ r I_d & I_d \end{bmatrix}$. Then the likelihood function is

$$\mathcal{P}_{G_1, G_2 | \pi^*} = \prod_{e \in E(G_1)} P\left(\beta_e(G_1,), \beta_{\pi^*(e)}(G_2)\right) \prod_{v \in V(G_1)} f(\boldsymbol{x}_v, \boldsymbol{y}_{\pi^*(v)}) \tag{1}$$

$$\stackrel{(a)}{\propto} \exp\left(\frac{\rho}{1 - \rho^2} \sum_{e \in E(G_1)} \beta_e(G_1) \beta_{\pi^*(e)}(G_2) + \frac{r}{1 - r^2} \sum_{v \in V(G_1)} \boldsymbol{x}_v \boldsymbol{y}_{\pi^*(v)}\right),$$

where (a) is because the quantities $\sum_{e \in E(G_1)} (\beta_e(G_1))^2$, $\sum_{e \in E(G_2)} (\beta_e(G_2))^2$, $\sum_{v \in V(G_1)} \boldsymbol{x}_v^\top \boldsymbol{x}_v$, and $\sum_{v \in V(G_1)} \boldsymbol{y}_{\pi^*(v)}^\top \boldsymbol{y}_{\pi^*(v)}$ are fixed given $(G_1, G_2)$. Consequently, our estimator takes the form

$$\hat{\pi} \in \arg\max_\pi \left\{ \frac{\rho}{1 - \rho^2} \sum_{e \in E(G_1)} \beta_e(G_1) \beta_{\pi(e)}(G_2) + \frac{r}{1 - r^2} \sum_{v \in V(G_1)} \boldsymbol{x}_v \boldsymbol{y}_{\pi(v)} \right\}. \tag{2}$$

Let $\varphi(x) = \frac{x}{1 - x^2}$ and $S_\pi(G_1, G_2) = \varphi(\rho) \sum_{e \in E(G_1)} \beta_e(G_1) \beta_{\pi(e)}(G_2) + \varphi(r) \sum_{v \in V(G_1)} \boldsymbol{x}_v \boldsymbol{y}_{\pi(v)}$. Indeed, $S_\pi(G_1, G_2)$ represents the similarity score under $\pi$ between $G_1$ and $G_2$. Then the estimator is equivalent to $\hat{\pi} \in \arg\max_\pi S_\pi(G_1, G_2)$. For any two bijections $\pi, \pi' : V(G_1) \mapsto V(G_2)$, let $\mathsf{d}(\pi, \pi') = n(1 - \mathrm{overlap}(\pi, \pi'))$. To prove the successful recovery of $\hat{\pi}$, it suffices to show that

$$S_{\pi^*}(G_1, G_2) > \max_{\pi : \mathsf{d}(\pi, \pi^*) \geq d_0} S_\pi(G_1, G_2) = \max_{k \geq d_0} \max_{\pi : \mathsf{d}(\pi, \pi^*) = k} S_\pi(G_1, G_2)$$

with high probability, where the thresholds $d_0 = 1$ and $d_0 = (1 - \delta)m$ correspond to the exact and partial recoveries, respectively. Let $\mathcal{T}_k$ denote the set of bijective mappings such that $\mathsf{d}(\pi, \pi^*) = k$. Then the failure event satisfies

$$\{\mathsf{d}(\hat{\pi}, \pi^*) = k\} \subseteq \{\exists \pi' \in \mathcal{T}_k : S_{\pi^*}(G_1, G_2) \leq S_{\pi'}(G_1, G_2)\}.$$

Accordingly we bound $\mathbb{P}[\mathsf{d}(\hat{\pi}, \pi^*) = k]$ separately for large and small values of $k$. The next two propositions provide those bounds.

**Proposition 1.** *If $d = \omega(\log n)$ and $n \log\left(\frac{1}{1 - \rho^2}\right) + 2d \log\left(\frac{1}{1 - r^2}\right) \geq (4 + \epsilon) \log n$ with some constant $0 < \epsilon < 1$, then for any constant $0 < \delta < 1$ and $k \geq \delta n$, there exists $\hat{\pi}$ such that*

$$\mathbb{P}[\mathsf{d}(\hat{\pi}, \pi^*) = k] \leq \exp\left(-n h\left(\frac{k}{n}\right)\right) \mathbf{1}_{\{k \leq n - 1\}} + \exp(-2 \log n) \mathbf{1}_{\{k = n\}} + \exp\left(-\frac{\epsilon k \log n}{32}\right),$$

*where $h(x) = -x \log x - (1 - x) \log(1 - x)$ is the binary entropy function.*

In view of Proposition 1, an upper bound is established for any $\mathsf{d}(\hat{\pi}, \pi^*) = k$ with $k \geq \delta n$. Summing over all $k \geq \delta n$ yields an error estimate for $\mathbb{P}[\mathrm{overlap}(\hat{\pi}, \pi^*) \geq \delta]$ in the partial recovery regime. However, the proposition only controls the error probability when $k \geq \delta n$. To derive an error bound in the exact recovery regime, it remains necessary to handle the case $k < \delta n$. Specifically, we establish the following proposition.

**Proposition 2.** *If $n \log\left(\frac{1}{1 - \rho^2}\right) + d \log\left(\frac{1}{1 - r^2}\right) \geq (4 + \epsilon) \log n$ with some constant $0 < \epsilon < 1$, then for any $k \leq \frac{\epsilon}{16} n$, the estimator $\hat{\pi}$ in equation 2 satisfies*

$$\mathbb{P}[\mathsf{d}(\hat{\pi}, \pi^*) = k] \leq \exp\left(-\frac{\epsilon}{8} k \log n\right).$$

The proofs of Propositions 1 and 2 are deferred to Appendices C.1 and C.2, respectively. By combining these two propositions, we obtain the possibility results stated in Theorems 1 and 2, through summing over $k \geq \delta n$ and $k \geq 1$, respectively. The main task behind Propositions 1 and 2 is to control the MLE score difference $Z_\pi = S_\pi(G_1, G_2) - S_{\pi^*}(G_1, G_2)$ uniformly over all permutations under a mixed Gaussian channel (continuous edges and high-dimensional features). For permutations with macroscopic Hamming distance, we adapt the cycle decomposition and Gaussian moment-generating-function computation from Wu et al. (2022) to this structural–feature setting, obtaining sharp bounds with exponent $n \log(1/(1 - \rho^2)) + 2d \log(1/(1 - r^2))$. For permutations very close to $\pi^*$, we represent $S_{\pi^*} - S_\pi$ as a quadratic form in a jointly Gaussian vector (edges and features together), decorrelate the coordinates, and apply the Hanson-Wright inequality (Hanson & Wright, 1971) to get uniform bounds and the sharp constant in the exact recovery threshold.

# 3 IMPOSSIBILITY RESULTS

In this section, we present information-theoretic impossibility results for the graph alignment problem. For the converse arguments, we adopt a Bayesian formulation by endowing the ground-truth permutation $\pi^*$ with the uniform prior on $\mathcal{S}_n$; under this prior, the MLE $\hat{\pi}$ minimizes the error probability among all estimators. To prove the impossibility results, it suffices to prove the failure of MLE, which corresponds to show the existence of a permutation $\pi'$ that achieves a higher likelihood than the true permutation $\pi^*$. However, such strategy only proves impossibility results for exact recovery regime (See Proposition 4). We will show the impossibility results for the partial recovery regime by Fano's method (see, e.g. (Cover & Thomas, 2006, Section 2.10)).

Let $M_\delta$ be a packing set of $\mathcal{S}_n$ such that two distinct elements $\pi, \pi' \in \mathcal{M}_\delta$ differs from a certain threshold. Specifically, we choose $\min_{\pi \neq \pi' \in \mathcal{M}} \mathrm{d}(\pi, \pi') > (1 - \delta)n$ in partial recovery regime and $\mathcal{M}_1 = \mathcal{S}_n$. The cardinality of $M_\delta$ measures the complexity of the parameter space under the corresponding metric. Let $\mathcal{P}$ denote the joint distribution of $(G_1, G_2)$, $\mathcal{Q}$ be any distribution over $(G_1, G_2)$, and $D_{\mathrm{KL}}$ denote the Kullback–Leibler (KL) divergence. Then, the mutual information $I(\pi^*; G_1, G_2)$ can be upper bounded by

$$I(G_1, G_2; \pi^*) = \mathbb{E}_{\pi^*}\left[D_{\mathrm{KL}}(\mathcal{P}_{G_1, G_2 | \pi^*} \| \mathcal{P}_{G_1, G_2})\right] \leq \max_{\pi \in \mathcal{S}_n} D_{\mathrm{KL}}(\mathcal{P}_{G_1, G_2 | \pi} \| \mathcal{Q}_{G_1, G_2}),$$

where the inequality is because

$$D_{\mathrm{KL}}(\mathcal{P}_{G_1, G_2 | \pi^*} \| \mathcal{P}_{G_1, G_2}) = D_{\mathrm{KL}}(\mathcal{P}_{G_1, G_2 | \pi^*} \| \mathcal{Q}_{G_1, G_2}) - D_{\mathrm{KL}}(\mathcal{P}_{G_1, G_2} \| Q_{G_1, G_2})$$

and the KL divergence $D_{\mathrm{KL}}(\mathcal{P}_{G_1, G_2} \| Q_{G_1, G_2}) \geq 0$ for any distribution $\mathcal{Q}$. By Fano's inequality, with $\pi^*$ being the discrete uniform prior in the packing set $\mathcal{M}_\delta$, for any estimator $\hat{\pi}$, we have

$$\mathbb{P}\left[\mathrm{overlap}(\hat{\pi}, \pi^*) < \delta\right] \geq 1 - \frac{I(\pi^*; G_1, G_2) + \log 2}{\log |\mathcal{M}_\delta|}. \tag{3}$$

Specifically, we have the following Lemma on $|\mathcal{M}_\delta|$ and $I(\pi^*; G_1, G_2)$.

**Lemma 1.** *For any $0 < \delta \leq 1$, we have*

$$|\mathcal{M}_\delta| \geq \left(\frac{\delta n}{e}\right)^{\delta n}, \quad I(\pi^*; G_1, G_2) \leq \frac{1}{2}\binom{n}{2}\log\left(\frac{1}{1 - \rho^2}\right) + \frac{nd}{2}\log\left(\frac{1}{1 - r^2}\right). \tag{4}$$

In view of Lemma 1, $n \log(\frac{1}{1-\rho^2})$ and $d \log(\frac{1}{1-r^2})$ correspond to the mutual information contributed by the graph structure and the by the node features, respectively. Since the edges and features are independent, the total mutual information is given by the sum $n \log(\frac{1}{1-\rho^2}) + d \log(\frac{1}{1-r^2})$. The proof of Lemma 1 is deferred to Appendix D.1. Fano's method lower bounds the Bayesian risk under the discrete uniform prior $\pi^*$ on $\mathcal{M}_\delta$, which in turn also lower bounds the minimax risk. In particular, by combining Fano's inequality in equation 3 with the relation in equation 4 in Lemma 1, we obtain the impossibility result for partial recovery with any constant $\delta \in (0, 1]$, as stated below.

**Proposition 3** (Impossibility result, partial recovery). *For any $0 < \delta \leq 1$, if $n \log\left(\frac{1}{1-\rho^2}\right) + 2d \log\left(\frac{1}{1-r^2}\right) \leq c \log n$ for some constant $c$, then*

$$\mathbb{P}\left[\mathrm{overlap}(\hat{\pi}, \pi^*) < \delta\right] \geq 1 - \frac{c}{4\delta}.$$

Proposition 3 provides an impossibility result for partial recovery, highlighting the relationship between the recovery probability and the threshold. This result also extends to the exact recovery regime when $\delta = 1$. The following proposition strengthens this conclusion under an assumption for the exact recovery regime, achieving both vanishing error and a sharp constant in the threshold.

**Proposition 4** (Impossibility result, exact recovery). *If $n \log\left(\frac{1}{1-\rho^2}\right) + d \log\left(\frac{1}{1-r^2}\right) + 4 \log d \leq (4 - \epsilon) \log n$ for some constant $\epsilon > 0$ under the assumption $r^2 \geq \frac{40}{d}$, then for any estimator $\hat{\pi}$,*

$$\mathbb{P}\left[\hat{\pi} \neq \pi^*\right] = 1 - o(1).$$

By Proposition 4, we derive sharp information-theoretic thresholds for exact recovery with a gap of $4 \log d$. The proofs of Propositions 3 and 4 are deferred to Appendices C.3 and C.4, respectively.

# 4 QPAlign: Quadratic Programming relaxation for attributed graph Alignment

In Sections 2 and 3, we have shown that the MLE in equation 2 achieves the optimal information-theoretic thresholds. However, this estimator requires an exhaustive search over all possible mappings in $\mathcal{S}_n$, which has a runtime of order $n!$. To address this computational bottleneck, we propose QPAlign, an approximation algorithm for attributed graph alignment.

Let $[n] \triangleq \{1, 2, \cdots, n\}$. Without loss of generality, we assume $V(G_1) = V(G_2) = [n]$, $\pi : [n] \mapsto [n]$ and $E(G_1) = E(G_2) = \{(i,j) : 1 \leq i < j \leq n\}$. Let $\Pi$ be the permutation matrix of $\pi$ with $\Pi_{ij} = \mathbf{1}_{\{\pi(i)=j\}}$ for any $1 \leq i, j \leq n$, and define $\lambda \triangleq \frac{\varphi(\rho)}{\varphi(\rho)+\varphi(r)}$. Then the MLE $\hat{\pi}$ in equation 2 is equivalent to minimizing

$$\lambda \sum_{1 \leq i < j \leq n} \left( \beta_{ij}(G_1) - \beta_{\pi(i)\pi(j)}(G_2) \right)^2 + (1-\lambda) \sum_{1 \leq i \leq n} \|\boldsymbol{x}_i - \boldsymbol{y}_{\pi(i)}\|^2. \tag{5}$$

Denote $A_1, A_2$ as the adjacent matrices of $G_1, G_2$. Let $B_1^i = \mathrm{diag}\{\boldsymbol{x}_{1i}, \boldsymbol{x}_{2i}, \cdots, \boldsymbol{x}_{ni}\}$ and $B_2^i = \mathrm{diag}\{\boldsymbol{y}_{1i}, \boldsymbol{y}_{2i}, \cdots, \boldsymbol{y}_{ni}\}$ for any $i \in [d]$, where $\boldsymbol{x}_{ki}$ corresponds to the $i-$component of vector $\boldsymbol{x}_k$. Then equation 5 is equivalent to

$$\lambda\|A_1 - \Pi A_2\Pi^\top\|_F^2 + (1-\lambda)\sum_{i=1}^d \|B_1^i - \Pi B_2^i\Pi^\top\|_F^2$$

$$= \lambda\|A_1\Pi - \Pi A_2\|_F^2 + (1-\lambda)\sum_{i=1}^d \|B_1^i\Pi - \Pi B_2^i\|_F^2 \triangleq f(\Pi), \tag{6}$$

where $\Pi \in \mathbb{P}^n \triangleq \left\{ \mathbf{P} \in \{0,1\}^{n \times n}, \mathbf{P}\mathbf{1} = \mathbf{1}, \mathbf{P}^\top\mathbf{1} = \mathbf{1} \right\}$. Indeed, this is an instance of the *quadratic assignment problem* (QAP) (Pardalos et al., 1994; Burkard et al., 1998), which is NP-hard to solve or to approximate (Makarychev et al., 2010). To obtain a computationally efficient algorithm for estimating $\hat{\pi}$, we employ a relaxation approach. Relaxing the set of permutations to Birkhoff polytope (the set of doubly stochastic matrices)

$$\mathbb{W}^n \triangleq \left\{ \mathbf{W} \in [0,1]^{n \times n} : \mathbf{W}\mathbf{1} = \mathbf{1}, \mathbf{W}^\top\mathbf{1} = \mathbf{1}, 0 \leq \mathbf{W}_{ij} \leq 1 \text{ for all } i,j \right\},$$

we derive the quadratic programming (QP) relaxation

$$\min_{\Pi \in \mathbb{W}^n} \left\{ \lambda\|A_1\Pi - \Pi A_2\|_F^2 + (1-\lambda)\sum_{i=1}^d \|B_1^i\Pi - \Pi B_2^i\|_F^2 \right\}. \tag{7}$$

We solve the above quadratic programming by projected gradient descent. Specifically, we project the matrix to $\mathbb{W}_n$ by Euclidean projection: $\Pi^{k+1} = \mathsf{Proj}_{\mathbb{W}_n}(\Pi^k - \eta\nabla f(\Pi^k))$, where $\eta$ is the step size. We have the following Theorem on the convergence guarantee for the gradient descent method.

**Theorem 3.** *For any two graphs $G_1, G_2$, there exists a universal constant $L = L(G_1, G_2)$ such that, for function $f$ defined in equation 6 and any $\eta \leq L^{-1}$, we have*

$$|f(\Pi^K) - f(\Pi')| \leq \frac{1}{2\eta K}\|\Pi^0 - \Pi'\|_F^2 \leq \frac{n}{\eta K}$$

*for any integer $K \geq 1$, where $\Pi' \in \arg\min_{\Pi \in \mathbb{W}^n} f(\Pi)$ and $\Pi^0$ is the initial state.*

Indeed, relaxing to Birkhoff polytope is widely adopted in graph matching (Vogelstein et al., 2015; Bommakanti et al., 2024), and has been proved tight in random graph models (Fan et al., 2023a;b). In practice, we often regard $\lambda$ as a tuning parameter and adapt model selection technique for picking $\lambda$. See Remark 1 for further details. In view of Theorem 3, we obtain a standard $O(1/K)$ convergence guarantee for the projected gradient descent scheme with exact Euclidean projections onto $\mathbb{W}_n$. However, in our implementation, we replace these exact projections by a few iterations of Sinkhorn scaling (Sinkhorn, 1964) as a fast approximate projection onto $\mathbb{W}_n$. In practice, since Sinkhorn requires nonnegative entries, we apply it to the truncated matrix $(\Pi^{(t+1)})_+$, obtained by setting all negative entries of $\Pi^{(t+1)}$ to zero.

Define $D \in \mathbb{R}^{n \times n}$ as $D_{kj} = \|\boldsymbol{x}_k - \boldsymbol{y}_j\|_2^2$. We note that $\sum_{i=1}^d \|B_1^i \Pi - \Pi B_2^i\|_F^2 = \sum_{k,j} D_{kj} \Pi_{kj}^2$. Note that $\sum_{k,j} \Pi_{kj}(1 - \Pi_{kj}) = 0$ for permutation matrix $\Pi$. We turn this constraint into a regularizer with parameter $\mu$ and derive the following program:

$$\min_{\Pi \in \mathbb{W}_n} \left\{ \lambda \|A_1 \Pi - \Pi A_2\|_F^2 \ + \ (1 - \lambda) \sum_{k,j} D_{kj} \Pi_{kj}^2 + \mu \sum_{k,j} \Pi_{kj}(1 - \Pi_{kj}) \right\}. \tag{8}$$

We solve the problem in equation 8 by QPAlign in Algorithm 1. In the following, we outline a general recipe for QPAlign.

- *Gradient descent.* We update $\Pi^{(t+1)} = \Pi^{(t)} - \eta G^{(t)}$ with step size $\eta > 0$, where the gradient is given by

$$G^{(t)} = 2\lambda(A_1^\top E - E A_2^\top) + 2(1 - \lambda)\left(D \circ \Pi^{(t)}\right) + \mu\left(J_{n \times n} - 2\Pi^{(t)}\right),$$

where $E = A_1 \Pi^{(t)} - \Pi^{(t)} A_2$ and $(D \circ \Pi^{(t)})_{ij} = D_{ij} \Pi_{ij}^{(t)}$, $J_{ij} = 1$ for any $i, j$.

- *Sinkhorn normalization.* After each gradient step, update $\Pi^{(t+1)}$ on the set of doubly stochastic matrices $\mathbb{W}^n$ using $K$ iterations of the Sinkhorn normalization procedure (Sinkhorn, 1964).

- *Rounding via Hungarian algorithm.* Once convergence is reached, the final doubly stochastic matrix is converted into a permutation matrix by solving the matching problem $\arg\max_{\pi \in \mathcal{S}_n} \sum_i \Pi_{i,\pi(i)}^{(t)}$ using the Hungarian algorithm (Kuhn, 1955; Munkres, 1957).

**Time complexity** The construction of the feature-distance matrix $D$ requires $O(dn^2)$ time. Each gradient step has a complexity of $O(n^3)$, and the Sinkhorn algorithm with $K$ iterations takes $O(Kn^3)$. Consequently, performing $T$ iterations of gradient descent and Sinkhorn projection costs $O(TKn^3)$. The final rounding via the Hungarian algorithm requires $O(n^3)$ (Munkres, 1957). Overall, the time complexity of QPAlign is $O((d + TKn)n^2)$.

---

**Algorithm 1** QPAlign: Quadratic Programming relaxation for attributed graph Alignment

1: **Input:** Adjacency matrices $A_1, A_2 \in \mathbb{R}^{n \times n}$; node features $\boldsymbol{x}_i, \boldsymbol{y}_i, 1 \le i \le n$; weights $\lambda, \mu > 0$; step size $\eta > 0$; Sinkhorn iters $K$; max iters $T$; tolerance tol.
2: **Output:** Estimated permutation $\hat{\pi}$.
3: Construct feature-distance matrix $D \in \mathbb{R}^{n \times n}$ with $D_{ij} = \|\boldsymbol{x}_i - \boldsymbol{y}_j\|_2^2$.
4: Initialize $\Pi^{(0)}$.
5: **for** $t = 0, \ldots, T - 1$ **do**
6:   $E \leftarrow A_1 \Pi^{(t)} - \Pi^{(t)} A_2$.
7:   $f^{(t)} \leftarrow \lambda \|E\|_F^2 + (1 - \lambda) \sum_{i,j} D_{ij}(\Pi_{ij}^{(t)})^2 + \mu \sum_{i,j} \Pi_{ij}^{(t)}(1 - \Pi_{ij}^{(t)})$.
8:   $G^{(t)} \leftarrow 2\lambda(A_1^\top E - E A_2^\top) + 2(1 - \lambda)\left(D \circ \Pi^{(t)}\right) + \mu\left(J_{n \times n} - 2\Pi^{(t)}\right)$.
9:   Gradient step: $\Pi^{(t+1)} \leftarrow \Pi^{(t)} - \eta G^{(t)}$.
10:   Truncate negative entries: $\Pi^{(t+1)} \leftarrow (\Pi^{(t+1)})_+$, $(\cdot)_+$: elementwise $\max\{\cdot, 0\}$.
11:   Update $\Pi^{(t+1)}$ on the Birkhoff polytope via Sinkhorn: $\Pi^{(t+1)} \leftarrow \textbf{Sinkhorn}((\Pi^{(t+1)})_+, K)$.
12:   **if** $t > 0$ and $|f^{(t)} - f^{(t-1)}| < $ tol **then**
13:     **break**
14:   **end if**
15: **end for**
16: Round to a permutation via Hungarian algorithm: $\hat{\pi} \leftarrow \arg\max_{\pi \in \mathcal{S}_n} \sum_i \Pi_{i,\pi(i)}^{(t)}$.
17: **return** $\hat{\pi}$.

---

**Remark 1** (Tuning parameter). *In practice, when solving the quadratic program in equation 8, the parameter $\lambda$ is typically difficult to estimate from the observations $G_1$ and $G_2$. Hence, both $\lambda$ and $\mu$ are usually treated as tuning parameters. One strategy for selecting them is cross-validation. Although we only have a single graph pair $(G_1, G_2)$, one can apply the algorithm to several randomly sampled subgraphs and choose the values of $\lambda$ and $\mu$ that minimize the average validation loss.*

**Remark 2** (Regularization term). *We add a regularization term $\sum_{k,j} \Pi_{kj}(1 - \Pi_{kj})$ in program 8. This term encourages the entries of $\Pi$ to approach 0 or 1. While some previous works employ a penalty of the form $\|\Pi\|_F^2$ to obtain an explicit solution (see, e.g. Fan et al. (2020)), our relaxation directly pushes the solution toward the boundary of the Birkhoff polytope. As a result, the intermediate matrix $\Pi^{(t)}$ becomes more concentrated near the vertices of the Birkhoff polytope, which allows the final rounding via the Hungarian algorithm to be more precise. Therefore, the inclusion of this regularization term helps to improve the overall accuracy of the estimated permutation $\hat{\pi}$.*

## 5 NUMERICAL RESULTS

### 5.1 SIMULATION STUDIES

In this subsection, we provide numerical results for QPAlign in Algorithm 1 on synthetic data. One related model is the correlated Gaussian-attributed Erdős-Rényi model (Yang & Chung, 2024), where the correlated pairs of edges follow a multivariate Bernoulli distribution with connection probability $p$ and correlation $\rho$. Fixing $n = 3000$ and $d = 512$ (with $p = 0.5$ in the Erdős-Rényi case), we run the algorithm and report the value of $\text{overlap}(\hat{\pi}, \pi^*)$ for varying correlations $\rho \in [0, 1]$ and $r \in [0, 1]$. We evaluate our method with step size $\eta = 10^{-4}$, $T = 400$, $K = 80$, $\lambda = 0.1$ and $\mu = 0.1$ for the experiments reported here.

Our results are summarized in Figure 1. Figure 1a displays the heatmap of the overlap under the featured correlated Gaussian Wigner model. We observe that the overlap increases smoothly with both $\rho$ and $r$, starting from nearly zero when both correlations vanish and approaching one when both correlations are close to unity. This indicates that the estimator $\hat{\pi}$ gradually aligns with the ground truth $\pi^*$ as the signal in either edges or features becomes stronger. For the low-dimensional setting with $n = 100$ and $d = 16$, we present additional results in Figure 4 in Appendix A.1, which exhibit the same qualitative trend, confirming the effectiveness of our method in both regimes. Importantly, these numerical results are consistent with the information-theoretic exact recovery thresholds given in Theorem 2. We also note that in certain intermediate regimes, there exists a statistic-computation gap: while exact recovery is theoretically possible, computationally achieving it may require stronger correlations. See Figure 5 in Appendix A.1 for a more detailed comparison between the information-theoretic thresholds and the empirical phase-transition boundaries of QPAlign.

Figure 1b shows the corresponding result under the featured correlated Erdős-Rényi model with $p = 0.5$. A qualitatively similar pattern is observed: the algorithm achieves high overlap once either $\rho$ or $r$ is sufficiently large. Together, these experiments confirm that our algorithm behaves stably across different correlation regimes and successfully interpolates between weak and strong signal cases, with $\text{overlap}(\hat{\pi}, \pi^*)$ ranging from 0 to 1 as $(\rho, r)$ varies from $(0, 0)$ to $(1, 1)$. The results show that our approach is effective for both weighted and unweighted graphs, which broadens the applicability relative to prior methods. See more comparisons with the benchmarks in Appendix A.1.

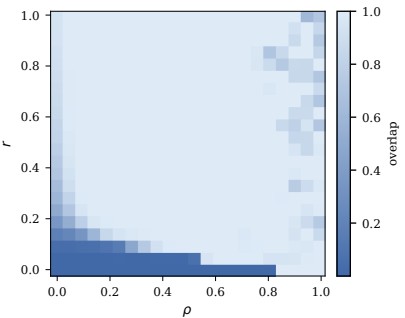 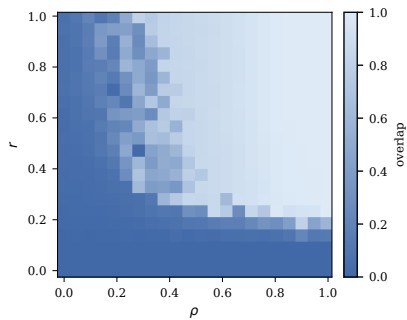

(a) Featured correlated Gaussian Wigner model with $n = 3000$ and $d = 512$.

(b) Featured correlated Erdős-Rényi model with $n = 3000$, $d = 512$, and $p = 0.5$.

Figure 1: Overlap between the estimator $\hat{\pi}$ in Algorithm 1 and the ground truth $\pi^*$ in two models with $n = 3000$ and $d = 512$, evaluated across varying correlations $\rho \in [0, 1]$ and $r \in [0, 1]$.

## 5.2 REAL DATA ANALYSIS

**ACM-DBLP dataset** The ACM-DBLP dataset (Tang et al., 2008) is a widely used benchmark for attributed graph alignment, containing 2,224 ground-truth matched pairs. In our construction, each node represents a paper from either the ACM or DBLP source, and edges are weighted by co-authorship relations. The ground-truth alignment is given by the set of papers appearing in both sources. We implemented the experiments with hyperparameters $\mu = 0.01$, $T = 1000$, $K = 200$, step size $\eta = 10^{-5}$, and $\lambda \in \{0, 0.2, 0.4, 0.6, 0.8, 1\}$.

**Douban (Online–Offline) dataset** The Douban Online–Offline dataset (Tang et al., 2008) is another widely used benchmark for attributed graph alignment, consisting of two graphs that share 1,118 ground-truth matched pairs. Each node represents a user, with edges in the online graph encoding platform interactions (e.g., replying to a post) and edges in the offline graph capturing co-attendance at social events. Node features are given by user locations. The online graph strictly contains all users from the offline graph, and the ground-truth alignment is defined by the users present in both. We implemented the experiments with hyperparameters $\mu = 0$, $T = 1000$, $K = 200$, step size $\eta = 5 \times 10^{-3}$, and $\lambda \in \{0, 0.2, 0.4, 0.6, 0.8, 1\}$.

Indeed, the ACM-DBLP dataset corresponds to a featured Gaussian-Wigner graph, while the Douban dataset is treated as a featured Erdős-Rényi graph, each representing different structural settings for graph alignment. We compare our method with three types of baselines: 1) based solely on edge structure (Grampa (Fan et al., 2023a), IsoRank (Singh et al., 2008), Umeyama (Umeyama, 1988), GW (Peyré et al., 2016)); 2) based solely on node features (MAP (Dai et al., 2019a), kNN); and 3) exploiting both edge structure and node features (FGW (Titouan et al., 2019), REGAL (Heimann et al., 2018), PARROT (Zeng et al., 2023)). To ensure a fair comparison, we follow the official implementations and parameter choices recommended in the original papers. Since the baselines are designed for different settings (edge-only, feature-only, or joint), the results should be viewed within their respective information settings rather than as direct head-to-head comparisons.

The results are reported in Figure 2 and Table 1. We report the experimental results averaged over 5 random seeds. The faint curves represent the results of individual runs, while the bold curves show their average. To fairly compare with baselines using the corresponding information source in Table 1, we conducted two ablated versions: $\lambda = 0$, which corresponds to using only feature information, and $\lambda = 1$, which corresponds to using only edge information. The results in Figure 2 and Table 1 both demonstrate that combining the two sources of information yields performance that surpasses relying on either source alone. See more details in Appendix A.2.

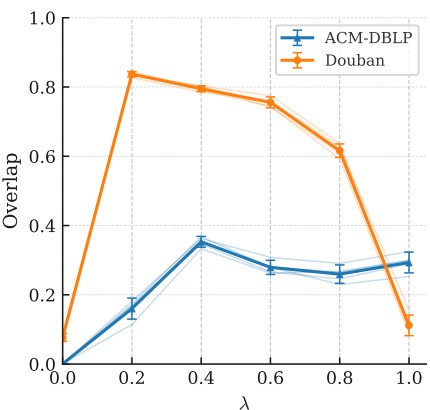

| | ACM-DBLP | Douban |
|---|---|---|
| QPAlign (max) | **0.3445** | 0.8370 |
| FGW | 0.0018 | 0.2773 |
| REGAL | 0.0301 | 0.1118 |
| PARROT | 0.0441 | **0.8462** |
| QPAlign ($\lambda = 0$) | 0.0004 | 0.0767 |
| MAP | 0.0004 | 0.0411 |
| kNN | 0.0004 | 0.0725 |
| QPAlign ($\lambda = 1$) | 0.2896 | 0.1118 |
| Grampa | 0.0746 | 0.0027 |
| IsoRank | 0.0018 | 0.0000 |
| Umeyama | 0.0346 | 0.0089 |
| GW | 0.0202 | 0.0000 |

Figure 2: Overlap vs. $\lambda$ on ACM-DBLP and Douban datasets.

Table 1: Experimental results in alignment across ACM-DBLP and Douban datasets.

**Spatial transcriptomic dataset** Spatial transcriptomic (ST) data (Ståhl et al., 2016) consist of gene expression profiles measured at spatially localized spots on a tissue slice, where each feature corresponds to a gene and the spatial coordinates represent the physical locations of the spots. We

use an ST slice containing 255 spots with 7,998 gene features as the base dataset. To enable quantitative evaluation with ground-truth correspondences, we generate simulated slices by rotating the spot coordinates and resampling expression counts after adding a pseudocount $\delta$ to each gene in each spot. We consider five noise levels ($\delta \in 0, 1, 2, 3, 4, 5$) to model increasing experimental variability. The detailed procedure of data construction is provided in Appendix A.3.

In the experiment, we further examined the sensitivity of our method with respect to $\lambda$. Recall that $\lambda = 0$ corresponds to using only the feature information, while $\lambda = 1$ corresponds to relying solely on the structural information. Across the entire range of $\lambda$, our method consistently achieved strong alignment performance, indicating remarkable robustness to the choice of $\lambda$. In comparison with widely adopted baselines for spatial transcriptomics data alignment, including BBKNN (Polański et al., 2020) and Harmony (Korsunsky et al., 2019), our approach yields consistently superior results. We implemented the experiments with parameters $T = 400$, $K = 80$, $\mu = 0.1$, $\eta = 10^{-5}$.

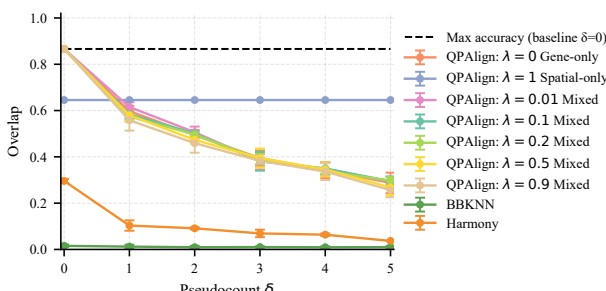

Figure 3: Alignment accuracy across different values of $\lambda$ in ST data

## 6 CONCLUSIONS

In this paper, we study the graph alignment problem under a featured Gaussian Wigner model, where two graphs follow a Gaussian Wigner model and each node is equipped with correlated Gaussian feature vectors. We derive rate-optimal information–theoretic thresholds for both partial and exact recovery in this setting. In addition, we propose QPAlign, a fast algorithm based on a quadratic programming relaxation over the Birkhoff polytope. Our main contributions are as follows.

- *Information-theoretic thresholds.* We establish the optimal rates for both partial recovery and exact recovery regimes. Furthermore, in the regime where $d = n^{o(1)}$ and $d = \omega(\log n)$, our thresholds for exact recovery are sharp up to a constant factor. The results are mainly depends on the analysis of the MLE, where a careful option for the weight on feature and edge is selected to obtain the optimal results.

- *Algorithm and validation on real datasets.* We introduce QPAlign, a quadratic-programming-based relaxation with a penalty term in equation 1 that encourages permutation-like solutions while remaining computationally efficient. We demonstrate the effectiveness of QPAlign on both synthetic and real attributed networks, where it achieves high-quality alignments and compares favorably with existing baselines.

Beyond the main contributions, there are also several important directions merit further explorations.

- *Dependencies between edges and features.* In our featured Gaussian Wigner model, the structural and feature channels are modeled in a relatively simple way. One generalization is to allow richer joint distributions in which edges and node features are generated from a common latent source, so that the edge variables may explicitly depend on the features.

- *Generalization to other graph models.* A natural direction is to extend the framework to more realistic random graph models with attributes, such as inhomogeneous Erdős-Rényi graphs (Ding et al., 2025), stochastic block models with community structure (Onaran et al., 2016; Yang & Chung, 2025), or more general graphon models (Gao et al., 2015). It is of interest to understand whether similar results continue to hold in these broader settings.

ETHICS STATEMENT

Our research is entirely theoretical and experimental on synthetic data and publicly available datasets. We are not aware of any ethical concerns related to discrimination, privacy, security, or misuse. All results are presented with academic integrity, and there are no conflicts of interest or sponsorship issues associated with this work.

REPRODUCIBILITY STATEMENT

We have made every effort to ensure the reproducibility of our results. Theoretical claims are supported by complete proofs provided in the appendix. Experimental details, including data generation procedures, model parameters, and evaluation metrics, are fully described in the main text and supplementary materials. To further facilitate verification, we submit a complete package of our implementation and experiment scripts as supplementary material, which allows others to reproduce all results reported in this paper.

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

## USE OF LARGE LANGUAGE MODELS (LLMs)

We acknowledge the use of Large Language Models (LLMs) solely as an auxiliary tool for polishing the language and improving the clarity of exposition. No part of the research ideation, methodology design, experimental implementation, or result analysis relied on LLMs. All technical content, proofs, and experiments were conceived, conducted, and validated entirely by the authors. The authors take full responsibility for the content of this paper.

## A  EXPERIMENTAL DETAILS

### A.1  SYNTHETIC DATA

Figure 4 reports the results for the low-dimensional setting with $n = 100$ and $d = 16$. We again observe that the overlap increases monotonically with both $\rho$ and $r$, starting near zero when both correlations vanish and approaching one when either correlation becomes large. These results mirror the high-dimensional case in Figure 1, thereby confirming that our method remains effective in both low-dimensional and high-dimensional regimes. Moreover, the numerical behavior aligns with the theoretical thresholds established in Theorem 2, while also highlighting the presence of a statistic–computation gap in certain intermediate regimes. In particular, Figure 5 illustrates how the empirical phase-transition boundaries of QPAlign, for different choices of $\lambda \in \{0.1, 0.2, \ldots, 0.9\}$, closely track the information-theoretic limit, thereby providing a direct connection between the algorithm's empirical success and our derived thresholds.

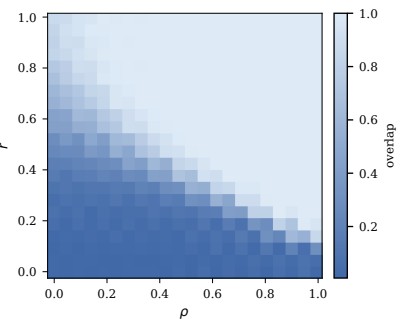
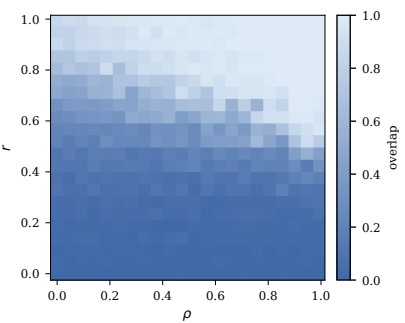

(a) Featured correlated Gaussian Wigner model with $n = 100$ and $d = 16$.

(b) Featured correlated Erdős-Rényi model with $n = 100$, $d = 16$, and $p = 0.5$.

Figure 4: Overlap between the estimator $\hat{\pi}$ in Algorithm 1 and the ground truth $\pi^*$ in two models with $n = 100$ and $d = 16$, evaluated across varying correlations $\rho \in [0, 1]$ and $r \in [0, 1]$.

To facilitate a comprehensive comparison, we evaluate our approach against FGW with various fixed values of $r$, as well as against purely topology-based methods, including GW, Grampa, IsoRank, and Umeyama. All evaluations are conducted on synthetic featured Gaussian–Wigner graphs with $n = 100$ vertices and $d = 16$ dimensions. For the correlation parameter, we report results in terms of $\sigma = \sqrt{(1 - \rho^2)/\rho^2} \in [0, 0.5]$, which serves as a noise-to-signal ratio relative to the original graph, as showed in Figure 6. We adopt $\sigma$ instead of $\rho$ since several algorithms exhibit sharp performance transitions when $\rho \approx 1$ (i.e., $\sigma \approx 0$), making results easier to interpret under this reparametrization.

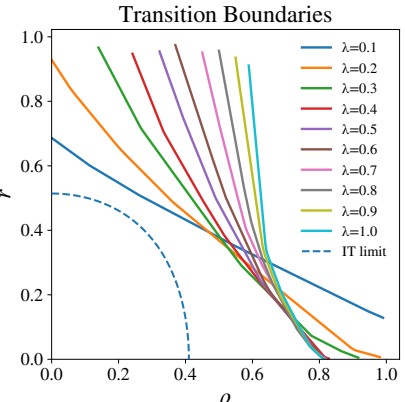

Figure 5: Phase-transition boundaries of QPAlign under different regularization parameters $\lambda$, together with the information-theoretic exact recovery limit.

In addition, we compare with FGW at fixed $\rho$ and with MAP across different values of $r \in [0, 1]$ in Figure 7. Because MAP degenerates and becomes numerically unstable at $r = 1$, we replace the endpoint with $r = 0.999$ for all methods to ensure consistent and stable evaluation. In the synthetic data experiments presented here, our method is evaluated with step size $\eta = 10^{-4}$, $T = 400$, $K = 80$, $\lambda = 0.1$, and $\mu = 0.1$.

Figure 6 reports the alignment accuracy as a function of $\sigma = \sqrt{(1 - \rho^2)/\rho^2}$ with $\lambda = 0.05, 0.1$, and $0.15$, respectively, where smaller $\sigma$ corresponds to stronger graph correlation. Our method consistently outperforms the purely edge-based baselines (GW, Grampa, IsoRank, Umeyama) and the joint edge–feature baseline FGW across different feature correlations $r$. Notably, even when $r$ is small (e.g., $r = 0.1$), our approach achieves higher overlap than FGW under the same setting, indicating robustness to weak feature correlation. By contrast, classical spectral and matching-based methods (Grampa, IsoRank, Umeyama, GW) quickly degrade as noise increases.

Figure 7 shows the overlap as a function of feature correlation $r$ with $\lambda = 0.05, 0.1$, and $0.15$, respectively, under different edge correlations $\rho$. Our method again demonstrates superior performance, achieving near-perfect alignment at much smaller $r$ compared to FGW and MAP. For example, with $\rho = 0.8$, our method reaches almost perfect overlap already at $r = 0.2$, whereas FGW and MAP require significantly larger $r$ to attain comparable accuracy. Overall, except for the degenerate case $r = 0$ or $\rho = 0$, our method consistently achieves higher overlap than existing baselines under the same $r$ or $\rho$.

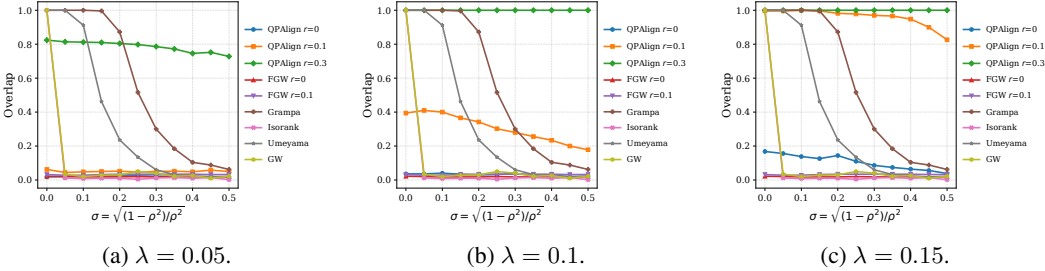

(a) $\lambda = 0.05$.  (b) $\lambda = 0.1$.  (c) $\lambda = 0.15$.

Figure 6: Overlap between the estimator $\hat{\pi}$ in Algorithm 1 and the ground truth $\pi^*$ evaluated by different algorithms across varying correlations $\sigma \in [0, 0.5]$ with different $\lambda$.

To initialize for synthetic datasets, we leverage both feature and degree information to construct a mixed similarity matrix. Specifically, given node feature matrices $X = [\boldsymbol{x}_1^\top, \cdots, \boldsymbol{x}_n^\top] \in \mathbb{R}^{n \times d}$ and $Y = [\boldsymbol{y}_1^\top, \cdots, \boldsymbol{y}_n^\top] \in \mathbb{R}^{n \times d}$, we first compute a feature similarity matrix as $S_{\text{feat}} = \max(XY^\top, 0)$, i.e., the inner product similarity clamped elementwise at zero. We then compute a degree similarity matrix by setting $d_1 = A_1 \mathbf{1}$ and $d_2 = A_2 \mathbf{1}$, and defining $S_{\text{deg}} = (1 + |d_1 \mathbf{1}^\top - \mathbf{1} d_2^\top|)^{-1}$. These two

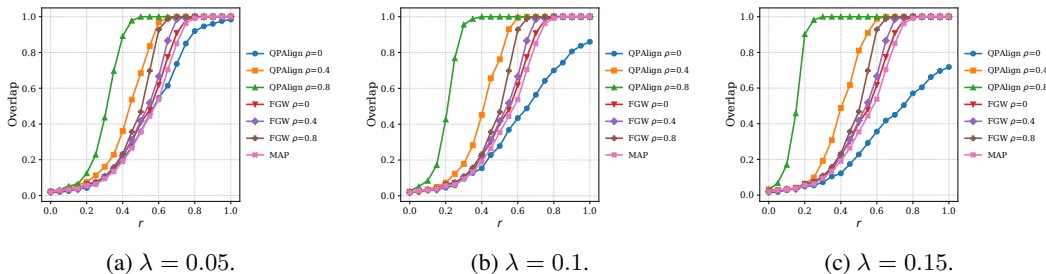

Figure 7: Overlap between the estimator $\hat{\pi}$ in Algorithm 1 and the ground truth $\pi^*$ evaluated by different algorithms across varying correlations $r \in [0,1]$ with different $\lambda$.

components are combined into a mixed similarity matrix, $S = S_{\text{feat}} + \nu S_{\text{deg}}$, which balances feature and structural signals. We empirically set $\nu = 0.1$. The initial transport plan $\Pi^{(0)}$ is then obtained by applying the Sinkhorn algorithm with $K$ iterations to $S$. Across all datasets, we further employ the Barzilai–Borwein (BB) step-size rule (Barzilai & Borwein, 1988) to adaptively determine the learning rate for the gradient descent updates.

### A.2 ACM-DBLP AND DOUBAN DATASETS

We introduce the construction of edges and features in the ACM-DBLP dataset as follows:

- *Features.* Features are constructed from authors and venues only, while paper titles are discarded. Author strings are lowercased, split on commas or semicolons, and tokenized by collapsing spaces into underscores. Venue names are tokenized into words, and merged phrase tokens are created. We use a pretrained RoBERTa model (Liu et al., 2021) to obtain embeddings of the corpus, and all representations are reduced to $d = 256$ dimensions via PCA before whitening to zero mean and unit variance.

- *Edges.* The graph is constructed by treating papers as nodes with edges defined by co-authorship and same-venue co-occurrence. We assign weights $\alpha = 1.0$ for shared authors and $\beta = 0.5$ for shared venues, the weight edge $\beta_{ij}(G)$ is given by

$$\tilde{\beta}_{ij}(G) = \alpha_1 C_{ij}^{\text{author}} + \alpha_2 C_{ij}^{\text{venue}}, \quad \beta_{ij}(G) = \frac{\tilde{\beta}_{ij}(G) - \mu}{\sigma},$$

where $C_{ij}^{\text{author}}$ and $C_{ij}^{\text{venue}}$ denote co-occurrence counts, and $\mu, \sigma$ are the global mean and standard deviation.

We employ the same Douban dataset used in PARROT and other prior works, without any additional processing.

**Baseline comparison.** For completeness, we note two method-specific adjustments. First, RE-GAL assumes non-negative adjacency, which does not strictly match our setting; we therefore follow the standard workaround of omitting nodes with negative degree when running REGAL. Second, PARROT is designed as an anchor-based semi-supervised method, while our experiments do not assume anchor nodes; in this case, we adopt the ablated variant without anchors described in the original paper.

For ACM-DBLP and Douban datasets, we adopt a random initialization followed by a projection step. In practice, we initialize $\Pi^{(0)}$ with a random matrix (using fixed seeds to ensure reproducibility), and then project it onto the Birkhoff polytope using the Sinkhorn algorithm. This initialization avoids numerical instabilities that may arise from directly relying on feature or degree similarities in high-dimensional sparse settings.

### A.3 SPATIAL TRANSCRIPTOMIC DATA

In this subsection, we describe the experimental setup on simulated spatial transcriptomic data. Synthetic slices were generated by perturbing both spatial coordinates and transcript counts. Let

$(X, Z)$ denote a transcript count matrix $X \in \mathbb{N}^{p \times n}$ and spot coordinates $Z \in \mathbb{R}^{2 \times n}$. To model sectioning variability, each coordinate was rotated by

$$z_i' = R(\theta)z_i, \quad R(\theta) = \begin{bmatrix} \cos\theta & -\sin\theta \\ \sin\theta & \cos\theta \end{bmatrix},$$

where $\theta$ was sampled uniformly from $[0, 2\pi)$. Spots mapped outside the array were discarded to mimic tissue loss. Pairwise distances $d_{ik} = \|z_i - z_k\|_2$ were used to ensure invariance to global translation and rotation.

Transcript counts were perturbed in two stages. First, spot-level UMI counts $N_i$ were sampled from a negative binomial distribution $N_i \sim \text{NB}(r, p)$ with mean $\mu$ and variance $\sigma^2$, modeling over-dispersion. Second, given $N_i$, gene-level counts were drawn from a multinomial distribution with probabilities

$$\pi_i = \frac{x_i + \delta}{\|x_i + \delta\|_1},$$

where the pseudocount $\delta$ controls smoothing: small $\delta$ preserves heterogeneity, while large $\delta$ yields more uniform profiles. After rotation, duplicate grid positions were resolved by keeping one spot per location, preserving a one-to-one mapping. Besides, Z-score transformation was applied to both edge and feature information, and $L^2$ normalization was performed on the feature vectors to better align with our theoretical settings.

This perturbation procedure—combining rigid-body rotation, spot loss, and controlled count noise—produces slices that retain biological structure while reflecting realistic variability. It ensures that slices retain key biological structures while reflecting realistic experimental variability such as tissue dropout and technical noise. For example, Zeira et al. (2022) used a similar framework to study alignment robustness under geometric perturbations. We evaluated our method across different values of $\lambda$, where $\lambda = 0$ and $\lambda = 1$ correspond to feature-only and structure-only settings, respectively. The results show that our approach consistently balances edge and feature information, achieving robust performance superior to BBKNN (Polański et al., 2020) and Harmony (Korsunsky et al., 2019).

Finally, we introduce our intialization steps. We exploit domain-specific structure for initialization. Similar to synthetic datasets, we construct $S_{\text{feat}}$ and $S_{\text{deg}}$, combine them with $\nu = 0.1$, and apply the Sinkhorn algorithm with $K$ iterations to obtain $\Pi^{(0)}$.

# B PROOF OF THEOREMS

## B.1 PROOF OF THEOREM 1

The impossibility result directly follows from Proposition 3. We then show the possibility result. By Proposition 1,

$$\mathbb{P}\left[\mathsf{d}(\hat{\pi}, \pi^*) = k\right] \leq \exp\left(-nh\left(\frac{k}{n}\right)\right)\mathbf{1}_{\{k \leq n-1\}} + \exp\left(-2\log n\right)\mathbf{1}_{\{k=n\}} + \exp\left(-\frac{\epsilon k \log n}{16}\right).$$

Summing over $k \geq (1-\delta)n$, we have

$$\sum_{k=(1-\delta)n}^{n-1} \exp\left(-nh\left(\frac{k}{n}\right)\right) \leq \sum_{k=1}^{n-1} \exp\left(-nh\left(\frac{k}{n}\right)\right)$$

$$\overset{(a)}{\leq} 2 \sum_{1 \leq k \leq n/2} \exp\left(-k\log\frac{n}{k}\right)$$

$$\leq 2\sum_{k=1}^{10\log n} \exp(-k\log\frac{n}{k}) + 2\sum_{10\log n \leq k \leq n/2} 2^{-k}$$

$$\leq 2e^{-\log n} \cdot 10\log n + 4 \cdot 2^{-10\log n} = n^{-1+o(1)},$$

where (a) follows from $h(x) = h(1-x)$ and $h(x) \geq x \log \frac{1}{x}$. Since

$$\sum_{k \geq (1-\delta)n} \exp\left(-\frac{1}{32} \epsilon k \log n\right) \leq \frac{\exp\left(-\frac{\epsilon}{32}(1-\delta)n \log n\right)}{1 - \exp\left(-\frac{\epsilon}{32} \log n\right)} = n^{-\Omega(n)},$$

we obtain that

$$\sum_{k=(1-\delta)n}^{n} \mathbb{P}\left[\mathsf{d}(\hat{\pi}, \pi^*) = k\right]$$

$$\leq \sum_{k=(1-\delta)n}^{n} \left[\exp\left(-nh\left(\frac{k}{n}\right)\right) \mathbf{1}_{\{k \leq n-1\}} + \exp\left(-2 \log n\right) \mathbf{1}_{\{k=n\}} + \exp\left(-\frac{\epsilon k \log n}{32}\right)\right]$$

$$\leq n^{-1+o(1)} + n^{-2} + n^{-\Omega(n)}. \tag{9}$$

Since $\mathbb{P}\left[\text{overlap}(\hat{\pi}, \pi^*) \geq \delta\right] \geq 1 - \sum_{k=(1-\delta)n}^{n} \mathbb{P}\left[\mathsf{d}(\hat{\pi}, \pi^*) = k\right]$, we finish the proof of Theorem 1.

### B.2 PROOF OF THEOREM 2

The impossibility results directly follows from Proposition 4. When $n \log\left(\frac{1}{1-\rho^2}\right) + d \log\left(\frac{1}{1-r^2}\right) \geq (4+\epsilon) \log n$, we have

$$n \log\left(\frac{1}{1-\rho^2}\right) + 2d \log\left(\frac{1}{1-r^2}\right) \geq (4+\epsilon) \log n,$$

and thus equation 9 holds for $\delta = 1 - \frac{\epsilon}{16}$. It remains to upper bound $\sum_{k=1}^{\epsilon n/16} \mathbb{P}\left[\mathsf{d}(\hat{\pi}, \pi^*) = k\right]$. By Proposition 2,

$$\sum_{k=1}^{\frac{\epsilon n}{16}} \mathbb{P}\left[\mathsf{d}(\hat{\pi}, \pi^*) = k\right] \leq \sum_{k=1}^{\frac{\epsilon n}{16}} \exp\left(-\frac{\epsilon}{8} k \log n\right) \leq \frac{\exp\left(-\epsilon \log n/8\right)}{1 - \exp\left(-\epsilon \log n/8\right)}.$$

Combining this with equation 9, we finish the proof of Theorem 2.

### B.3 PROOF OF THEOREM 3

Let $L \triangleq 2\left\{\lambda(\|A_1\|^2 + \|A_2\|^2)^2 + (1-\lambda) \sum_{i=1}^{d}(\|B_1^i\|_2 + \|B_2^i\|_2)^2\right\}$. We first show that $\nabla f$ is $L$-Lipschitz. Define the linear operator

$$T(X) \triangleq \left(\sqrt{\lambda}(A_1 X - X A_2), \sqrt{1-\lambda}(B_1^1 X - X B_2^1), \cdots, \sqrt{1-\lambda}(B_1^d X - X B_2^d)\right),$$

then $f(\Pi) = \|T(\Pi)\|_F^2 = \langle \Pi, T^* T \Pi \rangle$ and $\nabla f(\Pi) = 2T^* T(\Pi)$, where $T^*$ denotes the adjoint of $T$ with respect to the Frobenius inner product $\langle X, Y \rangle_F = \text{tr}(X^\top Y)$. Therefore,

$$\|\nabla f(X) - \nabla f(Y)\|_F \leq 2\|T\|^2 \|X - Y\|_F. \tag{10}$$

For each component of $T(X)$, $\|A_1 X - X A_2\|_F \leq (\|A_1\|_2 + \|A_2\|_2)\|X\|_F$, and similarly, $\|B_1^i X - X B_2^i\|_F \leq (\|B_1^i\|_2 + \|B_2^i\|_2)\|X\|_F$, which implies

$$\|T\|^2 \leq \lambda(\|A_1\|^2 + \|A_2\|^2)^2 + (1-\lambda) \sum_{i=1}^{d}(\|B_1^i\|_2 + \|B_2^i\|_2)^2.$$

Combining this with equation 10, we conclude that $\nabla f$ is $L$-Lipschitz.

Recall that $\Pi^{k+1} = \text{Proj}_{\mathbb{W}^n}(\Pi^k - \eta \nabla f(\Pi^k))$. Let $Y^k = \Pi^k - \eta \nabla f(\Pi^k)$. Since

$$\|X - Y^k\|_F^2 = \|X - \Pi^k + \eta \nabla f(\Pi^k)\|_F^2$$

$$= \|X - \Pi^k\|_F^2 + 2\eta \langle \nabla f(\Pi^k), X - \Pi^k \rangle + \eta^2 \|\nabla f(\Pi^k)\|_F^2,$$

we have

$$\langle \nabla f(\Pi^k), X - \Pi^k \rangle + \frac{1}{2\eta}\|X - \Pi^k\|_F^2 = \frac{1}{2\eta}\|X - Y^k\|_F^2 - \frac{\eta}{2}\|\nabla f(\Pi^k)\|_F^2.$$

Therefore, the Euclidean projection

$$\mathsf{Proj}_{\mathbb{W}^n}(\Pi^k - \eta\nabla f(\Pi^k)) = \argmin_{X \in \mathbb{W}^n} \|X - Y^k\|_F^2$$

$$= \argmin_{X \in \mathbb{W}^n} \left\{ \langle \nabla f(\Pi^k), X - \Pi^k \rangle + \frac{1}{2\eta}\|X - \Pi^k\|_F^2 \right\}.$$

Since $\mathbb{W}^n$ is convex, we have that $\Pi^{k+1} + t(\Pi - \Pi^{k+1}) \in \mathbb{W}^n$ for any $t \in (-1, 1)$ and $\Pi \in \mathbb{W}^n$. Since $\Pi^{k+1}$ minimizes $g(X) \triangleq \frac{1}{2}\|X - Y^k\|_F^2$, we have $\frac{d}{dt}g(\Pi^{k+1} + t(\Pi - \Pi^{k+1}))\big|_{t=0+} \geq 0$, which implies

$$\langle \Pi^{k+1} - Y^k, \Pi - \Pi^{k+1} \rangle \geq 0 \tag{11}$$

for any $\Pi \in \mathbb{W}^n$. Consequently, take $\Pi = \Pi'$ yields that

$$\langle \nabla f(\Pi^k), \Pi^{k+1} - \Pi' \rangle \leq \frac{1}{\eta}\langle \Pi^{k+1} - \Pi^k, \Pi^{k+1} - \Pi' \rangle$$

$$= \frac{1}{2\eta}\left( \|\Pi^k - \Pi'\|_F^2 - \|\Pi^{k+1} - \Pi'\|_F^2 - \|\Pi^{k+1} - \Pi^k\|_F^2 \right) \tag{12}$$

We then establish the upper bound for $|f(\Pi^K) - f(\Pi')|$. For $L-$Lipschitz function $f$, we have

$$f(Y) \leq f(X) + \langle \nabla f(X), Y - X \rangle + \frac{L}{2}\|Y - X\|_F^2, \quad \forall X, Y \in \mathbb{W}^n,$$

which implies

$$f(\Pi^{k+1}) \leq f(\Pi^k) + \langle \nabla f(\Pi^k), \Pi^{k+1} - \Pi^k \rangle + \frac{L}{2}\|\Pi^{k+1} - \Pi^k\|_F^2. \tag{13}$$

We decompose the second term as

$$\langle \nabla f(\Pi^k), \Pi^{k+1} - \Pi^k \rangle = \langle \nabla f(\Pi^k), \Pi^{k+1} - \Pi' \rangle + \langle \nabla f(\Pi^k), \Pi' - \Pi^k \rangle.$$

Since $f$ is convex, $f(\Pi') \geq f(\Pi^k) + \langle \nabla f(\Pi^k), \Pi' - \Pi^k \rangle$. Combining this with equation 12 and equation 13, we obtain that

$$f(\Pi^{k+1}) - f(\Pi') \leq \frac{1}{2\eta}\left( \|\Pi^k - \Pi'\|_F^2 - \|\Pi^{k+1} - \Pi'\|_F^2 - \|\Pi^{k+1} - \Pi^k\|_F^2 \right) + \frac{L}{2}\|\Pi^{k+1} - \Pi^k\|_F^2$$

$$= \frac{1}{2\eta}\left( \|\Pi^k - \Pi'\|_F^2 - \|\Pi^{k+1} - \Pi'\|_F^2 \right) + \frac{L\eta - 1}{2\eta}\|\Pi^{k+1} - \Pi^k\|_F^2$$

$$\leq \frac{1}{2\eta}\left( \|\Pi^k - \Pi'\|_F^2 - \|\Pi^{k+1} - \Pi'\|_F^2 \right),$$

where the last inequality follows from $\eta \leq 1/L$.

Take $\Pi = \Pi^k$ in equation 11, we obtain that

$$\langle \nabla f(\Pi^k), \Pi^{k+1} - \Pi^k \rangle \leq -\frac{1}{\eta}\|\Pi^{k+1} - \Pi^k\|_F^2.$$

Combining this with equation 13, we obtain

$$f(\Pi^{k+1}) - f(\Pi^k) \leq \frac{L\eta - 2}{2\eta}\|\Pi^{k+1} - \Pi^k\|_F^2 \leq 0.$$

Taking sum for $k = 0, 1, \cdots, K - 1$, since $f(\Pi^K) \leq f(\Pi^k)$ for any $k = 0, 1, \cdots K - 1$,

$$K|f(\Pi^K) - f(\Pi')| = K(f(\Pi^K) - f(\Pi')) \leq \sum_{k=0}^{K-1} f(\Pi^k) - Kf(\Pi') \leq \frac{1}{2\eta}\|\Pi^0 - \Pi'\|_F^2.$$

The Birkhoff-von Neumann theorem (see, e.g. (Horn & Johnson, 2012, Theorem 8.7.2)) states that a doubly stochastic matrix is a convex combination of permutation matrices, which implies $\mathbb{W}^n$ is the convex hull of $n \times n$ permutation matrices. For any $P, Q \in \mathbb{W}^n$, $(P, Q) \mapsto \|P - Q\|_F^2$ is convex for each variable, hence the maximum admits on the extreme points of $\mathbb{W}^n$, i.e., $P, Q$ are permutation matrices. For permutation matrices $P, Q$, $\operatorname{Tr}(P^\top P) = \operatorname{Tr}(Q^\top Q) = n$. Therefore,

$$\|\Pi^0 - \Pi'\|_F^2 \leq \|P - Q\|_F^2 = \operatorname{Tr}(P^\top P) + \operatorname{Tr}(Q^\top Q) - 2\operatorname{Tr}(P^\top Q) \leq 2n.$$

Consequently,

$$|f(\Pi^K) - f(\Pi')| \leq \frac{1}{2\eta K} \|\Pi^0 - \Pi'\|_F^2 \leq \frac{n}{\eta K}.$$

## C PROOF OF PROPOSITIONS

### C.1 PROOF OF PROPOSITION 1

It is shown in Wu et al. (2022) and Dai et al. (2019a) that when $n \log(1/(1-\rho^2)) \geq (4+\epsilon) \log n$ or $d \log(1/(1-r^2)) \geq (4+\epsilon) \log n$ there exists an estimator $\hat{\pi}$ such that $\mathbb{P}[\hat{\pi} = \pi^*] = 1 - o(1)$. In this paper, we focus on the remaining regime, where $n \log(1/(1-\rho^2)) \vee d \log(1/(1-r^2)) \leq C_0 \log n$ for some universal constant $C_0 > 4$, where $a \vee b = \max(a, b)$ for any $a, b \in \mathbb{R}$. Recall that we assume $d = \omega(\log n)$, which implies that $\rho, r = o(1)$.

For any bijective mappings $\pi' : V(G_1) \mapsto V(G_2)$, let $F_{\pi'} \triangleq \{v \in V(G_1) : \pi^*(v) = \pi'(v)\}$ be the set of fixed points. Recall that $\mathcal{T}_k = \{\pi \in \mathcal{S}_n : \mathsf{d}(\pi, \pi^*) = k\}$ and

$$\{\mathsf{d}(\hat{\pi}, \pi^*) = k\} \subseteq \{\exists \pi' \in \mathcal{T}_k : S_{\pi^*}(G_1, G_2) \leq S_{\pi'}(G_1, G_2)\},$$

where $S_\pi(G_1, G_2) = \varphi(\rho) \sum_{e \in E(G_1)} \beta_e(G_1) \beta_{\pi(e)}(G_2) + \varphi(r) \sum_{v \in V(G_1)} \boldsymbol{x}_v \boldsymbol{y}_{\pi(v)}$. Let $G[F]$ denote the induced subgraph of $G$ over a vertex set $F$. Then for any $\tau \in \mathbb{R}$, we have that

$$\{\mathsf{d}(\hat{\pi}, \pi^*) = k\}$$
$$\subseteq \{\exists \pi' \in \mathcal{T}_k : S_{\pi^*}(G_1, G_2) \leq S_{\pi'}(G_1, G_2)\}$$
$$= \{\exists \pi' \in \mathcal{T}_k : [S_{\pi^*}(G_1, G_2) - S_{\pi^*}(G_1[F_{\pi'}], G_2[F_{\pi'}])] \leq [S_{\pi'}(G_1, G_2) - S_{\pi'}(G_1[F_{\pi'}], G_2[F_{\pi'}])]\}$$
$$\subseteq \{\exists \pi' \in \mathcal{T}_k : S_{\pi^*}(G_1, G_2) - S_{\pi^*}(G_1[F_{\pi'}], G_2[F_{\pi'}]) < \tau\}$$
$$\cup \{\exists \pi' \in \mathcal{T}_k : S_{\pi'}(G_1, G_2) - S_{\pi'}(G_1[F_{\pi'}], G_2[F_{\pi'}]) \geq \tau\}.$$

We then bound the two events separately.

#### C.1.1 BAD EVENT OF WEAK SIGNAL

We first upper bound $\mathbb{P}[\exists \pi' \in \mathcal{T}_k : S_{\pi^*}(G_1, G_2) - S_{\pi^*}(G_1[F_{\pi'}], G_2[F_{\pi'}]) < \tau]$. We write $F = F_{\pi'}$ when $\pi'$ is given. Let $N_k = \binom{n}{2} - \binom{n-k}{2}$. Without loss of generality, we define $E(G_1) \backslash \binom{F}{2} = \{e_1, e_2, \cdots, e_{N_k}\}$ and $V(G_1) \backslash F = \{v_1, v_2, \cdots, v_k\}$. Let $X, Y \in \mathbb{R}^{(N_k+dk)\times 1}$ be defined as

$$X \triangleq \left(\beta_{e_1}(G_1), \beta_{e_2}(G_1), \cdots, \beta_{e_{N_k}}(G_1), \boldsymbol{x}_{v_1}^\top, \boldsymbol{x}_{v_2}^\top, \cdots, \boldsymbol{x}_{v_k}^\top\right)^\top,$$

$$Y \triangleq \left(\beta_{\pi^*(e_1)}(G_2), \beta_{\pi^*(e_2)}(G_2), \cdots, \beta_{\pi^*(e_{N_k})}(G_2), \boldsymbol{y}_{\pi^*(v_1)}^\top, \boldsymbol{y}_{\pi^*(v_2)}^\top, \cdots, \boldsymbol{y}_{\pi^*(v_k)}^\top\right)^\top.$$

Let $W = S_{\pi^*}(G_1, G_2) - S_{\pi^*}(G_1[F], G_2[F])$. Then, we have that $W = X^\top A Y$, where $A = \operatorname{diag}\{\varphi(\rho)I_{N_k}, \varphi(r)I_{dk}\}$ with $\|A\|_F^2 = \varphi(\rho)^2 N_k + \varphi(r)^2 dk$ and $\|A\|_2 = \varphi(\rho) \vee \varphi(r)$. The following Lemma provides concentration for $W$.

**Lemma 2.** *There exists a universal constant $C$, such that with probability at least $1 - \delta_0$,*

$$|W - (\rho\varphi(\rho)N_k + r\varphi(r)kd)| \leq C\left(\|A\|_F \sqrt{\log\frac{1}{\delta_0}} + \|A\|_2 \log\frac{1}{\delta_0}\right).$$

Pick $\tau = (\rho\varphi(\rho)N_k + r\varphi(r)kd) - a_k$, where

$$a_k = \begin{cases} 3C\sqrt{(\rho^2 N_k + r^2 kd)2nh(\frac{k}{n})}, & k \leq n-1, \\ 3C\sqrt{(n\rho^2 + dr^2)n \log n}, & k = n, \end{cases} \tag{14}$$

and $h(x) = -x \log x - (1-x) \log(1-x)$ is the binary entropy function.

**Case 1:** $k = n$.  We choose $\delta_0 = \exp\left(-2\log n\right)$ in Lemma 2. Recall that $\rho, r = o(1)$. Then, with probability at least $1 - \delta_0$,

$$|W - (\rho\varphi(\rho)N_k + r\varphi(r)dk)| \leq C\left[\sqrt{\varphi(\rho)^2 N_k + \varphi(r)^2 dk}\sqrt{2\log n} + ((\varphi(\rho) \vee \varphi(r))2\log n)\right]$$

$$\leq C\sqrt{4n\log n(dr^2 + n\rho^2)} + C\sqrt{n\log n(n\rho^2 + dr^2)}$$

$$= 3C\sqrt{(n\rho^2 + dr^2)n\log n} = a_n,$$

where the last inequality follows from

$$(\varphi(\rho) \vee \varphi(r))2\log n \leq \sqrt{n\rho^2 + dr^2}\log n \leq C\sqrt{n\log n(n\rho^2 + dr^2)}.$$

Consequently, we have $\mathbb{P}\left[W \leq \tau\right] \leq \exp\left(-2\log n\right)$.

**Case 2:** $\delta n \leq k \leq n - 1$.  We choose $\delta_0 = \exp\left(-2nh\left(\frac{k}{n}\right)\right)$ in Lemma 2. Then, with probability $1 - \delta_0$, we have

$$|W - (\rho\varphi(\rho)N_k + r\varphi(r)kd)|$$

$$\leq C\left[\left(\sqrt{\varphi(\rho)^2 N_k + \varphi(r)^2 dk}\sqrt{2nh(\frac{k}{n})}\right) + \left((\varphi(\rho) \vee \varphi(r))2nh(\frac{k}{n})\right)\right]$$

$$\leq C\left(\left(\sqrt{4(\rho^2 N_k + r^2 dk)}\sqrt{2nh(\frac{k}{n})}\right) + \left((\varphi(\rho) \vee \varphi(r))2nh(\frac{k}{n})\right)\right)$$

$$\leq 3C\sqrt{(\rho^2 N_k + r^2 dk)2nh(\frac{k}{n})},$$

where the last inequality follows from

$$(\varphi(\rho) \vee \varphi(r))2nh(\frac{k}{n}) \leq \sqrt{(\rho^2 N_k + r^2 kd)2nh(\frac{k}{n})}\sqrt{\frac{4n(\rho^2 + r^2)}{(\rho^2 N_k + r^2 kd)}}$$

$$\leq \sqrt{(\rho^2 N_k + r^2 kd)2nh(\frac{k}{n})}\sqrt{\frac{8n(\rho^2 + r^2)}{k(\rho^2 n + r^2 d)}},$$

and

$$\frac{8n(\rho^2 + r^2)}{k(\rho^2 n + r^2 d)} \leq \frac{16n}{\delta n(\rho^2 n + r^2 d)} \leq \frac{16n}{\delta n(4 + \epsilon/2)\log n} \leq 1,$$

where the last inequality is because $\rho, r = o(1)$ and $n\log(\frac{1}{1-\rho^2}) + d\log(\frac{1}{1-r^2}) \geq (4 + \epsilon)\log n$ implies $n\rho^2 + dr^2 \geq (4 + \epsilon/2)\log n$. Consequently, $\mathbb{P}\left[W \leq \tau\right] \leq \exp\left(-2nh\left(\frac{k}{n}\right)\right)$ when $k \leq n - 1$. By the union bound, we obtain

$$\mathbb{P}\left[\exists \pi' \in \mathcal{T}_k : S_{\pi^*}(G_1, G_2) - S_{\pi^*}(G_1[F_{\pi'}], G_2[F_{\pi'}]) < \tau\right]$$

$$\leq \mathbb{P}\left[\bigcup_{F \subseteq V(G_1):|F|=n-k} \{S_{\pi^*}(G_1, G_2) - S_{\pi^*}(G_1[F], G_2[F]) < \tau\}\right]$$

$$\leq \binom{n}{k}\mathbb{P}\left[S_{\pi^*}(G_1, G_2) - S_{\pi^*}(G_1[F], G_2[F]) < \tau\right]$$

$$\leq \binom{n}{k}\mathbb{P}\left[W \leq \tau\right]\mathbf{1}_{\{k \leq n-1\}} + \mathbb{P}\left[W \leq \tau\right]\mathbf{1}_{\{k=n\}}$$

$$\leq \exp\left(-nh\left(\frac{k}{n}\right)\right)\mathbf{1}_{\{k \leq n-1\}} + \exp\left(-2\log n\right)\mathbf{1}_{\{k=n\}}, \tag{15}$$

where the last inequality is because $\binom{n}{k} \leq \exp\left(nh\left(\frac{k}{n}\right)\right)$.

### C.1.2 BAD EVENT OF STRONG NOISE

We then upper bound $\mathbb{P}\left[\exists \pi' \in \mathcal{T}_k : S_{\pi'}(G_1, G_2) - S_{\pi'}(G_1[F_{\pi'}], G_2[F_{\pi'}]) \geq \tau\right]$. Given $\pi'$, we define $Z \triangleq S_{\pi'}(G_1, G_2) - S_{\pi'}(G_1[F_{\pi'}], G_2[F_{\pi'}])$. We also write $F = F_{\pi'}$ when $\pi'$ is given. By Chernoff's inequality, for any $t > 0$,

$$\mathbb{P}\left[Z \geq \tau\right] \leq e^{-t\tau} \mathbb{E}\left[e^{tZ}\right].$$

In order to compute the moment generating function $\mathbb{E}\left[e^{tZ}\right]$, we introduce the definition of orbits.

**Cycle decomposition**  For any $\sigma \in \mathcal{S}_n$, it induces a permutation $\sigma^{\mathsf{E}}$ on the edge set $\binom{V(G_1)}{2}$ with $\sigma^{\mathsf{E}}((u,v)) \triangleq (\sigma(u), \sigma(v))$ for $u, v \in V(G_1)$. We refer to $\sigma$ and $\sigma^{\mathsf{E}}$ as a node permutation and edge permutation. Each permutation can be decomposed as disjoint cycles known as orbits. Orbits of $\sigma$ (resp. $\sigma^{\mathsf{E}}$) are referred as *node orbits* (resp. *edge orbits*). For example, a node orbit $(u_1, u_2, \cdots, u_k)$ indicates that $u_{i+1} = \sigma(u_i)$ for $1 \leq i \leq k-1$ and $u_1 = \sigma(u_k)$. Let $n_k$ (resp. $N_k$) denote the number of $k$-node (resp. $k$-edge) orbits in $\sigma$ (resp. $\sigma^{\mathsf{E}}$).

For any $\pi' \in \mathcal{T}_k$, let $\sigma \triangleq (\pi^*)^{-1} \circ \pi'$. Define $\mathcal{C}_i^{\mathsf{V}}$ and $\mathcal{C}_i^{\mathsf{E}}$ the set of *node orbits* and *edge orbits* of length $i$ induced by $\sigma$, respectively. Denote $\mathcal{C}^{\mathsf{V}} = \cup_{i \geq 1}\mathcal{C}_i^{\mathsf{V}}$ and $\mathcal{C}^{\mathsf{E}} = \cup_{i \geq 1}\mathcal{C}_i^{\mathsf{E}}$. Then, $V(G_1) = \cup_{i \geq 1}\left\{v : v \in \mathcal{C}_i^{\mathsf{V}}\right\}$, $E(G_1) = \cup_{i \geq 1}\left\{e : e \in \mathcal{C}_i^{\mathsf{E}}\right\}$, and $\mathcal{C}_1^{\overline{\mathsf{V}}} = F$. Let

$$Z^{\mathsf{E}} = \varphi(\rho) \sum_{e \in E(G_1) \backslash \binom{F}{2}} \beta_e(G_1)\beta_{\pi'(e)}(G_2), \quad Z^{\mathsf{V}} = \varphi(r) \sum_{v \in V(G_1) \backslash F} \boldsymbol{x}_v \boldsymbol{y}_{\pi'(v)}. \tag{16}$$

Then $Z = Z^{\mathsf{V}} + Z^{\mathsf{E}}$. Since $Z^{\mathsf{V}}$ and $Z^{\mathsf{E}}$ are independent, we obtain that

$$\mathbb{E}\left[e^{tZ}\right] = \mathbb{E}\left[e^{tZ^{\mathsf{V}}}\right] \mathbb{E}\left[e^{tZ^{\mathsf{E}}}\right]. \tag{17}$$

We then derive the upper bounds for $\mathbb{E}\left[e^{tZ^{\mathsf{V}}}\right]$ and $\mathbb{E}\left[e^{tZ^{\mathsf{E}}}\right]$, respectively. For any edge cycle $C = \left\{e_1, e_2, \cdots, e_{|C|}\right\}$ with $e_{i+1} = \sigma^{\mathsf{E}}(e_i)$ for all $1 \leq i \leq |C|-1$ and $e_1 = \sigma^{\mathsf{E}}(e_{|C|})$, we define the cumulant generating function as

$$\kappa_{|C|}^{\mathsf{E}}(t) = \log \mathbb{E}\left[\exp\left(t\varphi(\rho) \sum_{i=1}^{|C|} \beta_{e_i}(G_1)\beta_{\pi'(e_i)}(G_2)\right)\right],$$

where we define $(u_{|C|+1}, v_{|C|+1}) = (u_1, v_1)$. Similarly, for any node cycle $C = \left\{v_1, \cdots, v_{|C|}\right\}$, we define $v_{|C|+1} = v_1$ and

$$\kappa_{|C|}^{\mathsf{V}}(t) = \log \mathbb{E}\left[\exp\left(t\varphi(r) \sum_{i=1}^{|C|} \boldsymbol{x}_{v_i} \boldsymbol{y}_{\pi'(v_i)}\right)\right].$$

The lower-order cumulants can be calculated directly:

$$\kappa_1^{\mathsf{E}}(t) = -\frac{1}{2}\log(1 - 2t\rho\varphi(\rho) - t^2\varphi^2(\rho)(1 - \rho^2)),$$

$$\kappa_1^{\mathsf{V}}(t) = -\frac{d}{2}\log(1 - 2tr\varphi(r) - t^2\varphi^2(r)(1 - r^2)),$$

$$\kappa_2^{\mathsf{E}}(t) = -\frac{1}{2}\log(1 - 2t^2\varphi^2(\rho)(1 + \rho^2) + t^4\varphi^4(\rho)(1 - \rho^2)^2),$$

$$\kappa_2^{\mathsf{V}}(t) = -\frac{d}{2}\log(1 - 2t^2\varphi^2(r)(1 + r^2) + t^4\varphi^4(r)(1 - r^2)^2).$$

Let $N_k = \binom{n}{2} - \binom{n-k}{2}$. The following Lemma provides an upper bound on the cumulant function $\log \mathbb{E}\left[\exp(tZ)\right]$.

**Lemma 3.** *If* $\mathsf{d}(\pi^*, \pi') = k$, *for any* $0 < t \leq (\rho^{-1} - 2) \wedge (r^{-1} - 2)$, *we have*

$$\log \mathbb{E}\left[\exp(tZ)\right] \leq \frac{N_k}{2}\kappa_2^{\mathsf{E}}(t) + \frac{k}{2}\left(\kappa_1^{\mathsf{E}}(t) - \frac{1}{2}\kappa_2^{\mathsf{E}}(t) + \frac{1}{2}\kappa_2^{\mathsf{V}}(t)\right).$$

Recall that $\tau = (\rho\varphi(\rho)N_k + r\varphi(r)kd) - a_k$, where

$$a_k = \begin{cases} 3C\sqrt{(\rho^2 N_k + r^2 kd)2nh(\frac{k}{n})}, & k \le n-1 \\ 3C\sqrt{(n\rho^2 + dr^2)n\log n}, & k = n \end{cases} \tag{18}$$

and $h(x) = -x\log x - (1-x)\log(1-x)$ is the binary entropy function. We then show that

$$\left(1 - \frac{\epsilon}{16}\right)(\rho\varphi(\rho)N_k + r\varphi(r)kd) \le \tau \le \rho\varphi(\rho)N_k + r\varphi(r)kd.$$

Recall that $\rho, r = o(1)$. For any $\delta n \le k \le n-1$, since $n\log\left(\frac{1}{1-\rho^2}\right) + 2d\log\left(\frac{1}{1-r^2}\right) \ge (4+\epsilon)\log n$, we have $n\rho^2 + dr^2 \ge (2 + \epsilon/4)\log n$. Therefore, we obtain $\frac{n}{k}h\left(\frac{k}{n}\right) \le \frac{h(\delta)}{\delta} \le \frac{\epsilon^2}{2^{13}C^2}(n\rho^2 + dr^2)$ for sufficiently large $n$. When $k \le n-1$, we have

$$a_k = 2C\sqrt{(\rho^2 N_k + r^2 kd)2nh(\frac{k}{n})} \le 2C\sqrt{(\rho^2 N_k + r^2 kd)\frac{\epsilon^2}{2^{12}C^2}(n\rho^2 + dr^2)k}$$

$$\overset{(a)}{\le} \frac{\epsilon}{32}\sqrt{(\rho^2 N_k + r^2 kd)4(\rho^2 N_k + r^2 kd)} = \frac{\epsilon}{16}(\rho^2 N_k + r^2 kd) \le \frac{\epsilon}{16}(\rho\varphi(\rho)N_k + r\varphi(r)kd),$$

where (a) is because $nk \le 4N_k$. For $k = n$, since $n\log(\frac{1}{1-\rho^2}) + 2d\log(\frac{1}{1-r^2}) \ge (4+\epsilon)\log n$ and $\rho^2, r^2 = o(1)$, we conclude that $n\rho^2 + dr^2 \ge \frac{1}{2}(n\rho^2 + 2dr^2) \ge 2\log n$ holds for sufficiently large $n$. Therefore,

$$a_k = 3C\sqrt{(n\rho^2 + dr^2)n\log n} \le 3C\sqrt{(n\rho^2 + dr^2)\frac{(n\rho^2 + dr^2)n}{2}}$$

$$\le \frac{4C}{\sqrt{n}}(\rho\varphi(\rho)n^2 + r\varphi(r)nd) \le \frac{\epsilon}{16}(\rho\varphi(\rho)N_k + r\varphi(r)kd).$$

Therefore, we conclude

$$\left(1 - \frac{\epsilon}{16}\right)(\rho\varphi(\rho)N_k + r\varphi(r)kd) \le \tau \le \rho\varphi(\rho)N_k + r\varphi(r)kd.$$

We then upper bound $\mathbb{P}[Z \ge \tau]$. We note that

$$\mathbb{P}[Z \ge \tau] \le e^{-t\tau}\mathbb{E}\left[e^{tZ}\right] \le \exp\left(-t\tau + \frac{N_k}{2}\kappa_2^{\mathsf{E}}(t) + \frac{k}{2}\left(\kappa_1^{\mathsf{E}}(t) - \frac{1}{2}\kappa_2^{\mathsf{E}}(t) + \frac{1}{2}\kappa_2^{\mathsf{V}}(t)\right)\right).$$

Pick $t = 1$. Since $\rho, r = o(1)$, we have $\rho^2, r^2 \le \frac{\epsilon}{256}$. Recall that $\varphi(\rho) = \frac{\rho}{1-\rho^2}$. Then,

$$\kappa_1^{\mathsf{E}}(t) - \frac{1}{2}\kappa_2^{\mathsf{E}}(t) = \frac{1}{4}\log\frac{1 - 2t^2\varphi^2(\rho)(1+\rho^2) + t^4\varphi^4(\rho)(1-\rho^2)^2}{(1 - 2t\rho\varphi(\rho) - t^2\varphi^2(\rho)(1-\rho^2))^2}$$

$$= \frac{1}{4}\log\left(1 + \frac{4t\rho^2}{1 - \rho^2(1+t)^2}\right) \le \frac{\rho^2}{1 - 4\rho^2} \le 2,$$

where the last inequality is because $\log(1+x) \le x$ and $\rho < \frac{1}{4}$. We then bound $\kappa_2^{\mathsf{E}}(t)$. We note that

$$\frac{\kappa_2^{\mathsf{E}}(t)}{\rho\varphi(\rho)} = -\frac{1-\rho^2}{2\rho^2}\log(1 - 2t^2\varphi^2(\rho)(1+\rho^2) + t^4\varphi^4(\rho)(1-\rho^2)^2)$$

$$= -\frac{1-\rho^2}{2\rho^2}\log\left(1 - \frac{\rho^2(2+\rho^2)}{(1-\rho^2)^2}\right) \overset{(a)}{\le} 1 + 4\rho^2 \le \left(1 + \frac{\epsilon}{64}\right),$$

where the inequality (a) is from Lemma 6. Hence, we have $\kappa_2^{\mathsf{E}}(t) \le \left(1 + \frac{\epsilon}{64}\right)\rho\varphi(\rho)$. Similarly, we have

$$\kappa_2^{\mathsf{V}}(t) = -\frac{d}{2}\log(1 - 2t^2\varphi^2(r)(1+r^2) + t^4\varphi^4(r)(1-r^2)^2) \le \left(1 + \frac{\epsilon}{64}\right)dr\varphi(r).$$

Therefore, for $t = 1$,

$$- t\tau + \frac{N_k}{2} \kappa_2^{\mathsf{E}}(t) + \frac{k}{2} \left( \kappa_1^{\mathsf{E}}(t) - \frac{1}{2} \kappa_2^{\mathsf{E}}(t) \right) + \frac{k}{2} \kappa_2^{\mathsf{V}}(t)$$

$$\leq - \left( 1 - \frac{\epsilon}{16} \right) \left( \rho \varphi(\rho) N_k + r \varphi(r) k d \right) + \left( \frac{1}{2} + \frac{\epsilon}{128} \right) \left( \rho \varphi(\rho) N_k + k d r \varphi(r) \right) + k$$

$$\leq - \left( \frac{1}{2} - \frac{\epsilon}{32} \right) \left( \rho \varphi(\rho) N_k + r \varphi(r) k d \right) + k$$

$$\leq - \left( \frac{1}{2} - \frac{\epsilon}{32} \right) \left( 2 + \frac{\epsilon}{4} \right) k \log n + k,$$

where the last inequality is because $N_k = \binom{n}{2} - \binom{n-k}{2} \geq \frac{1}{2} \left( 1 - \frac{1}{n} \right) kn$ and

$$\rho \varphi(\rho) N_k + r \varphi(r) k d \geq k \left( \frac{n-1}{2} \log(\frac{1}{1-\rho^2}) + d \log(\frac{1}{1-r^2}) \right) \geq k(2 + \frac{\epsilon}{4}) \log n.$$

Consequently, by the union bound and $|\mathcal{T}_k| = \binom{n}{k} k! \leq n^k$,

$$\mathbb{P} \left[ \exists \pi' \in \mathcal{T}_k : S_{\pi'}(G_1, G_2) - S_{\pi'}(G_1[F_{\pi'}], G_2[F_{\pi'}]) \geq \tau \right]$$

$$\leq n^k \mathbb{P} \left[ Z \geq \tau \right]$$

$$\leq \exp \left( k \log n - \left( \frac{1}{2} - \frac{\epsilon}{32} \right) \left( 2 + \frac{\epsilon}{4} \right) k \log n + k \right) \leq \exp \left( - \frac{\epsilon}{32} k \log n \right).$$

Combining this with equation 15, we obtain that

$$\mathbb{P} \left[ \mathsf{d}(\hat{\pi}, \pi^*) = k \right] \leq \mathbb{P} \left[ \exists \pi' \in \mathcal{T}_k : S_{\pi^*}(G_1, G_2) - S_{\pi^*}(G_1[F_{\pi'}], G_2[F_{\pi'}]) < \tau \right]$$

$$+ \mathbb{P} \left[ \exists \pi' \in \mathcal{T}_k : S_{\pi'}(G_1, G_2) - S_{\pi'}(G_1[F_{\pi'}], G_2[F_{\pi'}]) \geq \tau \right]$$

$$\leq \exp \left( -nh \left( \frac{k}{n} \right) \right) \mathbf{1}_{\{k \leq n-1\}} + \exp \left( -2 \log n \right) \mathbf{1}_{\{k=n\}} + \exp \left( - \frac{\epsilon k \log n}{32} \right).$$

## C.2 Proof of Proposition 2

For any $\pi' \in \mathcal{T}_k$, let

$$Y_{\pi'} \triangleq \varphi(\rho) \sum_{e \in E(G_1) \setminus \mathcal{C}_1^{\mathsf{E}}} \left( \beta_e(G_1) \beta_{\pi'(e)}(G_2) - \beta_e(G_1) \beta_{\pi^*(e)}(G_2) \right)$$

$$+ \varphi(r) \sum_{v \in V(G_1) \setminus \mathcal{C}_1^{\mathsf{V}}} \left( \boldsymbol{x}_v^{\top} \boldsymbol{y}_{\pi'(v)} - \boldsymbol{x}_v^{\top} \boldsymbol{y}_{\pi^*(v)} \right),$$

where $\mathcal{C}_1^{\mathsf{E}}$ and $\mathcal{C}_1^{\mathsf{V}}$ denote the edge orbit and vertex orbit of length 1 induced by $\sigma = (\pi^*)^{-1} \circ \pi'$. For notational simplicity, we write $Y = Y_{\pi'}$ when $\pi'$ is given. Then for any $t > 0$, $\{\mathsf{d}(\hat{\pi}, \pi^*) = k\} \subseteq \{\exists \pi' \in \mathcal{T}_k : Y \geq 0\}$ and

$$\mathbb{P} \left[ \mathsf{d}(\hat{\pi}, \pi^*) = k \right] \leq \mathbb{P} \left[ \exists \pi' \in \mathcal{T}_k : Y \geq 0 \right] \overset{(a)}{\leq} |\mathcal{T}_k| \mathbb{P} \left[ Y \geq 0 \right] \overset{(b)}{\leq} n^k \mathbb{E} \left[ \exp \left( tY \right) \right], \quad (19)$$

where (a) uses union bound and (b) follows from Chernoff's inequality and $|\mathcal{T}_k| = \binom{n}{k} k! \leq n^k$. Let

$$Y^{\mathsf{E}} \triangleq \varphi(\rho) \sum_{e \in E(G_1) \setminus \mathcal{C}_1^{\mathsf{E}}} \left( \beta_e(G_1) \beta_{\pi'(e)}(G_2) - \beta_e(G_1) \beta_{\pi^*(e)}(G_2) \right),$$

$$Y^{\mathsf{V}} \triangleq \varphi(r) \sum_{v \in V(G_1) \setminus \mathcal{C}_1^{\mathsf{V}}} \left( \boldsymbol{x}_v^{\top} \boldsymbol{y}_{\pi'(v)} - \boldsymbol{x}_v^{\top} \boldsymbol{y}_{\pi^*(v)} \right).$$

Then $Y = Y^{\mathsf{E}} + Y^{\mathsf{V}}$ and $\mathbb{E} \left[ \exp(tY) \right] = \mathbb{E} \left[ \exp(tY^{\mathsf{E}}) \right] \mathbb{E} \left[ \exp \left( tY^{\mathsf{V}} \right) \right]$. We then derive the upper bounds for $\mathbb{E} \left[ \exp \left( tY^{\mathsf{V}} \right) \right]$ and $\mathbb{E} \left[ \exp \left( tY^{\mathsf{E}} \right) \right]$, respectively. For any edge cycle $C =$

$\{e_1, e_2, \cdots, e_{|C|}\}$ with $e_{i+1} = \sigma^{\mathsf{E}}(e_i)$ for all $1 \leq i \leq |C| - 1$ and $e_1 = \sigma^{\mathsf{E}}(e_{|C|})$, we define the cumulant generating function as

$$\mu_{|C|}^{\mathsf{E}}(t) = \log \mathbb{E}\left[\exp\left(t\varphi(\rho)\sum_{i=1}^{|C|}\beta_{e_i}(G_1)\beta_{\pi'(e_i)}(G_2) - t\varphi(\rho)\sum_{i=1}^{|C|}\beta_{e_i}(G_1)\beta_{\pi^*(e_i)}(G_2)\right)\right],$$

where we define $(u_{|C|+1}, v_{|C|+1}) = (u_1, v_1)$. Similarly, for any node cycle $C = \{v_1, \cdots, v_{|C|}\}$, we define $v_{|C|+1} = v_1$ and

$$\mu_{|C|}^{\mathsf{V}}(t) = \log \mathbb{E}\left[\exp\left(t\varphi(r)\sum_{i=1}^{|C|}\boldsymbol{x}_{v_i}\boldsymbol{y}_{\pi'(v_i)} - t\varphi(r)\sum_{i=1}^{|C|}\boldsymbol{x}_{v_i}\boldsymbol{y}_{\pi^*(v_i)}\right)\right].$$

The lower-order cumulants can be calculated directly:

$$\mu_2^{\mathsf{E}}(t) = -\frac{1}{2}\log\left(1 + \frac{\rho^2}{1-\rho^2}(4t - 4t^2)\right), \quad \mu_2^{\mathsf{V}}(t) = -\frac{d}{2}\log\left(1 + \frac{r^2}{1-r^2}(4t - 4t^2)\right).$$

Recall that $N_k = \binom{n}{2} - \binom{n-k}{2}$. The following Lemma provides an upper bound on the cumulant function $\log \mathbb{E}\left[\exp\left(tY\right)\right]$.

**Lemma 4.** *If* $\mathsf{d}(\pi^*, \pi') = k$, *for any* $0 < t < 1$, *we have*

$$\log \mathbb{E}\left[\exp\left(tY\right)\right] \leq \frac{1}{2}\left(N_k - \frac{k}{2}\right)\mu_2^{\mathsf{E}}(t) + \frac{k}{2}\mu_2^{\mathsf{V}}(t).$$

Pick $t = \frac{1}{2}$. By Lemma 4,

$$\log \mathbb{E}\left[\exp\left(tY\right)\right] \leq -\frac{nk}{4}\left(1 - \frac{k+2}{2n}\right)\log\left(\frac{1}{1-\rho^2}\right) - \frac{kd}{4}\log\left(\frac{1}{1-r^2}\right).$$

Combining this with equation 19, we have

$$\mathbb{P}\left[\mathsf{d}(\hat{\pi}, \pi^*) = k\right] \leq n^k \mathbb{E}\left[\exp\left(tY\right)\right]$$
$$\leq \exp\left(k\log n - \frac{nk}{4}\left(1 - \frac{k+2}{2n}\right)\log\left(\frac{1}{1-\rho^2}\right) - \frac{kd}{4}\log\left(\frac{1}{1-r^2}\right)\right)$$
$$\leq \exp\left(k\log n - \frac{k}{4}\left(1 - \frac{k+2}{2n}\right)\left(n\log\left(\frac{1}{1-\rho^2}\right) + d\log\left(\frac{1}{1-r^2}\right)\right)\right)$$
$$\stackrel{(a)}{\leq} \exp\left(-\left(\frac{\epsilon}{4} - \frac{k+2}{2n}\left(1 + \frac{\epsilon}{4}\right)\right)k\log n\right) \stackrel{(b)}{\leq} \exp\left(-\frac{\epsilon}{8}k\log n\right),$$

where (a) is because $n\log\left(\frac{1}{1-\rho^2}\right) + d\log\left(\frac{1}{1-r^2}\right) \geq (4+\epsilon)\log n$; (b) follows from $k \leq \frac{\epsilon}{16}n$.

### C.3 PROOF OF PROPOSITION 3

We directly apply Fano's inequality in equation 3. For any $0 < \delta < 1$, by Lemma 1,

$$\mathbb{P}\left[\mathrm{overlap}(\hat{\pi}, \pi^*) < \delta\right] \geq 1 - \frac{I(\pi^*; G_1, G_2) + \log 2}{\log|\mathcal{M}_\delta|}$$
$$\geq 1 - \frac{\binom{n}{2}\frac{1}{2}\log(\frac{1}{1-\rho^2}) + \frac{nd}{2}\log(\frac{1}{1-r^2}) + \log 2}{\delta n \log(\delta n/e)} \geq 1 - \frac{c}{4\delta},$$

where the last inequality follows from $n\log\left(\frac{1}{1-\rho^2}\right) + 2d\log\left(\frac{1}{1-r^2}\right) \leq c\log n$.

## C.4 PROOF OF PROPOSITION 4

In this subsection, we provide the proof on Proposition 4. Recall that $S_\pi(G_1, G_2) = \varphi(\rho) \sum_{e \in E(G_1)} \beta_e(G_1) \beta_{\pi(e)}(G_2) + \varphi(r) \sum_{v \in V(G_1)} \boldsymbol{x}_v \boldsymbol{y}_{\pi(v)}$. Define

$$\mathcal{E}(\pi^*, \pi') \triangleq \{(G_1, G_2) : S_{\pi*}(G_1, G_2) \leq S_{\pi'}(G_1, G_2)\},$$
$$\mathcal{M} \triangleq \{\pi' \in \mathcal{S}_n : \pi' \neq \pi^*, (G_1, G_2) \in \mathcal{E}(\pi^*, \pi')\}.$$

Since the true permutation $\pi^*$ is uniformly distributed, the MLE $\hat{\pi}_{\mathrm{ML}}$ minimizes the error probability among all estimators. Therefore, to prove the impossibility result, it suffices to prove the failure of MLE. We note that $\hat{\pi}$ in equation 2 achieves exact recovery is equivalent to $\mathcal{M} = \emptyset$. To prove the impossibility of exact recovery, it suffices to show $\mathbb{P}[|\mathcal{M}| = 0] = o(1)$.

Let $I = |\mathcal{M} \cap \mathcal{T}_2|$ with $\mathcal{T}_2 = \{\pi' \in \mathcal{S}_n : \mathsf{d}(\pi^*, \pi') = 2\}$. Then $I \leq |\mathcal{M}|$. By Chebyshev's inequality, we have

$$\mathbb{P}[|\mathcal{M}| = 0] \leq \mathbb{P}[I = 0] \leq \mathbb{P}\left[(I - \mathbb{E}[I])^2 \geq (\mathbb{E}[I])^2\right] \leq \frac{\mathrm{Var}[I]}{(\mathbb{E}[I])^2}. \tag{20}$$

Given $\pi'$, let $\epsilon_1 \triangleq \mathbb{P}[(G_1, G_2) \in \mathcal{E}(\pi^*, \pi')]$. Since $|\mathcal{T}_2| = \binom{n}{2}$, the expectation $\mathbb{E}[I]$ is then given by

$$\mathbb{E}[I] = \sum_{\pi' \in \mathcal{T}_2} \mathbb{P}[(G_1, G_2) \in \mathcal{E}(\pi^*, \pi')] = \binom{n}{2} \epsilon_1.$$

We then compute the second moment $\mathbb{E}[I^2]$. Note that

$$I^2 = \left(\sum_{\pi' \in \mathcal{T}_2} \mathbf{1}_{\{(G_1, G_2) \in \mathcal{E}(\pi^*, \pi')\}}\right)^2$$
$$= \sum_{\pi' \in \mathcal{T}_2} \mathbf{1}_{\{(G_1, G_2) \in \mathcal{E}(\pi^*, \pi')\}} + \sum_{\pi_1, \pi_2 \in \mathcal{T}_2 : \pi_1 \neq \pi_2} \mathbf{1}_{\{(G_1, G_2) \in \mathcal{E}(\pi^*, \pi_1)\}} \mathbf{1}_{\{(G_1, G_2) \in \mathcal{E}(\pi^*, \pi_2)\}}. \tag{21}$$

It remains to compute $\sum_{\pi_1 \neq \pi_2 \in \mathcal{T}_2} \mathbb{P}[(G_1, G_2) \in \mathcal{E}(\pi^*, \pi_1) \cap \mathcal{E}(\pi^*, \pi_2)]$. Indeed, we note that $\mathsf{d}(\pi_1, \pi_2) \in \{3, 4\}$ for $\pi_1 \neq \pi_2 \in \mathcal{T}_2$. The number of pairs $(\pi_1, \pi_2)$ with $\pi_1 \neq \pi_2 \in \mathcal{T}_2$ with $\mathsf{d}(\pi_1, \pi_2) = 3$ and $\mathsf{d}(\pi_1, \pi_2) = 4$ are $6\binom{n}{3}$ and $6\binom{n}{4}$, respectively. For $\pi_1 \neq \pi_2 \in \mathcal{T}_2$ with $\mathsf{d}(\pi_1, \pi_2) = 4$, since $S_{\pi*}(G_1, G_2) - S_{\pi_1}(G_1, G_2)$ and $S_{\pi*}(G_1, G_2) - S_{\pi_2}(G_1, G_2)$ are independent, we have

$$\mathbb{P}[(G_1, G_2) \in \mathcal{E}(\pi^*, \pi_1) \cap \mathcal{E}(\pi^*, \pi_2)] = \mathbb{P}[(G_1, G_2) \in \mathcal{E}(\pi^*, \pi_1)] \mathbb{P}[(G_1, G_2) \in \mathcal{E}(\pi^*, \pi_2)] = \epsilon_1^2.$$

For $\pi_1 \neq \pi_2 \in \mathcal{T}_2$ with $\mathsf{d}(\pi_1, \pi_2) = 3$, we have the following Lemma.

**Lemma 5.** *For any $\pi_1 \neq \pi_2 \in \mathcal{T}_2$ with $\mathsf{d}(\pi_1, \pi_2) = 3$, we have*

$$\mathbb{P}[(G_1, G_2) \in \mathcal{E}(\pi^*, \pi_1) \cap \mathcal{E}(\pi^*, \pi_2)] \leq (1 - \rho^2)^{\frac{3(n-2)}{4}} (1 - r^2)^{\frac{3d}{4}}.$$

Next we prove a lower bound of $\epsilon_1$. For any $\pi' \in \mathcal{T}_2$, we assume that $\pi^*(v) = \pi'(v)$ for any $v \in V(G_1) \setminus \{v_1, v_2\}$, $\pi^*(v_1) = \pi'(v_2)$, and $\pi^*(v_2) = \pi'(v_1)$. Consequently,

$$
\epsilon_1 = \mathbb{P}\left[(G_1, G_2) \in \mathcal{E}(\pi^*, \pi')\right]
$$

$$
= \mathbb{P}\left[\varphi(\rho) \sum_{e \in E(G_1)} \beta_e(G_1) \left(\beta_{\pi'(e)}(G_2) - \beta_{\pi^*(e)}(G_2)\right) \right.
$$

$$
\left. + \varphi(r) \sum_{v \in V(G_1)} \boldsymbol{x}_v^\top \left(\boldsymbol{y}_{\pi'(v)} - \boldsymbol{y}_{\pi^*(v)}\right) \geq 0\right]
$$

$$
= \mathbb{P}\left[\varphi(\rho) \sum_{v \in V(G_1) \setminus \{v_1, v_2\}} (\beta_{vv_1}(G_1) - \beta_{vv_2}(G_1))(\beta_{\pi^*(vv_1)}(G_2) - \beta_{\pi^*(vv_2)}(G_2)) \right.
$$

$$
\left. + \varphi(r)(\boldsymbol{x}_{v_1} - \boldsymbol{x}_{v_2})^\top (\boldsymbol{y}_{\pi^*(v_1)} - \boldsymbol{y}_{\pi^*(v_2)}) \leq 0\right]
$$

$$
\geq \mathbb{P}\left[\sum_{v \in V(G_1) \setminus \{v_1, v_2\}} (\beta_{vv_1}(G_1) - \beta_{vv_2}(G_1))(\beta_{\pi^*(vv_1)}(G_2) - \beta_{\pi^*(vv_2)}(G_2)) \leq 0\right]
$$

$$
\cdot \mathbb{P}\left[(\boldsymbol{x}_{v_1} - \boldsymbol{x}_{v_2})^\top (\boldsymbol{y}_{\pi^*(v_1)} - \boldsymbol{y}_{\pi^*(v_2)}) \leq 0\right].
$$

We then bound the probability of two events separately. We note that

$$
\begin{bmatrix} X_v \\ Y_v \end{bmatrix} \triangleq \begin{bmatrix} \beta_{vv_1}(G_1) - \beta_{vv_2}(G_1) \\ \beta_{\pi^*(vv_1)}(G_2) - \beta_{\pi^*(vv_2)}(G_2) \end{bmatrix} \sim \mathcal{N}\left(\begin{bmatrix} 0 \\ 0 \end{bmatrix}, 2 \begin{bmatrix} 1 & \rho \\ \rho & 1 \end{bmatrix}\right).
$$

Let $\xi_v \overset{\text{i.i.d.}}{\sim} \mathcal{N}(0, 1)$ for any $v \in V(G_1) \setminus \{v_1, v_2\}$. We note that

$$
\mathbb{P}\left[\sum_{v \in V(G_1) \setminus \{v_1, v_2\}} X_v Y_v \geq 0\right]
$$

$$
= \mathbb{E}\left[\mathbb{P}\left[\sum_{v \in V(G_1) \setminus \{v_1, v_2\}} X_v Y_v \geq 0 \,\middle|\, \{v \in V(G_1) \setminus \{v_1, v_2\}\}\right]\right]
$$

$$
\overset{(a)}{=} \mathbb{E}\left[\mathbb{P}\left[\sum_{v \in V(G_1) \setminus \{v_1, v_2\}} \rho X_v^2 + 2\sqrt{1 - \rho^2} X_v \xi_v \leq 0 \,\middle|\, \{v \in V(G_1) \setminus \{v_1, v_2\}\}\right]\right]
$$

$$
\overset{(b)}{=} \mathbb{E}\left[\mathbb{P}\left[\mathcal{N}(0, 1) \geq \frac{\rho \sqrt{\sum_{v \in V(G_1) \setminus \{v_1, v_2\}} X_v^2}}{\sqrt{2(1 - \rho^2)}} \,\middle|\, \{v \in V(G_1) \setminus \{v_1, v_2\}\}\right]\right],
$$

where (a) is because $Y_v | X_v \sim \mathcal{N}(\rho X_v, 2(1 - \rho^2))$ and $\{Y_v | X_v, v \in V(G_1) \setminus \{v_1, v_2\}\}$ are independent; (b) is because

$$
\sum_{v \in V(G_1) \setminus \{v_1, v_2\}} X_v \xi_v \,\middle|\, \{X_v : v \in V(G_1) \setminus \{v_1, v_2\}\} \sim \mathcal{N}\left(0, 2(1 - \rho^2) \sum_{v \in V(G_1) \setminus \{v_1, v_2\}} X_v^2\right).
$$

By Lemma 7, since $\sum_{v \in V(G_1) \setminus \{v_1, v_2\}} \frac{1}{2} X_v^2 \sim \chi^2(n - 2)$, we have

$$
\mathbb{P}\left[\sum_{v \in V(G_1) \setminus \{v_1, v_2\}} X_v^2 \leq 2(n + \sqrt{n \log n})\right] = 1 - o(1).
$$

Consequently,

$$\mathbb{P}\left[\sum_{v \in V(G_1)\setminus\{v_1,v_2\}} X_v Y_v \geq 0\right]$$

$$\geq \mathbb{E}\left[(1-o(1))\mathbb{P}\left[\mathcal{N}(0,1) \geq \frac{\rho\sqrt{2(n+\sqrt{n\log n})}}{\sqrt{2(1-\rho^2)}}\right]\right]$$

$$\overset{(a)}{\geq} \frac{1-o(1)}{\sqrt{2\pi}}\exp\left(-\frac{1}{2}\frac{\rho^2(n+\sqrt{n\log n})}{1-\rho^2}\right)\frac{2}{\frac{\rho\sqrt{n+\sqrt{n\log n}}}{\sqrt{1-\rho^2}}+\sqrt{4+\frac{\rho^2(n+\sqrt{n\log n})}{1-\rho^2}}}$$

$$\overset{(b)}{\geq} \frac{1}{16\sqrt{\log n}}(1-\rho^2)^{-\frac{1}{2}(n-2)(1+o(1))},$$

where (a) is because $\mathbb{P}[Z \geq t] \geq \frac{2}{t+\sqrt{t^2+4}}\frac{1}{\sqrt{2\pi}}\exp\left(-\frac{1}{2}t^2\right)$ for $Z \sim \mathcal{N}(0,1)$ (Birnbaum, 1942); (b) is because $n\log\left(\frac{1}{1-\rho^2}\right) \leq (4-\epsilon)\log n$ implies $\frac{\rho^2}{1-\rho^2}(n+\sqrt{n\log n}) \leq 4\log n$, $\frac{1-o(1)}{\sqrt{2\pi}}\cdot\frac{2}{\sqrt{4\log n}+\sqrt{4+4\log n}} \geq \frac{1}{16\sqrt{\log n}}$, and $\frac{\rho^2}{1-\rho^2} = (1+o(1))\log\left(\frac{1}{1-\rho^2}\right)$. It follows from (Kunisky & Niles-Weed, 2022, Proposition 4.3) that when $r^2 \geq \frac{40}{d}$,

$$\mathbb{P}\left[(\boldsymbol{x}_{v_1} - \boldsymbol{x}_{v_2})^\top(\boldsymbol{y}_{\pi^*(v_1)} - \boldsymbol{y}_{\pi^*(v_2)}) \leq 0\right] \geq \frac{1}{1000\sqrt{d}}(1-r^2)^{\frac{d}{2}}.$$

Consequently,

$$\epsilon_1 \geq \frac{1}{16000\sqrt{d\log n}}(1-\rho^2)^{\frac{n-2}{2}(1+o(1))}(1-r^2)^{\frac{d}{2}}.$$

By Lemma 5, for any $\pi_1 \neq \pi_2 \in \mathcal{T}_2$ with $\mathsf{d}(\pi_1, \pi_2) = 3$, we have

$$\epsilon_2 \triangleq \mathbb{P}\left[(G_1, G_2) \in \mathcal{E}(\pi^*, \pi_1) \cap \mathcal{E}(\pi^*, \pi_2)\right] \leq (1-\rho^2)^{\frac{3(n-2)}{4}}(1-r^2)^{\frac{3d}{4}}.$$

By equation 20 and equation 21,

$$\mathbb{P}[|\mathcal{M}| = 0] \leq \frac{\mathbb{E}[I^2] - (\mathbb{E}[I])^2}{(\mathbb{E}[I])^2} = \frac{\binom{n}{2}\epsilon_1 + 6\binom{n}{3}\epsilon_2 + 6\binom{n}{4}\epsilon_1^2 - \binom{n}{2}^2\epsilon_1^2}{\binom{n}{2}^2\epsilon_1^2} \leq \frac{4}{n^2\epsilon_1} + \frac{4\epsilon_2}{n\epsilon_1^2}.$$

Since $n\log\left(\frac{1}{1-\rho^2}\right) + d\log\left(\frac{1}{1-r^2}\right) + 4\log d \leq (4-\epsilon)\log n$, we obtain

$$n^2\epsilon_1 \geq \frac{1}{16000\sqrt{\log n}}$$
$$\cdot \exp\left(2\log n - \frac{1}{2}\log d - \frac{n-2}{2}(1+o(1))\log\left(\frac{1}{1-\rho^2}\right) - \frac{d}{2}\log\left(\frac{1}{1-r^2}\right)\right)$$
$$\geq \frac{1}{16000\sqrt{\log n}}\exp\left(\frac{\epsilon}{4}\log n\right)$$

and

$$\frac{n\epsilon_1^2}{\epsilon_2} \geq \frac{1}{256\cdot 10^6\log n}$$
$$\cdot \exp\left(\log n - \log d - \frac{n-2}{4}(1+o(1))\log\left(\frac{1}{1-\rho^2}\right) - \frac{d}{4}\log\left(\frac{1}{1-r^2}\right)\right)$$
$$\geq \frac{1}{256\cdot 10^6\log n}\exp\left(\frac{\epsilon}{8}\log n\right).$$

Therefore, we obtain $\mathbb{P}[|\mathcal{M}| = 0] \leq \frac{4}{n^2\epsilon_1} + \frac{4\epsilon_2}{n\epsilon_1^2} = o(1)$, we finish the proof.

# D    PROOF OF LEMMAS

## D.1    PROOF OF LEMMA 1

We first lower bound the packing number $|\mathcal{M}_\delta|$. For any $0 < \delta < 1$ and $\pi \in \mathcal{S}_n$, let $B(\pi, r) \triangleq \{\pi' : \mathsf{d}(\pi, \pi') \leq r\}$ denote the ball of radius $r$ centered at $\pi$. By a standard volume argument (Polyanskiy & Wu, 2025, Theorem 27.3), we have

$$|\mathcal{M}_\delta| \geq \frac{|\mathcal{S}_n|}{\max_\pi |B(\pi, (1 - \delta)n)|} = \frac{n!}{\max_\pi |B(\pi, (1 - \delta)n)|}.$$

To upper bound $|B(\pi, (1 - \delta)n)|$, we first choose $\delta n$ elements from the domain of $\pi$ and map to the same value as $\pi$, and the remaining domain and range of size $n - \delta n$ and the mapping are selected arbitrarily. We get $B(\pi, (1 - \delta)n) \leq \binom{n}{\delta n}(n - \delta n)!$. Consequently,

$$|\mathcal{M}_\delta| \geq \frac{n!}{\max_\pi |B(\pi, (1 - \delta)n)|} \geq \frac{n!}{\binom{n}{\delta n}(n - \delta n)!} = (\delta n)! \geq \left(\frac{\delta n}{e}\right)^{\delta n}.$$

We then upper bound the mutual information. Recall that in equation 1 the likelihood function is given by

$$\mathcal{P}_{G_1,G_2|\pi^*} = \prod_{e \in E(G_1)} P(\beta_e(G_1), \beta_{\pi^*(e)}(G_2)) \prod_{v \in V(G_1)} f(\boldsymbol{x}_v, \boldsymbol{y}_{\pi^*(v)}).$$

Next, we introduce an auxiliary distribution $\mathcal{Q}$ under which $G_1$ and $G_2$ are independent, while maintaining the same marginals as under $\mathcal{P}$. Denote $Q(\cdot, \cdot)$ as the distribution of two independent standard normals and $g(\boldsymbol{x}, \boldsymbol{y})$ as the multivariate normal distribution $\mathcal{N}\left(\boldsymbol{0}, \begin{bmatrix} I_d & O \\ O & I_d \end{bmatrix}\right)$. Then

$$\mathcal{Q}_{G_1,G_2} = \prod_{e \in E(G_1)} Q(\beta_e(G_1), \beta_{\pi^*(e)}(G_2)) \prod_{v \in V(G_1)} g(\boldsymbol{x}_v, \boldsymbol{y}_{\pi(v)}).$$

The KL-divergence between the product measures $\mathcal{P}_{G_1,G_2|\pi^*}$ and $\mathcal{Q}_{G_1,G_2}$ is given by

$$D_{\mathrm{KL}}(\mathcal{P}_{G_1,G_2|\pi^*}\|\mathcal{Q}_{G_1,G_2}) = \binom{n}{2}D_{\mathrm{KL}}(P\|Q) + nD_{\mathrm{KL}}(f\|g).$$

We note that

$$\begin{aligned}
D_{\mathrm{KL}}(P\|Q) &= \iint P(a,b) \log\left(\frac{P(a,b)}{Q(a,b)}\right) \mathrm{d}a \mathrm{d}b \\
&= \iint P(a,b) \left[\frac{1}{2}\log\left(\frac{1}{1-\rho^2}\right) + \frac{\rho a b}{1-\rho^2} - \frac{\rho^2(a^2+b^2)}{2(1-\rho^2)}\right] \mathrm{d}a \mathrm{d}b \\
&= \frac{1}{2}\log\left(\frac{1}{1-\rho^2}\right) + \frac{\rho^2}{1-\rho^2} - \frac{2\rho^2}{2(1-\rho^2)} = \frac{1}{2}\log\left(\frac{1}{1-\rho^2}\right).
\end{aligned}$$

Similarly, $D_{\mathrm{KL}}(f\|g) = \frac{d}{2}\log\left(\frac{1}{1-r^2}\right)$. Consequently,

$$\begin{aligned}
I(\pi^*; G_1, G_2) &= \mathbb{E}_{\pi^*}\left[D_{\mathrm{KL}}(\mathcal{P}_{G_1,G_2|\pi^*}\|\mathcal{P}_{G_1,G_2})\right] \\
&\leq \max_{\pi \in \mathcal{S}_n} D_{\mathrm{KL}}(\mathcal{P}_{G_1,G_2|\pi}\|\mathcal{Q}_{G_1,G_2}) = \binom{n}{2}\frac{1}{2}\log(\frac{1}{1-\rho^2}) + \frac{nd}{2}\log(\frac{1}{1-r^2}).
\end{aligned}$$

## D.2    PROOF OF LEMMA 2

Note that $W = X^\top A Y = \frac{1}{4}(X+Y)^\top A(X+Y) - \frac{1}{4}(X-Y)^\top A(X-Y)$ and

$$\begin{aligned}
\mathbb{E}\left[(X+Y)^\top A(X+Y)\right] &= (2 + 2\rho)\varphi(\rho)N_k + (2 + 2r)\varphi(r)dk \\
\mathbb{E}\left[(X-Y)^\top A(X-Y)\right] &= (2 - 2\rho)\varphi(\rho)N_k + (2 - 2r)\varphi(r)dk.
\end{aligned}$$

By Hanson-Wright inequality (Hanson & Wright, 1971), there exists some universal constant $C$ such that

$$\mathbb{P}\left[\left|\frac{1}{4}(X+Y)^\top A(X+Y) - \mathbb{E}\left[\frac{1}{4}(X+Y)^\top A(X+Y)\right]\right|\right.$$

$$\left. \geq \frac{C}{2}\left(\|A\|_F \sqrt{\log\left(\frac{1}{\delta_0}\right)} \vee \|A\|_2 \log\left(\frac{1}{\delta_0}\right)\right)\right] \leq \frac{\delta_0}{2},$$

$$\mathbb{P}\left[\left|\frac{1}{4}(X-Y)^\top A(X-Y) - \mathbb{E}\left[\frac{1}{4}(X-Y)^\top A(X-Y)\right]\right|\right.$$

$$\left. \geq \frac{C}{2}\left(\|A\|_F \sqrt{\log\left(\frac{1}{\delta_0}\right)} \vee \|A\|_2 \log\left(\frac{1}{\delta_0}\right)\right)\right] \leq \frac{\delta_0}{2}$$

for any $\delta_0 > 0$. Consequently,

$$\mathbb{P}\left[\left|X^\top AY - \rho\varphi(\rho)N_k - r\varphi(r)dk\right| \geq C\left(\|A\|_F \sqrt{\log\left(\frac{1}{\delta_0}\right)} \vee \|A\|_2 \log\left(\frac{1}{\delta_0}\right)\right)\right]$$

$$\leq \mathbb{P}\left[\left|\frac{1}{4}(X+Y)^\top A(X+Y) - \mathbb{E}\left[\frac{1}{4}(X+Y)^\top A(X+Y)\right]\right|\right.$$

$$\left. \geq \frac{C}{2}\left(\|A\|_F \sqrt{\log\left(\frac{1}{\delta_0}\right)} \vee \|A\|_2 \log\left(\frac{1}{\delta_0}\right)\right)\right]$$

$$+ \mathbb{P}\left[\left|\frac{1}{4}(X-Y)^\top A(X-Y) - \mathbb{E}\left[\frac{1}{4}(X-Y)^\top A(X-Y)\right]\right|\right.$$

$$\left. \geq \frac{C}{2}\left(\|A\|_F \sqrt{\log\left(\frac{1}{\delta_0}\right)} \vee \|A\|_2 \log\left(\frac{1}{\delta_0}\right)\right)\right] \leq \delta_0.$$

We finish the proof of Lemma 2.

### D.3    PROOF OF LEMMA 3

By equation 17, the cumulant generating function is given by

$$\log \mathbb{E}\left[\exp\left(tZ\right)\right] = \log \mathbb{E}\left[\exp\left(tZ^{\mathsf{V}}\right)\right] + \log \mathbb{E}\left[\exp\left(tZ^{\mathsf{E}}\right)\right].$$

We first calculate $\mathbb{E}\left[\exp\left(tZ^{\mathsf{E}}\right)\right]$. Define the moment generating function (MGF) as $m_k^{\mathsf{E}} = \exp\left(\kappa_k^{\mathsf{E}}\right)$ for any $k \geq 1$. For any edge cycle $C = \{e_1, e_2, \cdots, e_k\}$ with $e_{i+1} = \sigma^{\mathsf{E}}(e_i)$ for all $1 \leq i \leq k-1$ and $e_1 = \sigma^{\mathsf{E}}(e_k)$, let $A_{i-1} = \beta_{e_i}(G_1)$ and $B_i = \beta_{\pi'(e_i)}(G_2)$ for any $1 \leq i \leq k$, and we set $A_k = A_0$ for notational simplicity. Since $\pi^*(e_{i+1}) = \pi'(e_i)$, each pair $(A_i, B_i)$ follows bivariate normal distribution $\mathcal{N}\left(0, \begin{bmatrix} 1 & \rho \\ \rho & 1 \end{bmatrix}\right)$, and thus the conditional distribution is given by $A_i | B_i \sim \mathcal{N}(\rho B_i, 1 - \rho^2)$. Consequently, the MGF is given by

$$m_k^{\mathsf{E}} = \mathbb{E}\left[\mathbb{E}\left[\prod_{i=1}^k \exp\left(t\varphi(\rho)A_{i-1}B_i\right) \Big| B_1, \cdots B_k\right]\right]$$

$$= \mathbb{E}\left[\prod_{i=1}^k \exp\left(t\rho\varphi(\rho)B_{i-1}B_i + \frac{1}{2}t^2\varphi(\rho)^2 B_i^2(1-\rho^2)\right)\right],$$

where the last equality is because $\mathbb{E}\left[\exp\left(tX\right)\right] = \exp\left(t\mu + \frac{1}{2}t^2\sigma^2\right)$ for $X \sim \mathcal{N}(\mu, \sigma^2)$.

Let $\lambda_1, \lambda_2$ denote the roots of the quadratic function $x^2 - \left[1 - t^2\varphi(\rho)^2(1-\rho^2)\right]x + t^2\varphi(\rho)^2\rho^2 = 0$. Since $t \leq \frac{1}{\rho} - 2$, we have $\lambda_1 + \lambda_2 > 0$ and the discriminant $\left[1 - t^2\varphi(\rho)^2(1-\rho^2)\right]^2 - 4t^2\varphi(\rho)^2\rho^2 >$

0. Since $\lambda_1\lambda_2 > 0$, we have $\lambda_1 > \lambda_2 > 0$. Define the matrix

$$
\mathbf{J}_k \triangleq
\begin{bmatrix}
\lambda_1^{1/2} & -\lambda_2^{1/2} & 0 & \cdots & 0 \\
0 & \lambda_1^{1/2} & -\lambda_2^{1/2} & \cdots & 0 \\
0 & 0 & \lambda_1^{1/2} & \cdots & 0 \\
\vdots & \vdots & \vdots & \ddots & -\lambda_2^{1/2} \\
-\lambda_2^{1/2} & 0 & \cdots 0 & 0 & \lambda_1^{1/2}
\end{bmatrix}
\in \mathbb{R}^{k\times k}.
$$

Denote $\mathbf{B}_k = [B_1, B_2, \cdots, B_k]^\top$. Then we have

$$
m_k^{\mathsf{E}} = \mathbb{E}\left[\prod_{i=1}^k \exp\left(t\rho\varphi(\rho)B_{i-1}B_i + \frac{1}{2}t^2\varphi(\rho)^2 B_i^2(1-\rho^2)\right)\right]
$$

$$
= \int\cdots\int \left(\frac{1}{\sqrt{2\pi}}\right)^k \exp\left(-\frac{1}{2}\sum_{i=1}^k \left(B_i^2 - \left(2t\rho\varphi(\rho)B_{i-1}B_i + t^2\varphi(\rho)^2(1-\rho^2)B_i^2\right)\right)\right)
$$

$$
\mathrm{d}B_1\cdots\mathrm{d}B_k
$$

$$
= \int\cdots\int \left(\frac{1}{\sqrt{2\pi}}\right)^k \exp\left(-\frac{1}{2}\sum_{i=1}^k \left(\lambda_1^{1/2}B_{i-1} - \lambda_2^{1/2}B_i\right)^2\right)\mathrm{d}B_1\cdots\mathrm{d}B_k
$$

$$
= \int\cdots\int \left(\frac{1}{\sqrt{2\pi}}\right)^k \exp\left(-\frac{1}{2}\mathbf{B}_k^\top \mathbf{J}_k^\top \mathbf{J}_k\mathbf{B}_k\right)\mathrm{d}B_1\cdots\mathrm{d}B_k = [\det(\mathbf{J}_k)]^{-1} = \frac{1}{\lambda_1^{k/2} - \lambda_2^{k/2}}.
$$

We then calculate $\mathbb{E}\left[\exp\left(tZ^{\mathsf{V}}\right)\right]$. Define the moment generating function (MGF) as $m_k^{\mathsf{V}} = \exp\left(\kappa_k^{\mathsf{V}}\right)$ for any $k \geq 1$. For any vertex cycle $C = \{v_1, \cdots, v_k\}$ with $v_{i+1} = \sigma(v_i)$ for any $1 \leq i \leq k-1$ and $v_1 = \sigma(v_k)$, let $\tilde{A}_{i-1} = \boldsymbol{x}_{v_i}$ and $\tilde{B}_i = \boldsymbol{y}_{\pi'(v_i)}$, and we set $\tilde{A}_k = \tilde{A}_0$ for notational simplicity. Since $\pi^*(v_{i+1}) = \pi'(v_i)$, each pair $(\tilde{A}_i, \tilde{B}_i) \sim \mathcal{N}(\mathbf{0}, \begin{bmatrix} I_d & rI_d \\ rI_d & I_d \end{bmatrix})$. Similarly,

$$
m_k^{\mathsf{V}} = \mathbb{E}\left[\mathbb{E}\left[\prod_{i=1}^k \exp\left(t\varphi(r)\tilde{A}_{i-1}^\top\tilde{B}_i\right)\Big|\tilde{B}_1, \cdots\tilde{B}_k\right]\right]
$$

$$
= \mathbb{E}\left[\prod_{i=1}^k \exp\left(tr\varphi(r)\tilde{B}_{i-1}^\top\tilde{B}_i + \frac{1}{2}t^2\varphi(r)^2\tilde{B}_i^\top\tilde{B}_i(1-r^2)\right)\right]
$$

$$
= \prod_{j=1}^d \mathbb{E}\left[\prod_{i=1}^k \exp\left(tr\varphi(r)\tilde{B}_{i-1,j}^\top\tilde{B}_{i,j} + \frac{1}{2}t^2\varphi(r)^2\tilde{B}_{i,j}^\top\tilde{B}_{i,j}(1-r^2)\right)\right],
$$

where $\tilde{B}_{i,j}$ denotes the $j$−th element of vector $\tilde{B}_i$ and the last equality is because $\tilde{B}_{i,j}$ and $\tilde{B}_{i',j'}$ are independent for any $(i,j) \neq (i',j')$. Let $\mu_1 > \mu_2$ denote two roots of the quadratic equation $x^2 - \left[1 - t^2\varphi(r)^2(1-r^2)\right]x + t^2\varphi(r)^2r^2 = 0$. Since $t \leq \frac{1}{r} - 2$, we have $\mu_1 + \mu_2 > 0$ and the discriminant $\left[1 - t^2\varphi(r)^2(1-r^2)\right]^2 - 4t^2\varphi(r)^2r^2 > 0$. Since $\mu_1\mu_2 > 0$, we have $\mu_1 > \mu_2 > 0$. By a similar argument with calculation for the edge cycle, we have

$$
m_k^{\mathsf{V}} = \left(\frac{1}{\mu_1^{k/2} - \mu_2^{k/2}}\right)^d.
$$

For any $k \geq 2$, we have $\lambda_1^{k/2} - \lambda_2^{k/2} \geq (\lambda_1 - \lambda_2)^{k/2}$, and thus $m_k^{\mathsf{E}} \leq (m_2^{\mathsf{E}})^{k/2}$ for any $k \geq 2$. Similarly, $m_k^{\mathsf{V}} \leq (m_2^{\mathsf{V}})^{k/2}$ for any $k \geq 2$. Recall $Z^{\mathsf{E}}$ and $Z^{\mathsf{V}}$ defined in equation 16. We have

$$\log \mathbb{E}\left[\exp(tZ)\right] = \log \mathbb{E}\left[\exp(tZ^{\mathsf{E}})\right] + \log \mathbb{E}\left[\exp(tZ^{\mathsf{V}})\right]$$

$$= \sum_{C \in \mathcal{C}^{\mathsf{E}} \setminus \binom{F}{2}} \kappa_{|C|}^{\mathsf{E}} + \sum_{C \in \mathcal{C}^{\mathsf{V}} \setminus F} \kappa_{|C|}^{\mathsf{V}}$$

$$\overset{(a)}{\leq} \sum_{i \geq 2} \sum_{C \in \mathcal{C}_i^{\mathsf{E}}} \frac{|C|}{2} \kappa_2^{\mathsf{E}}(t) + \sum_{C \in \mathcal{C}_1^{\mathsf{E}} \setminus \binom{F}{2}} \kappa_1^{\mathsf{E}}(t) + \sum_{i \geq 2} \sum_{C \in \mathcal{C}_i^{\mathsf{V}}} \frac{|C|}{2} \kappa_2^{\mathsf{V}}(t)$$

$$= \sum_{C \in \mathcal{C}^{\mathsf{E}} \setminus \binom{F}{2}} \frac{|C|}{2} \kappa_2^{\mathsf{E}}(t) + \sum_{C \in \mathcal{C}_1^{\mathsf{E}} \setminus \binom{F}{2}} \left(\kappa_1^{\mathsf{E}}(t) - \frac{1}{2}\kappa_2^{\mathsf{E}}(t)\right) + \sum_{C \in \mathcal{C}^{\mathsf{V}} \setminus F} \frac{|C|}{2} \kappa_2^{\mathsf{V}}(t)$$

$$\overset{(b)}{\leq} \frac{N_k}{2} \kappa_2^{\mathsf{E}}(t) + \frac{k}{2}\left(\kappa_1^{\mathsf{E}} - \frac{1}{2}\kappa_2^{\mathsf{E}}(t) + \frac{1}{2}\kappa_2^{\mathsf{V}}(t)\right),$$

where (a) is because $\kappa_k^{\mathsf{E}}(t) \leq \frac{k}{2}\kappa_2^{\mathsf{E}}(t)$ and $\kappa_k^{\mathsf{V}}(t) \leq \frac{k}{2}\kappa_2^{\mathsf{V}}(t)$ for any $k \geq 2$; (b) is because $\sum_{C \in \mathcal{C}^{\mathsf{E}} \setminus \binom{F}{2}} |C| = \binom{n}{2} - \binom{n-k}{2} = N_k$, $\sum_{C \in \mathcal{C}^{\mathsf{V}} \setminus F} |C| = n - (n-k) = k$. It remains to show $|\mathcal{C}_1^{\mathsf{E}} \setminus \binom{F}{2}| \leq \frac{k}{2}$. Indeed, for any $e = uv \in C \in \mathcal{C}_1^{\mathsf{E}} \setminus \binom{F}{2}$, we have $\pi'(uv) = \pi^*(uv)$. Since $e \notin \binom{F}{2}$, we have $\pi'(u) = \pi^*(v)$ and $\pi'(v) = \pi^*(u)$, which contribute two mismatched vertices in the reconstruction of the underlying mapping. Since the total number of mismatched vertices for $\pi \in \mathcal{T}_k$ equals $k$, we have $|\mathcal{C}_1^{\mathsf{E}} \setminus \binom{F}{2}| \leq \frac{k}{2}$. Therefore, we finish the proof of Lemma 3.

### D.4 PROOF OF LEMMA 4

The cumulant generating function is given by

$$\log \mathbb{E}\left[\exp\left(tY\right)\right] = \log \mathbb{E}\left[\exp\left(tY^{\mathsf{V}}\right)\right] + \log \mathbb{E}\left[\exp\left(tY^{\mathsf{E}}\right)\right].$$

We first calculate $\mathbb{E}\left[\exp\left(tY^{\mathsf{E}}\right)\right]$. Define the moment generating function (MGF) as $\tilde{m}_k^{\mathsf{E}} = \exp\left(\mu_k^{\mathsf{E}}\right)$ for any $k \geq 1$. For any edge cycle $C = \{e_1, e_2, \cdots, e_k\}$ with $e_{i+1} = \sigma^{\mathsf{E}}(e_i)$ for all $1 \leq i \leq k-1$ and $e_1 = \sigma^{\mathsf{E}}(e_k)$, let $A_{i-1} = \beta_{e_i}(G_1)$ and $B_i = \beta_{\pi'(e_i)}(G_2)$ for any $1 \leq i \leq k$, and we set $A_k = A_0$ for notational simplicity. Since $\pi^*(e_{i+1}) = \pi'(e_i)$, each pair $(A_i, B_i)$ follows bivariate normal distribution $\mathcal{N}\left(0, \begin{bmatrix} 1 & \rho \\ \rho & 1 \end{bmatrix}\right)$, and thus the conditional distribution is given by $A_i | B_i \sim \mathcal{N}(\rho B_i, 1 - \rho^2)$. Consequently, the MGF is given by

$$\tilde{m}_k^{\mathsf{E}} = \mathbb{E}\left[\mathbb{E}\left[\prod_{i=1}^{k} \exp\left(t\varphi(\rho)(A_{i-1}B_i - A_{i-1}B_{i-1})\right) \big| B_1, \cdots B_k\right]\right]$$

$$= \mathbb{E}\left[\prod_{i=1}^{k} \exp\left(t\rho\varphi(\rho)B_{i-1}(B_i - B_{i-1}) + \frac{1}{2}t^2\varphi(\rho)^2(B_i - B_{i-1})^2(1 - \rho^2)\right)\right]$$

$$= \mathbb{E}\left[\prod_{i=1}^{k} \exp\left((t\rho\varphi(\rho) - t^2\varphi(\rho)^2(1 - \rho^2))(B_{i-1}B_i - B_i^2)\right)\right],$$

where the second equality is because $\mathbb{E}\left[\exp\left(tX\right)\right] = \exp\left(t\mu + \frac{1}{2}t^2\sigma^2\right)$ for $X \sim \mathcal{N}(\mu, \sigma^2)$.

Let $\lambda_1, \lambda_2$ denote the roots of the quadratic function

$$x^2 - \left[1 - 2(t^2\varphi(\rho)^2(1 - \rho^2) - t\rho\varphi(\rho))\right]x + (t^2\varphi(\rho)^2(1 - \rho^2) - t\rho\varphi(\rho))^2 = 0.$$

Since $0 < t < 1$, we have

$$f(t, \rho) \triangleq t^2\varphi(\rho)^2(1 - \rho^2) - t\rho\varphi(\rho) = (t^2 - t)\frac{\rho^2}{1 - \rho^2} \in \left(-\frac{1}{4}, 0\right).$$

Therefore, we have $\lambda_1 + \lambda_2 = 1 - 2f(t,\rho) > 0$, $\lambda_1\lambda_2 = f(t,\rho)^2 > 0$ and the discriminant $(1 - 2f(t,\rho))^2 - 4f(t,\rho)^2 = 1 - 4f(t,\rho) > 0$, and thus $\lambda_1 > \lambda_2 > 0$. Define the matrix

$$
\mathbf{J}_k \triangleq \begin{bmatrix}
\lambda_1^{1/2} & -\lambda_2^{1/2} & 0 & \cdots & 0 \\
0 & \lambda_1^{1/2} & -\lambda_2^{1/2} & \cdots & 0 \\
0 & 0 & \lambda_1^{1/2} & \cdots & 0 \\
\vdots & \vdots & \vdots & \ddots & -\lambda_2^{1/2} \\
-\lambda_2^{1/2} & 0 & \cdots 0 & 0 & \lambda_1^{1/2}
\end{bmatrix} \in \mathbb{R}^{k \times k}.
$$

Denote $\mathbf{B}_k = [B_1, B_2, \cdots, B_k]^\top$. Then we have

$$
\tilde{m}_k^{\mathsf{E}} = \mathbb{E}\left[ \prod_{i=1}^{k} \exp\left( (t\rho\varphi(\rho) - t^2\varphi(\rho)^2(1-\rho^2))(B_{i-1}B_i - B_i^2) \right) \right]
$$

$$
= \int \cdots \int \left( \frac{1}{\sqrt{2\pi}} \right)^k \exp\left( -\frac{1}{2}\sum_{i=1}^{k} B_i^2 \right)
$$

$$
\exp\left( \sum_{i=1}^{k} \left[ (t\rho\varphi(\rho) - t^2\varphi(\rho)^2(1-\rho^2))(B_{i-1}B_i - B_i^2) \right] \right) dB_1 \cdots dB_k
$$

$$
= \int \cdots \int \left( \frac{1}{\sqrt{2\pi}} \right)^k \exp\left( -\frac{1}{2}\sum_{i=1}^{k} \left( \lambda_1^{1/2}B_{i-1} - \lambda_2^{1/2}B_i \right)^2 \right) dB_1 \cdots dB_k
$$

$$
= \int \cdots \int \left( \frac{1}{\sqrt{2\pi}} \right)^k \exp\left( -\frac{1}{2}\mathbf{B}_k^\top \mathbf{J}_k^\top \mathbf{J}_k \mathbf{B}_k \right) dB_1 \cdots dB_k
$$

$$
= [\det(\mathbf{J}_k)]^{-1} = \frac{1}{\lambda_1^{k/2} - \lambda_2^{k/2}}.
$$

We then calculate $\mathbb{E}\left[ \exp\left( tY^{\mathsf{V}} \right) \right]$. Define the moment generating function (MGF) as $\tilde{m}_k^{\mathsf{V}} = \exp\left( \mu_k^{\mathsf{V}} \right)$ for any $k \geq 1$. For any vertex cycle $C = \{v_1, \cdots, v_k\}$ with $v_{i+1} = \sigma(v_i)$ for any $1 \leq i \leq k-1$ and $v_1 = \sigma(v_k)$, let $\tilde{A}_{i-1} = \boldsymbol{x}_{v_i}$ and $\tilde{B}_i = \boldsymbol{y}_{\pi'(v_i)}$, and we set $\tilde{A}_k = \tilde{A}_0$ for notational simplicity. Since $\pi^*(v_{i+1}) = \pi'(v_i)$, each pair $(\tilde{A}_i, \tilde{B}_i) \sim \mathcal{N}(\mathbf{0}, \begin{bmatrix} I_d & rI_d \\ rI_d & I_d \end{bmatrix})$. Similarly,

$$
m_k^{\mathsf{V}} = \mathbb{E}\left[ \mathbb{E}\left[ \prod_{i=1}^{k} \exp\left( t\varphi(r)(\tilde{A}_{i-1}^\top \tilde{B}_i - \tilde{A}_{i-1}^\top \tilde{B}_{i-1}) \right) \Big| \tilde{B}_1, \cdots \tilde{B}_k \right] \right]
$$

$$
= \mathbb{E}\left[ \prod_{i=1}^{k} \exp\left( tr\varphi(r)\tilde{B}_{i-1}^\top(\tilde{B}_i - \tilde{B}_{i-1}) + \frac{1}{2}t^2\varphi(r)^2(\tilde{B}_i - \tilde{B}_{i-1})^\top(\tilde{B}_i - \tilde{B}_{i-1})(1-r^2) \right) \right]
$$

$$
= \prod_{j=1}^{d} \mathbb{E}\left[ \prod_{i=1}^{k} \exp\left( (tr\varphi(r) - t^2\varphi(r)^2(1-r^2))(B_{i-1,j}B_{i,j} - B_{i,j}^2) \right) \right],
$$

where $\tilde{B}_{i,j}$ denotes the $j-$th element of vector $\tilde{B}_i$ and the last equality is because $\tilde{B}_{i,j}$ and $\tilde{B}_{i',j'}$ are independent for any $(i,j) \neq (i',j')$. Let $\mu_1 > \mu_2$ denote two roots of the quadratic equation

$$
x^2 - \left[ 1 - 2(t^2\varphi(r)^2(1-r^2) - tr\varphi(r)) \right] x + (t^2\varphi(r)^2(1-r^2) - tr\varphi(r))^2 = 0.
$$

Since $0 < t < 1$, we have

$$
f(t,r) \triangleq t^2\varphi(r)^2(1-r^2) - tr\varphi(r) = (t^2 - t)\frac{r^2}{1-r^2} \in \left( -\frac{1}{4}, 0 \right).
$$

Therefore, we have $\lambda_1 + \lambda_2 = 1 - 2f(t,r) > 0$, $\lambda_1\lambda_2 = f(t,r)^2 > 0$ and the discriminant $(1 - 2f(t,r))^2 - 4f(t,r)^2 = 1 - 4f(t,r) > 0$, and thus $\mu_1 > \mu_2 > 0$. By a similar argument with

calculation for the edge cycle, we have

$$\tilde{m}_k^{\mathsf{V}} = \left(\frac{1}{\mu_1^{k/2} - \mu_2^{k/2}}\right)^d.$$

For any $k \geq 2$, we have $\lambda_1^{k/2} - \lambda_2^{k/2} \geq (\lambda_1 - \lambda_2)^{k/2}$, and thus $\tilde{m}_k^{\mathsf{E}} \leq (\tilde{m}_2^{\mathsf{E}})^{k/2}$ for any $k \geq 2$. Similarly, $\tilde{m}_k^{\mathsf{V}} \leq (\tilde{m}_2^{\mathsf{V}})^{k/2}$ for any $k \geq 2$.

Then, we upper bound $\log \mathbb{E}\left[\exp\left(tY\right)\right]$. We have

$$\log \mathbb{E}\left[\exp(tY)\right] = \log \mathbb{E}\left[\exp(tY^{\mathsf{E}})\right] + \log \mathbb{E}\left[\exp(tY^{\mathsf{V}})\right]$$

$$= \sum_{C \in \mathcal{C}^{\mathsf{E}} \setminus \mathcal{C}_1^{\mathsf{E}}} \mu_{|C|}^{\mathsf{E}} + \sum_{C \in \mathcal{C}^{\mathsf{V}} \setminus F} \mu_{|C|}^{\mathsf{V}}$$

$$\leq \sum_{C \in \mathcal{C}^{\mathsf{E}} \setminus \mathcal{C}_1^{\mathsf{E}}} \frac{|C|}{2} \mu_2^{\mathsf{E}}(t) + \sum_{C \in \mathcal{C}^{\mathsf{V}} \setminus F} \frac{|C|}{2} \mu_2^{\mathsf{V}}(t),$$

where the inequality is because $\mu_k^{\mathsf{E}}(t) \leq \frac{k}{2}\mu_2^{\mathsf{E}}(t)$ and $\mu_k^{\mathsf{V}}(t) \leq \frac{k}{2}\mu_2^{\mathsf{V}}(t)$ for any $k \geq 2$. We note that

$$\mu_2^{\mathsf{E}}(t) = -\frac{1}{2}\log\left(1 + \frac{\rho^2}{1-\rho^2}(4t - 4t^2)\right) < 0, \text{ for any } 0 < t < 1.$$

Consequently,

$$\log \mathbb{E}\left[\exp(tY)\right] \leq \sum_{C \in \mathcal{C}^{\mathsf{E}} \setminus \mathcal{C}_1^{\mathsf{E}}} \frac{|C|}{2} \mu_2^{\mathsf{E}}(t) + \sum_{C \in \mathcal{C}^{\mathsf{V}} \setminus F} \frac{|C|}{2} \mu_2^{\mathsf{V}}(t)$$

$$\leq \frac{1}{2}\left(N_k - \frac{k}{2}\right) \mu_2^{\mathsf{E}}(t) + \frac{k}{2}\mu_2^{\mathsf{V}}(t),$$

where the inequality is because $\sum_{C \in \mathcal{C}^{\mathsf{E}} \setminus \binom{F}{2}} |C| = \binom{n}{2} - \binom{n-k}{2} = N_k$, $\sum_{C \in \mathcal{C}^{\mathsf{V}} \setminus F} |C| = n - (n-k) = k$, and $|\mathcal{C}_1^{\mathsf{E}} \setminus \binom{F}{2}| \leq \frac{k}{2}$. We finish the proof of Lemma 4.

### D.5 Proof of Lemma 5

In this subsection, without loss of generality, we assume $V(G_1) = V(G_2) = [n]$. Define adjacent matrices $A, B \in \mathbb{R}^{n \times n}$ with $A_{ij} = \beta_{ij}(G_1)$ and $B_{ij} = \beta_{ij}(G_2)$ for any $1 \leq i < j \leq n$. Let $X, Y \in \mathbb{R}^{n \times 1}$ with $X_i = \boldsymbol{x}_i$ and $Y_i = \boldsymbol{y}_i$ with $1 \leq i \leq n$. For any $\pi \in \mathcal{S}_n$, define $A^\pi \in \mathbb{R}^{n \times n}$ with $A_{ij}^\pi = A_{\pi(i)\pi(j)}$ for any $1 \leq i < j \leq n$, $X^\pi \in \mathbb{R}^{n \times 1}$ with $X_i^\pi = X_{\pi(i)}$ for any $1 \leq i \leq n$. For two matrices $A$ and $B$, define the inner product as $\langle A, B \rangle = \sum_{1 \leq i < j \leq n} A_{ij}B_{ij}$. Then,

$$S_\pi(G_1, G_2) = \varphi(\rho) \sum_{e \in E(G_1)} \beta_e(G_1)\beta_{\pi(e)}(G_2) + \varphi(r) \sum_{v \in V(G_1)} \boldsymbol{x}_v \boldsymbol{y}_{\pi(v)}$$

$$= \varphi(\rho)\langle A, B^\pi \rangle + \varphi(r)\langle X, Y^\pi \rangle.$$

For any $\pi_1 \neq \pi_2 \in \mathcal{T}_2$ with $\mathsf{d}(\pi_1, \pi_2) = 3$,

$$\mathbb{P}\left[(G_1, G_2) \in \mathcal{E}(\pi^*, \pi_1) \cap \mathcal{E}(\pi^*, \pi_2)\right]$$

$$= \mathbb{E}\left[\mathbf{1}_{\left\{S_{\pi^*}(G_1,G_2) \leq S_{\pi_1}(G_1,G_2)\right\}} \mathbf{1}_{\left\{S_{\pi^*}(G_1,G_2) \leq S_{\pi_2}(G_1,G_2)\right\}}\right]$$

$$\leq \mathbb{E}\left[\exp\left(\frac{1}{2}\left(S_{\pi_1}(G_1,G_2) - S_{\pi^*}(G_1,G_2)\right)\right)\exp\left(\frac{1}{2}\left(S_{\pi_2}(G_1,G_2) - S_{\pi^*}(G_1,G_2)\right)\right)\right]$$

$$= \mathbb{E}\left[\exp\left(\varphi(\rho)\left(\frac{1}{2}\langle A, B^{\pi_1} \rangle + \frac{1}{2}\langle A, B^{\pi_2} \rangle - \langle A, B^{\pi^*} \rangle\right)\right)\right]$$

$$\cdot \mathbb{E}\left[\exp\left(\varphi(r)\left(\frac{1}{2}\langle X, Y^{\pi_1} \rangle + \frac{1}{2}\langle X, Y^{\pi_2} \rangle - \langle X, Y^{\pi^*} \rangle\right)\right)\right].$$

For simplicity, we denote $\mathrm{d}A\mathrm{d}B = \mathrm{d}A_{12}\mathrm{d}A_{13}\cdots\mathrm{d}A_{n-1\,n}\mathrm{d}B_{12}\mathrm{d}B_{13}\cdots\mathrm{d}B_{n-1\,n}$. For the first term, we note that

$$\mathbb{E}\left[\exp\left(\varphi(\rho)\left(\frac{1}{2}\langle A, B^{\pi_1}\rangle + \frac{1}{2}\langle A, B^{\pi_2}\rangle - \langle A, B^{\pi^*}\rangle\right)\right)\right]$$

$$= \left(\frac{1}{2\pi\sqrt{1-\rho^2}}\right)^{\binom{n}{2}}$$

$$\cdot \int\cdots\int \exp\left(\frac{\varphi(\rho)}{2}\left(\langle A, B^{\pi_1}\rangle + \langle A, B^{\pi_2}\rangle\right) - \frac{1}{2(1-\rho^2)}\left(\|A\|_F^2 + \|B\|_F^2\right)\right)\mathrm{d}A\mathrm{d}B.$$

Let $\mathrm{vec}(A) = (A_{12}, A_{13}, \cdots, A_{21}, \cdots A_{n-1\,n})$ for any adjacent matrix $A$. For any $\pi_1 \neq \pi_2 \in \mathcal{T}_2$, define permutation matrices $\Pi_1^{\mathsf{E}}$ and $\Pi_2^{\mathsf{E}} \in \{0,1\}^{\binom{n}{2}\times\binom{n}{2}}$ as

$$\mathrm{vec}(B^{\pi_1}) = \Pi_1^{\mathsf{E}}\mathrm{vec}(B), \quad \mathrm{vec}(B^{\pi_2}) = \Pi_2^{\mathsf{E}}\mathrm{vec}(B).$$

Then,

$$\frac{\varphi(\rho)}{2}\left(\langle A, B^{\pi_1}\rangle + \langle A, B^{\pi_2}\rangle\right) - \frac{1}{2(1-\rho^2)}\left(\|A\|_F^2 + \|B\|_F^2\right)$$

$$= \frac{\rho}{2(1-\rho^2)}\mathrm{vec}(A)^\top(\Pi_1^{\mathsf{E}} + \Pi_2^{\mathsf{E}})\mathrm{vec}(B) - \frac{1}{2(1-\rho^2)}(\|\mathrm{vec}(A)\|_2^2 + \|\mathrm{vec}(B)\|_2^2)$$

$$= -\frac{1}{2(1-\rho^2)}\begin{bmatrix}\mathrm{vec}(A)\\\mathrm{vec}(B)\end{bmatrix}^\top \Sigma \begin{bmatrix}\mathrm{vec}(A)\\\mathrm{vec}(B)\end{bmatrix},$$

where $\Sigma \triangleq \begin{bmatrix} I_{\binom{n}{2}\times\binom{n}{2}} & -\frac{\rho}{2}(\Pi_1^{\mathsf{E}} + \Pi_2^{\mathsf{E}}) \\ -\frac{\rho}{2}(\Pi_1^{\mathsf{E}} + \Pi_2^{\mathsf{E}})^\top & I_{\binom{n}{2}\times\binom{n}{2}} \end{bmatrix}$. Therefore,

$$\mathbb{E}\left[\exp\left(\varphi(\rho)\left(\frac{1}{2}\langle A, B^{\pi_1}\rangle + \frac{1}{2}\langle A, B^{\pi_2}\rangle - \langle A, B^{\pi^*}\rangle\right)\right)\right]$$

$$= \left(\frac{1}{2\pi\sqrt{1-\rho^2}}\right)^{\binom{n}{2}}\int\cdots\int \exp\left(-\frac{1}{2(1-\rho^2)}\begin{bmatrix}\mathrm{vec}(A)\\\mathrm{vec}(B)\end{bmatrix}^\top \Sigma \begin{bmatrix}\mathrm{vec}(A)\\\mathrm{vec}(B)\end{bmatrix}\right)\mathrm{d}A\mathrm{d}B$$

$$= \sqrt{\frac{(1-\rho^2)^{\binom{n}{2}}}{\det(\Sigma)}}.$$

Let $\sigma \triangleq \pi_2 \circ \pi_1^{-1}$ Recall that $\mathcal{C}_i^{\mathsf{E}}$ and $\mathcal{C}_i^{\mathsf{V}}$ denote the set of edge orbits and node orbits with length $i$ induced by $\sigma$. It follows from (Dai et al., 2019a, Lemmas 4.2 and 4.3) that

$$\det(\Sigma) = \prod_{k=1}^n\left(\prod_{j=1}^k\left(1 - \frac{\rho^2}{2}\left(1 + \cos\left(\frac{2\pi j}{k}\right)\right)\right)\right)^{|\mathcal{C}_k^{\mathsf{E}}|}$$

$$\geq (1-\rho^2)^{|\mathcal{C}_1^{\mathsf{E}}|}\prod_{k=2}^n(1-\rho^2)^{k|\mathcal{C}_k^{\mathsf{E}}|/2} = (1-\rho^2)^{\frac{1}{2}\left(\binom{n}{2}+|\mathcal{C}_1^{\mathsf{E}}|\right)},$$

where the inequality follows from Lemma 8 and the last equality is because $\sum_{k=1}^n k|\mathcal{C}_k^{\mathsf{E}}| = \binom{n}{2}$. Therefore,

$$\mathbb{E}\left[\exp\left(\varphi(\rho)\left(\frac{1}{2}\langle A, B^{\pi_1}\rangle + \frac{1}{2}\langle A, B^{\pi_2}\rangle - \langle A, B^{\pi^*}\rangle\right)\right)\right] \leq (1-\rho^2)^{\frac{1}{4}\left(\binom{n}{2}-|\mathcal{C}_1^{\mathsf{E}}|\right)}.$$

Similarly,

$$\mathbb{E}\left[\exp\left(\varphi(r)\left(\frac{1}{2}\langle X, Y^{\pi_1}\rangle + \frac{1}{2}\langle X, Y^{\pi_2}\rangle - \langle X, Y^{\pi^*}\rangle\right)\right)\right] \leq (1-r^2)^{\frac{d(n-|\mathcal{C}_1^{\mathsf{V}}|)}{4}}.$$

For any $\pi_1 \neq \pi_2 \in \mathcal{T}_2$ with $\mathsf{d}(\pi_1, \pi_2) = 3$. Let $F \triangleq \{i \in [n] : \pi_1(i) = \pi_2(i)\}$. Then there exists $1 \leq i, j, k \leq n$ with $i \neq j \neq k$ such that,

$$\pi_1(i) = \pi_2(j), \pi_1(j) = \pi_2(k), \pi_1(k) = \pi_2(i), \text{ and } \pi_1(\ell) = \pi_2(\ell) \text{ for any } \ell \in [n] \setminus \{i, j, k\}.$$

Then,

$$|\mathcal{C}_1^{\mathsf{V}}| = |\{\ell \in [n] : \ell \notin \{i, j, k\}\}| = n - 3,$$

$$|\mathcal{C}_1^{\mathsf{E}}| = |(\ell_1, \ell_2) : 1 \leq \ell_1 < \ell_2 \leq n, \ell_1, \ell_2 \notin \{i, j, k\}| = \binom{n}{2} - \binom{n-3}{2} = 3(n-2).$$

Consequently,

$$\mathbb{P}\left[(G_1, G_2) \in \mathcal{E}(\pi^*, \pi_1) \cap \mathcal{E}(\pi^*, \pi_2)\right]$$

$$\leq \mathbb{E}\left[\exp\left(\varphi(\rho)\left(\frac{1}{2}\langle A, B^{\pi_1}\rangle + \frac{1}{2}\langle A, B^{\pi_2}\rangle - \langle A, B^{\pi^*}\rangle\right)\right)\right]$$

$$\cdot \mathbb{E}\left[\exp\left(\varphi(r)\left(\frac{1}{2}\langle X, Y^{\pi_1}\rangle + \frac{1}{2}\langle X, Y^{\pi_2}\rangle - \langle X, Y^{\pi^*}\rangle\right)\right)\right]$$

$$\leq (1 - \rho^2)^{\frac{3(n-2)}{4}}(1 - r^2)^{\frac{3d}{4}}.$$

# E    AUXILIARY RESULTS

**Lemma 6.** *For $0 < \rho^2 < \frac{1}{10}$, we have*

$$-\frac{1 - \rho^2}{2\rho^2}\log\left(1 - \frac{\rho^2(2 + \rho^2)}{(1 - \rho^2)^2}\right) \leq 1 + 4\rho^2.$$

*Proof.* Let $x \triangleq \rho^2 \in (0, \frac{1}{10})$ and define

$$t \triangleq \frac{x(2 + x)}{(1 - x)^2}, \quad (0 < t < 1).$$

Since $x < \frac{1}{4}$, we have

$$1 - t = \frac{(1 - x)^2 - x(2 + x)}{(1 - x)^2} = \frac{1 - 4x}{(1 - x)^2} > 0.$$

For any $0 < t < 1$,

$$-\log(1 - t) = \sum_{k=1}^{\infty}\frac{t^k}{k} = t + \sum_{k=2}^{\infty}\frac{t^k}{k} \leq t + \frac{1}{2}\sum_{k=2}^{\infty}t^k = t + \frac{t^2}{2(1 - t)}.$$

Consequently, we have

$$-\frac{1 - x}{2x}\log\left(1 - \frac{x(2 + x)}{(1 - x)^2}\right) - (1 + 4x) \leq \frac{1 - x}{2x}\left(t + \frac{t^2}{2(1 - t)}\right) - (1 + 4x)$$

$$= \frac{3x\left(-21x^2 + 20x - 2\right)}{4(4x^2 - 5x + 1)}.$$

For $0 < x < \frac{1}{4}$, we have $4x^2 - 5x + 1 > 0$. The numerator factor $f(x) = -21x^2 + 20x - 2$ is strictly increasing on $[0, \frac{1}{10}]$ (since $f'(x) = -42x + 20 > 0$), and $f\left(\frac{1}{10}\right) = -0.21 < 0$. Thus $f(x) < 0$ for all $0 < x \leq \frac{1}{10}$, making the whole fraction nonpositive. Therefore, for any $0 < \rho^2 < \frac{1}{10}$,

$$-\frac{1 - \rho^2}{2\rho^2}\log\left(1 - \frac{\rho^2(2 + \rho^2)}{(1 - \rho^2)^2}\right) \leq 1 + 4\rho^2. \qquad \square$$

**Lemma 7** (Chernoff's inequality for Chi-squared distribution). *Suppose $\xi$ follows the chi-squared distribution with $n$ degrees of freedom. Then, for any $\delta > 0$,*

$$\mathbb{P}\left[\xi > (1 + \delta)n\right] \leq \exp\left(-\frac{n}{2}\left(\delta - \log(1 + \delta)\right)\right), \tag{22}$$

$$\mathbb{P}\left[\xi < (1 - \delta)n\right] \leq \exp\left(-\frac{\delta^2}{4}n\right). \tag{23}$$

*Proof.* The result follows from (Ghosh, 2021, Theorems 1 and 2). □

**Lemma 8.** *For any integer $k \geq 1$ and any $0 \leq \rho \leq 1$, we have*

$$\prod_{j=1}^{k} \left( \left( 1 - \frac{\rho^2}{2} \right) - \frac{\rho^2}{2} \cos \frac{2\pi j}{k} \right) \geq (1 - \rho^2)^{k/2}.$$

*Proof.* We note that

$$\left( 1 - \frac{\rho^2}{2} \right) - \frac{\rho^2}{2} \cos \frac{2\pi j}{k} = 1 - \rho^2 \cos^2 \left( \frac{\pi j}{k} \right),$$

so the product equals

$$P = \prod_{j=1}^{k} \left( 1 - \rho^2 \cos^2 (\frac{\pi j}{k}) \right).$$

For any $a, t \in [0, 1]$, we have $1 - at \geq (1 - a)^t$, which follows from concavity of $g(t) = \ln(1 - at) - t \ln(1 - a)$ and $g(0) = g(1) = 0$. Applying this with $a = \rho^2$ and $t = \cos^2(\pi j/k)$ yields

$$1 - \rho^2 \cos^2 (\frac{\pi j}{k}) \geq (1 - \rho^2)^{\cos^2(\pi j/k)}.$$

Multiplying over $j = 1, \ldots, k$ gives

$$P \geq (1 - \rho^2)^{\sum_{j=1}^{k} \cos^2(\pi j/k)}.$$

Since $\sum_{j=1}^{k} \cos^2(\pi j/k) = \sum_{j=1}^{k} \frac{1 + \cos(2\pi j/k)}{2} = \frac{k}{2}$, we conclude that

$$P = \prod_{j=1}^{k} \left( \left( 1 - \frac{\rho^2}{2} \right) - \frac{\rho^2}{2} \cos \frac{2\pi j}{k} \right) \geq (1 - \rho^2)^{k/2}.$$

□

