# OpenReview forum: "Sharp Statistical Limits and Algorithm for Attributed Graph Alignment"
_ICLR.cc/2026/Conference — Submitted to ICLR 2026_

### Official Review · Reviewer_FeZq · 2025-10-15

**Soundness:** 3
**Presentation:** 2
**Contribution:** 1
**Rating:** 2
**Confidence:** 4

**Summary:**

This paper studies the information-theoretic limits of graph alignment when both edge structures and node covariates are observed, modeled through a feature-correlated Gaussian Wigner model. The authors show that incorporating node features improves the recovery thresholds for both partial and exact recovery compared to the purely structural (correlated Wigner) case. On the algorithmic side, they propose a practical approach based on a Birkhoff relaxation of the quadratic assignment problem (QAP) corresponding to the maximum likelihood estimator, and evaluate its performance on synthetic and real datasets.

**Strengths:**

The main strength of the paper lies in establishing, in a rigorous manner, the information-theoretic thresholds for exact and partial recovery in the graph matching problem under the feature-correlated Gaussian Wigner model. The analysis seems technically sound and provides a consistent extension of existing results to a setting that combines structural and feature information.

**Weaknesses:**

1) **Motivations.** The new model introduced in this work is motivated by the existence of networks with negatively correlated features or negative edge weights (lines 44–48). However, the formal definition of the model only considers positive correlations $r\in (0,1)$, and the experimental datasets involve graphs with positive edge weights. This discrepancy weakens the motivation. Moreover, the influence of node covariates on graph matching has already been studied under models such as the correlated Gaussian-attributed Erdős–Rényi model. The present model thus appears to be a slight variation, primarily chosen for analytical tractability rather than capturing new phenomena.

2) **Novelty of the contribution.** From a technical standpoint, it is unclear what new challenges the proposed model introduces. The proof seems to mimic the approach used by Wu et al. (2022). Because edges and node covariates are assumed independent, the likelihood conveniently decomposes into two independent terms, making the analysis of the key statistic  $Z$ (used to bound the misalignment probability) relatively straightforward.

3) **Presentation and clarity of the writing.** The presentation could be significantly improved by clarifying the motivation and the formal definition of the model (see remarks below) and by providing more explicit comparisons with prior work, especially regarding the proof techniques used to establish the main results. The limitations of the lower bound should also be discussed, as it does not always match the upper bound; this gap weakens the claim that the derived threshold is optimal.

**Questions:**

1) The definition of the model needs to be clarified. The vertex sets $V(G_1)$ and $V(G_2)$ are not defined. Moreover, the definitions of $G_1$ and $G_2$ themselves rely on these vertex sets, creating a circular dependence.

2) There is no discussion after Theorem 2 about the condition $r\geq 40/d$, and the additional $4\ log\ d$ additive factor appearing in the lower bound. To my understanding, the lower bound matches the upper bound only if one assumes a specific scaling for $d=d(n)$. This condition should be made explicit.

3) The proof sketch should emphasize what specific challenges arise in the new setting studied. As currently written, it follows the standard approach for deriving information-theoretic thresholds and does not make clear what aspects of the new setting require different arguments.

4) I don't understand the computational complexity estimate: for each gradient step, one needs to compute $A_1 \Pi^{(t)}$, but since $ \Pi^{(t)}$ is not a permutation matrix, I think the sample complexity should correspond to the general sample complexity for dense matrix multiplication and is not of order $n^2$.

5) If I'm not mistaken, the use of the Sinkhorn (1964) projection algorithm requires the input matrix to have non-negative entries, but due to the gradient step, it is not clear that $\Pi^{(t)}$ still has non-negative entries.

6) Typos:
- In Remark 2, isn't the constraint added to program 8 instead of 7?
- line 294: "we derive at the quadratic programming"
- Ponctuation is missing after equation (5).
- Legend in Figure 3: it should be QPAlign and not QPALign

---

> ### Author Response · Authors · 2025-11-20
> **Response letter to Reviewer FeZq (part 1)**
>
> We thank the reviewer for the valuable comments.
>
> **Q1: Clarifications on the definitions of the model.**
>
> Thanks for pointing it out! We have revised the manuscript accordingly to make the definition fully self-contained. Specifically, we rewrite the Definition 1 with "Let $G_1$ and $G_2$ be two weighted random graphs with vertex sets $V(G_1),V(G_2)$ such that $|V(G_1)| = |V(G_2)| = n$.
> Let $\pi^\star$ denote..." and then state the edge and feature conditions under this notation.
>
>
> **Q2: Discussions on the assumptions in Theorems and the additional gap $4\log d$.**
>
> We thank the reviewer for pointing out that the origin of the condition $r^2 \ge 40/d$ and the additional $4\log d$ term in the lower bound was not sufficiently explained. We agree and clarified this after Theorem 2 in the revised version.
>
> - **Role of the assumption $r^2 \ge 40/d$ and the extra $4\log d$ term.** The condition $r^2 \ge 40/d$ is *not* needed to obtain the correct rate for exact recovery: even without this assumption, our impossibility result for partial recovery already implies a rate-optimal condition of the form
> $
> n\log\frac{1}{1-\rho^2} + d\log\frac{1}{1-r^2} \lesssim \log n.
> $
> We impose $r^2 \ge 40/d$ only in order to **sharpen the constant** in the converse bound, by invoking a large-deviation inequality of Kunisky and Niles-Weed (SODA 2022), which requires a constraint of the form $r^2 \gtrsim 1/d$. This refinement introduces an additional $4\log d$ term in our lower bound. The appearance of such a $4\log d$ term is also known in Gaussian database recovery problems without graph structure, and removing this logarithmic slack in $d$ remains open (see, e.g.,  Kahraman, 2024).
>
>
>
>
> - **Tight results on $d = d(n)$.** By Theorem 2, when $d = \omega(\log n)$ and $d = n^{o(1)}$, we can derive sharp information-theoretic results, namely $n\log(1/(1-\rho^2))+d\log(1/(1-r^2))$ near $(4\pm \epsilon)\log n$. When $d = \omega(\log n)$, we can derive rate-optimal results for both partial and exact recoveries. As for the assumption $d = \omega(\log n)$, it is standard in Gaussian database alignment: in the low per-feature correlation regime each feature coordinate carries only $o(1)$ mutual information, so the total information scales as $o(d)$, while feasibility of alignment requires $\Theta(\log n)$ bits; hence one necessarily has $d =\omega( \log n)$ (see, e.g., Dai, 2023).
>
> **Q3: Challenges in the new settings.**
>
> We thank the reviewer for this comment. While our overall strategy indeed follows the common information-theoretic analysis, the featured Gaussian Wigner setting introduces two specific challenges that we will highlight more clearly in the proof sketch:
>
> - (i) To the best of our knowledge, existing attributed alignment models do not provide information-theoretic thresholds for partial recovery, whereas we derive essentially tight partial-recovery thresholds and then relate them to exact recovery. The main technical challenge here is that partial recovery requires a much more refined analysis of the estimator: instead of merely ruling out or establishing exact recovery, we must precisely control the overlap of the estimator as a function of $(\rho,r)$. Our estimator $\hat\pi$ is the MLE under the featured Gaussian Wigner model, which admits a convenient variational form; this allows us to carry out a delicate large-deviation analysis and obtain sharp thresholds for any prescribed overlap level, and then connect these partial-recovery thresholds to the exact-recovery regime.
>
> - (ii) We analyze an estimator that carefully combines graph structure and node features in a single objective, while previous works typically treat the structural and feature parts in a more decoupled manner, so we must control a joint score and the resulting trade-off between structural and feature correlations.
>
> **Q4: Computational complexity.**
>
> We thank the reviewer for catching this mistake. For dense adjacency matrices, the cost of computing $A_1\Pi^{(t)}$ and $\Pi^{(t)}A_2$ is indeed $O(n^3)$ (or $O(n^\omega),2<\omega<3$ with fast matrix multiplication), not $O(n^2)$ as we wrote. We have revised accordingly in the paper (see line 348).
>
> **Q5: Details on Sinkhorn algorithm.**
>
> We thank the reviewer for pointing this out.
> Indeed, the classical Sinkhorn algorithm assumes a non-negative input matrix. In our implementation, after each gradient step we first apply an element-wise truncation
> $(\Pi^{(t)})\_{+} := \max(\Pi^{(t)}, 0)$,
> and then run the Sinkhorn iterations on $(\Pi^{(t)})_+$. Hence the matrix fed into Sinkhorn always has non-negative entries. In practice, the negative entries produced by the gradient step are small in magnitude, so this truncation has negligible impact on the solution but allows us to keep the fast Sinkhorn-based projection. We have clarified this preprocessing step in the revised version.
>
> **Q6: Typos.**
>
> We thank the reviewer for carefully spotting these typos. We have corrected them in the revision.

---

> ### Author Response · Authors · 2025-11-20
> **Response letter to Reviewer FeZq (part 2)**
>
> **Q7: Motivations.**
>
> We thank the reviewer for these helpful comments on the motivation. We have added clarifications in line 62.
>
> (i) **Sign of correlations.** The restriction to positive correlations in Definition 1 is purely for convenience. If edge weights are negatively correlated, say $(\beta_e(G_1), \beta_{\pi^\star(e)}(G_2))$ has correlation $\rho<0$, then $(\beta_e(G_1), -\beta_{\pi^\star(e)}(G_2))$ has correlation $-\rho>0$; equivalently, the pair $(G_1,G_2)$ with parameter $\rho$ has the same distribution as $(G_1,\tilde G_2)$ with parameter $|\rho|$, where $\tilde G_2$ is obtained from $G_2$ by flipping the sign of all edge weights. The same symmetry holds for node features: if $(\mathbf{x}\_v,\mathbf{y}\_{\pi^\star(v)})$ has correlation $r<0$, then $({\mathbf{x}}_v,-{\mathbf{y}}_{\pi^\star(v)})$ has correlation $-r>0$. Since our likelihood, estimator and thresholds are invariant under these sign flips, all results depend only on $\rho^2$ and $r^2$. In the revision we will explicitly allow $\rho,r\in(-1,1)$ in the definition and state this symmetry, making clear that the model covers negatively correlated features and signed edge weights as in our motivating examples.
>
> (ii) **Relation to existing attributed models.** Compared to correlated Gaussian-attributed Erdős–Rényi models, our featured Gaussian Wigner model is not only chosen for tractability: it is designed to capture settings where both edge weights and features are naturally continuous and possibly signed, and it unifies purely structural Gaussian Wigner and Gaussian database models as special cases. On the technical side, we derive essentially tight information-theoretic thresholds (for both partial and exact recovery) for an estimator that jointly uses graph structure and node covariates; existing attributed models, to the best of our knowledge, either focus on exact recovery only or treat graph and feature information in a more decoupled manner.
>
>
> **Q8: Novelty of the contribution.**
>
> From a technical standpoint, our contribution departs from existing work in two main ways:
> (i) Even for attributed alignment models, we are not aware of prior results that provide *information-theoretic thresholds for partial recovery*. Our analysis derives essentially tight partial-recovery thresholds in the featured Gaussian Wigner setting and then connects them to exact recovery.
> (ii) We acknowledge that, because edges and node covariates are independent, the log-likelihood and its mgf factorize into structural and feature parts. The nontrivial aspect is that we analyze the **true MLE that jointly uses both sources** in a single quadratic assignment objective. Proposition 2 is specific to this MLE: its proof relies on the exact likelihood ratio, and if one perturbs the coefficients away from the MLE weights, the argument no longer yields the same sharp constant in the threshold. We have revised the proof sketch to explicitly acknowledge the factorization and to emphasize that the novelty lies in the precise MLE-based combination and the resulting sharp constants.
>
>
> **Q9: Presentation and writing.**
>
> Thanks for the valuable suggestions. We have revised the paper accordingly. First, we clarified the model in the introduction by more carefully explaining the featured Gaussian Wigner formulation and its connection to attributed graph alignment. Second, we made the comparison with prior work more explicit in the introduction, clearly positioning our information-theoretic thresholds and algorithmic results relative to existing attributed alignment and Gaussian database alignment models. Third, we added Theorem 3 to provide a convergence guarantee for QPAlign and expanded the discussion on how its empirical phase transitions relate to the information-theoretic thresholds. We also clarified several assumptions in the statements of the main theorems and added a short conclusion section summarizing our contributions and future directions.
>
>
> **References**
>
> [1] Dmitriy Kunisky and Jonathan Niles-Weed. Strong recovery of geometric planted matchings. Proceedings of the 2022 Annual ACM–SIAM Symposium on Discrete Algorithms (SODA), pages 834–876, 2022.
>
> [2] Zeynep Kahraman. Database alignment: fundamental limits and multiple databases setting. PhD thesis, Boston University, 2024.
>
> [3] Osman Emre Dai, Daniel Cullina, and Negar Kiyavash. Gaussian database alignment and Gaussian planted matching. arXiv preprint arXiv:2307.02459, 2023.

---

> > ### Comment · Reviewer_FeZq · 2025-11-26
> >
> > Thank you for the clarifications and the corrections. It would, however, have been easier to assess the revision had the changes been highlighted in color.
> >
> > That being said, I remain unconvinced about the insights this work offers. Extending the Gaussian-attributed Erdős–Rényi model to the featured Gaussian Wigner model appears to be a relatively incremental step. The analysis of the correlated Wigner model is well known, and the fact that incorporating node features yields an additive improvement in the recovery threshold is not surprising. One could equally introduce categorical features, Poisson-distributed weights, or other natural variants, and motivate them in similar terms as “models that incorporate both structural and feature information, beyond purely topology-based settings.” But in my opinion, one should either have strong practical motivations (application to a specific dataset, new algorithmic design, etc) or show that by adding these twists the theoretical analysis changes significantly and provides new insights, i.e., the slight change of the model has substantial theoretical consequences and requires the development of new techniques.
> >
> > However, based on the current presentation of Section 2, the technical challenges arising from continuous weights are not clearly articulated. The exposition does not convey the core difficulties of the problem—most of which are encapsulated in Propositions 1 and 2—nor does it explain the techniques employed (for instance, the cycle decomposition used to compute the m.g.f. of $Z$). In addition, the manuscript does not acknowledge that these techniques are not new and were already developed in previous work (e.g., Wu et al., 2022).

---

> > > ### Author Response · Authors · 2025-11-27
> > > **Replies to Reviewer FeZq**
> > >
> > > Thank you very much for your detailed follow-up and for pointing out where the exposition was unclear. In the new revision, all changes addressing your comment are highlighted in blue in the PDF.
> > >
> > > **1. Motivation and modeling choice (Section 1, around line 44).**
> > >
> > > To better justify the move from Erdős–Rényi–type models to a featured Gaussian Wigner model, we added a short paragraph in the introduction (Section 1, line 44) that contrasts existing work on attributed alignment with binary/community-based models and several representative real-world examples where edges are naturally weighted and nodes carry attributes (gene co-expression networks, social and information networks, etc.). This paragraph explains that our featured correlated Gaussian Wigner model is intended as a simple canonical model for such weighted attributed networks, complementing the Gaussian-attributed ER/SBM literature rather than just adding a technical twist.
> > >
> > > **2. Technical challenges and relation to prior work (Section 2, around line 209).**
> > > In Section 2, we inserted a brief technical overview (around line 209) that explicitly states the core difficulty behind Propositions 1 and 2—controlling the MLE score difference under a mixed Gaussian channel with continuous edge weights and high-dimensional features—and summarizes the main tools used. We now make clear that
> > >
> > > (i) for permutations at macroscopic Hamming distance we *adapt* the cycle decomposition and Gaussian moment-generating-function computation from the correlated Gaussian Wigner literature to this structural–feature setting;
> > >
> > > (ii) for permutations very close to the ground truth we view the score difference as a quadratic form in a jointly Gaussian vector and use a Hanson–Wright–type inequality after decorrelating the coordinates. This paragraph is meant to convey both where the continuous-weight setting creates additional work and how our analysis builds on, and
> > > extends, existing techniques.
> > >
> > > We hope these changes make both the modelling motivation and the technical contribution of Section 2 clearer and more transparent.

---

### Official Review · Reviewer_2pFs · 2025-10-30

**Soundness:** 3
**Presentation:** 3
**Contribution:** 2
**Rating:** 4
**Confidence:** 4

**Summary:**

This paper studies the graph alignment (or graph matching) problem under a featured correlated Gaussian Wigner model, where the goal is to recover a latent bijective mapping between two correlated graphs. Each graph consists of correlated weighted edges—modeled as correlated standard normals with correlation coefficient
$\rho \in (0,1)$—and correlated node features, modeled as multivariate Gaussian vectors with correlation $r \in (0,1)$.

The authors first derive information-theoretic limits for partial and exact recovery using a maximum-likelihood estimation (MLE) framework. Their results show that the combined signal-to-noise ratio (SNR) from both edges and features, given by
$n \log\frac{1}{1-\rho^2} + d \log\frac{1}{1-r^2},$
must exceed the threshold
$(4\pm \epsilon)\log n$ for successful recovery.

Since MLE is computationally infeasible due to its exponential complexity in $n$, the authors propose a polynomial-time algorithm (QPAlign) based on a quadratic programming relaxation of the MLE objective. The relaxation replaces the permutation constraint with the Birkhoff polytope (the set of doubly stochastic matrices) and adds a regularizer to guide the solution. Numerical experiments on both synthetic correlated Gaussian-attributed Erdős–Rényi graphs and real-world datasets (ACM-DBLP and Douban) demonstrate the effectiveness of QPAlign for partial recovery.

**Strengths:**

* The paper extends classical correlated Gaussian Wigner models to include correlated node features, establishing a unified featured correlated Gaussian Wigner model and showing that the joint use of edges and features enhances the effective SNR for alignment.
* It proposes a computationally efficient relaxation of the quadratic assignment formulation using the Birkhoff polytope, achieving $O(n^3)$ complexity and demonstrating promising empirical results.

**Weaknesses:**

* Gap in information-theoretic bounds: The achievability and converse results (Theorems 1 & 2) are not tight. In particular, achievability requires $d = \omega(\log n)$, while the converse does not. The paper lacks discussion or intuition for this additional condition, leaving an unexplained theoretical gap.
* Limited theoretical novelty: The MLE-based analysis largely follows established techniques from prior works on correlated Gaussian Wigner models (Wu et al., 2022), Gaussian databases (Dai et al., 2019a), and correlated Gaussian-attributed ER models (Yang & Chung, 2024). Since the edge and feature correlations are independent, the overall SNR simply adds, leading to minimal new analytical challenges. The authors should clarify any technical novelty introduced in their proofs. Additionally, a relevant reference—“Exact Matching in Correlated Networks with Node Attributes for Improved Community Recovery” (Yang & Chung, IEEE T-IT 2025)—also derives similar IT limits under correlated SBMs. The relation and any technical differences from these prior works need to be discussed.
* No theoretical guarantees for the algorithm: While the QAP relaxation via the Birkhoff polytope has been adopted in the literature (e.g., Vogelstein et al., 2015; Bonmakanti et al., 2024), the paper provides no theoretical connection between the proposed algorithm and the derived IT limits. The lack of recovery or approximation guarantees leaves a gap between theory and computation.
* Experimental limitations: The experiments evaluate QPAlign but do not compare its empirical performance against the information-theoretic thresholds derived earlier. Moreover, for real datasets such as ACM-DBLP and Douban, the node and edge distributions are highly non-uniform due to community structure, casting doubt on the practical relevance of the uniform correlated Gaussian Wigner assumption mainly considered in the theoretical derivations of this paper.

**Questions:**

1. What is the intuition behind requiring $d = \omega(\log n)$ for achievability, while the converse result has no such condition? Can this gap be closed, or is it a fundamental limitation of the current proof technique?
2. Beyond combining independent edge and feature correlations, what new analytical challenges or proof techniques distinguish this work from prior studies (Wu et al., 2022; Dai et al., 2019a; Yang & Chung, 2024)?
3. Does QPAlign possess any recovery or approximation guarantees—e.g., conditions under which it converges to the true permutation or achieves a bounded alignment error?
4. Can you establish a connection between the algorithm’s empirical success and the derived information-theoretic thresholds?
5. For real datasets with community structure, how realistic is the assumption of uniform edge/feature distributions in the correlated Gaussian Wigner model?

---

> ### Author Response · Authors · 2025-11-20
> **Response letter to Reviewer 2pFs  (part 1)**
>
> We thank the reviewer for the valuable comments.
>
> **Q1: Assumptions $d = \omega(\log n)$.**
>
> Thanks for the question.
> The assumption $d = \omega(\log n)$ is standard and essentially information–theoretically necessary in Gaussian database alignment: in the low per-feature correlation regime each feature coordinate carries only $o(1)$ mutual information, so the total information scales as $o(d)$, while feasibility of alignment requires $\Theta(\log n)$ bits; hence one necessarily has $d =\omega( \log n)$ (see, e.g., Dai, 2023). Without this assumption, our techniques still yield rate-optimal thresholds (up to universal constants), but the leading constant would no longer be sharp; we therefore impose $d = \omega(\log n)$ precisely to match the information–theoretic constant.
> We have added clarifications in line 115 on the results without this assumption.
>
> **Q2: New analytical challenges.**
>
> Beyond simply combining independent edge and feature correlations, our analysis differs from prior work in two main ways.
> First, even for attributed alignment models, we are not aware of earlier results that give information-theoretic thresholds for partial recovery; we derive essentially tight partial recovery thresholds in the featured Gaussian Wigner model and then link them to exact recovery. Second, although the likelihood factorizes into structural and feature parts, we analyze the true joint MLE as a single quadratic assignment problem: Proposition 2 is specific to this MLE and uses the exact likelihood ratio to obtain the sharp constant in the threshold, and this argument breaks down if one perturbs the coefficients away from the MLE weights. Existing works treat graph and feature information in a more decoupled way and do not provide such a sharp MLE-based joint analysis.
> We have added detailed discussion in line 519.
>
> **Q3: Recovery guarantees on QPAlign.**
>
> Thanks for the suggestion. In the revision, we have added Theorem 3, which provides a theoretical guarantee for the projected gradient descent method applied to the quadratic program (7). This provides a convergence and approximation guarantee at the level of the relaxed optimization problem.
> A full statistical guarantee that the minimizer $\Pi'$ (and its rounded permutation $\hat\pi$) coincides with the true permutation $\pi^\star$ requires a separate analysis of the tightness of the relaxation.
> Intuitively, one would need to show that, in a suitable high-signal regime, the population version of the quadratic objective has a unique minimizer at the permutation matrix for $\pi^\star$ and that the empirical objective concentrates around this population landscape, in the spirit of the tightness analyses for convex relaxations in related Gaussian graph matching models (e.g., Fan et al. ICML 2020). Establishing such a result for our nonconvex quadratic program in the featured Wigner setting appears technically challenging, and we therefore leave a sharp recovery guarantee for $\hat\pi$ as an interesting direction for future work.
>
> **Q4: Empirical success and the IT thresholds.**
>
> Thanks for the suggestion. We have added Figure 5 (line 864), which illustrates the connection between the information-theoretic thresholds and the empirical success boundary of QPAlign. The observed algorithmic behavior aligns with the information-theoretic thresholds, and further reveals a non-trivial statistical–computational gap for this problem.
>
> **Q5: Real datasets with community structure.**
>
> We agree that ACM-DBLP and Douban exhibit strong community structure and non-uniform node/edge statistics, so they are not literally generated from the homogeneous featured correlated Gaussian Wigner model. In our work, this model is used only as a standard, analytically tractable benchmark to characterize information-theoretic limits, not as a realistic generator for these datasets. QPAlign itself does not assume Gaussianity or homogeneous edge probabilities: it simply takes similarity matrices from edges and features and can be applied to general attributed graphs. In this sense, the algorithm can be interpreted in a purely deterministic way, without committing to any particular probabilistic model—much like ordinary least squares can be used without invoking the Gaussian MLE interpretation. We have added a brief discussion of this point in the conclusion and highlighted extensions to heterogeneous models with community structure (e.g., attributed SBMs or inhomogeneous/graphon-type models; Onaran et al., 2016; Yang and Chung, 2025) as an important direction for future work.

---

> ### Author Response · Authors · 2025-11-20
> **Response letter to Reviewer 2pFs (part 2)**
>
> **Q6: Gap in information-theoretical thresholds.**
>
>
> Thank you for pointing out this issue. We have added a short discussion after Theorem 2 to clarify the origin of the gap.
>
> - **On the condition $r^2 \ge 40/d$ and the extra $4\log d$ term.** This assumption is *not* needed to obtain the correct rate for exact recovery: even without it, our impossibility result for partial recovery already implies a rate-optimal condition of the form
> $n \log \frac{1}{1-\rho^2} + d \log \frac{1}{1-r^2} \lesssim \log n$.
> We impose $r^2 \ge 40/d$ only to sharpen the constant in the converse bound, by invoking a large-deviation inequality of Kunisky and Niles-Weed (SODA 2022), which introduces an additional $4\log d$ term in our lower bound. The appearance of such a $4\log d$ term is also known in Gaussian database recovery problems without graph structure, and removing this logarithmic slack in $d$ remains open (see, e.g., Kahraman, 2024).
>
> - **Tight results on $d = d(n)$.** By Theorem 2, when $d = \omega(\log n)$ and $d = n^{o(1)}$, we can derive sharp information-theoretic results, namely
> $n \log(1/(1-\rho^2)) + d \log(1/(1-r^2))$
> near $(4 \pm \varepsilon)\log n$.
> As for the assumption $d = \omega(\log n)$, it is standard and essentially information-theoretically necessary in Gaussian database alignment: in the low per-feature correlation regime each feature coordinate carries only $o(1)$ mutual information, so the total information scales as $o(d)$, while feasibility of alignment requires $\Theta(\log n)$ bits; hence one necessarily has $d = \omega(\log n)$ (see, e.g., Dai, 2023). Without this assumption, our techniques still yield rate-optimal thresholds (up to universal constants), but the leading constant would no longer be sharp.
>
> **References**
>
> [1] Osman Emre Dai, Daniel Cullina, and Negar Kiyavash. Gaussian database alignment and Gaussian planted matching. arXiv preprint arXiv:2307.02459, 2023.
>
> [2] Zhou Fan, Cheng Mao, Yihong Wu, and Jiaming Xu. Spectral graph matching and regularized quadratic relaxations: Algorithm and theory. In Proceedings of the 37th International Conference on Machine Learning (ICML), pages 2985–2995, 2020.
>
> [3] Efe Onaran, Siddharth Garg, and Elza Erkip. Optimal de-anonymization in random graphs with community structure. In 2016 50th Asilomar conference on signals, systems and computers, pp. 709–713. IEEE, 2016.
>
> [4] Joonhyuk Yang and Hye Won Chung. Exact matching in correlated networks with node attributes for improved community recovery. IEEE Transactions on Information Theory, 71(10):7916–7941, 2025.
>
> [5] Dmitriy Kunisky and Jonathan Niles-Weed. Strong recovery of geometric planted matchings. Proceedings of the 2022 Annual ACM–SIAM Symposium on Discrete Algorithms (SODA), pages 834–876, 2022.
>
> [6] Zeynep Kahraman. Database alignment: fundamental limits and multiple databases setting. PhD thesis, Boston University, 2024.

---

### Official Review · Reviewer_QxbF · 2025-10-31

**Soundness:** 2
**Presentation:** 1
**Contribution:** 1
**Rating:** 0
**Confidence:** 5

**Summary:**

This paper studies graph alignment with both edge structure and node features under the featured correlated Gaussian Wigner model. The authors establish information-theoretic thresholds for vertex mapping recovery and propose QPAlign, a quadratic programming-based algorithm.

**Strengths:**

- Addresses a relevant problem by incorporating node features alongside graph topology
- Aims to provide theoretical guarantees with information-theoretic analysis
- Proposes a practical algorithm with experimental validation

**Weaknesses:**

Critical issue with problem formulation:
- The paper suffers from a fundamental lack of clarity in its problem definition. Definition 1 is ambiguous and appears inconsistent: It is unclear whether graphs $G₁$ and $G₂$ are given as input or generated by the model. If they are given, the condition that both ${u,v}$ and ${π*(u),π*(v)}$ must be edges in $G₁$ and $G₂$ respectively only makes sense if $G₁$ and $G₂$ are isomorphic.
- Equation (1) does not clarify this confusion, as the generation process for $G₁$ and $G₂$ is not properly specified.
- If the graphs are indeed assumed to be isomorphic, the claimed results that do not depend on edge density become highly questionable.

The paper requires major revision to properly define the problem before its contributions can be evaluated.

**Questions:**

1- Are $G₁$ and $G₂$ assumed to be isomorphic? Please clarify this explicitly.
2- How exactly are $G₁$ and $G₂$ generated in your model?
3- How does edge density affect your theoretical results?

---

> ### Author Response · Authors · 2025-11-20
> **Response letter to Reviewer QxbF**
>
> We thank the reviewer for the detailed comments. We have revised the paper to clarify the model definition and hope to address the concerns as follows.
>
> (i) **Generated graphs and role of $\pi^\star$.** In our setting, $G_1$ and $G_2$ are not arbitrary given inputs but are **generated by the model together with a latent permutation**. In the revised Definition 1 we now state this explicitly: we first sample a bijection $\pi^\star:V(G_1)\to V(G_2)$, and then, for each unordered pair $\{u,v\}$, we draw the edge-weight pair $(\beta_{uv}(G_1),\beta_{\pi^\star(u)\pi^\star(v)}(G_2))$ as a correlated Gaussian vector, and similarly for the node features. Thus $(G_1,G_2,\pi^\star)$ are jointly sampled under a featured correlated Gaussian Wigner model.
>
> (ii) **No isomorphism assumption.**
> From a practical perspective, many applications of graph alignment involve noisy, weighted, or attributed networks (e.g., biological interaction networks, recommendation graphs, or social networks), where exact graph isomorphism at the unweighted level is neither realistic nor required for meaningful alignment. In these settings, it is natural to assume that the two graphs are generated from a common latent permutation and then corrupted by independent noise, rather than being exactly isomorphic as simple graphs.
>
> Technically, the phrase that both $(u,v)$ in $G_1$ and $(\pi^\star(u),\pi^\star(v))$ in $G_2$ “must be edges’’ should be understood as the standard coordinatewise coupling in correlated Gaussian Wigner models. Our graphs are weighted: every unordered pair $\{u,v\}$ carries a (Gaussian) edge weight, so there is no binary "edge present/absent" indicator as in unweighted simple graphs, and we do not assume that $G_1$ and $G_2$ are isomorphic as simple graphs. Concretely, for each unordered pair $\{u,v\}$ we jointly generate $(\beta_{uv}(G_1),\,\beta_{\pi^\star(u)\pi^\star(v)}(G_2)$ as a pair of correlated Gaussian variables with correlation $\rho$, independently across pairs. We have rewritten Definition 1 to emphasize this generative, weighted formulation and to eliminate any possible ambiguity about isomorphism assumptions.
>
> (iii) **Edge density.** In this Gaussian Wigner setting, after standardizing the (Gaussian) edge weights, there is no Erdős–Rényi--style edge-density parameter $p$ or notion of “edge present/absent’’ as in unweighted graphs. The relevant notion of signal strength is instead given by the correlation parameters $\rho$ (edges) and $r$ (features), and our information-theoretic thresholds therefore depend on $(\rho, r)$ rather than on an edge probability.
>
> We hope that these clarifications, together with the revised definition in the main text, address the reviewer's concerns about the problem formulation.

---

> > ### Comment · Reviewer_QxbF · 2025-11-20
> >
> > I still think that using the term "weighted random graphs" is rather misleading. I understand it as you have an underlying discrete graph with weights on the edges (and no weight if there is no edge). From what I understand, your model corresponds to the case where the underlying graph is the complete graph.
> >
> > In this case, it seems that your model with $r=0$ corresponds exactly to the model studied by Ganassali in "Sharp threshold for alignment of graph databases with Gaussian weights" https://arxiv.org/abs/2010.16295 where he finds an information theoretic threshold at $n\rho^2 = 4 \log n$. His result seems to contradict your result when $\rho$ is close to one.

---

> > > ### Author Response · Authors · 2025-11-21
> > > **Replies to Reviewer QxbF**
> > >
> > > Thanks for the follow-up comment and for pointing us to Ganassali’s work.
> > >
> > > On the terminology: in our paper, “weighted random graphs” refers to a complete graph whose edges carry random correlated Gaussian weights. In particular, there is no underlying sparse or unweighted graph; the randomness is entirely in the Gaussian edge weights. We believe this use is standard in the Gaussian Wigner literature.
> > >
> > > Regarding the comparison with Ganassali (2020), there is in fact no contradiction. When $r = 0$, our model coincides with the Gaussian-weighted complete graphs studied in that paper. In our notation, the information-theoretic transition for exact recovery is expressed as
> > > $n\log(1/(1-\rho^2)) \approx 4\log n$.
> > > In the regime where the transition occurs, we necessarily have $n\rho^2 \asymp \log n$, hence $\rho^2 \to 0$ and
> > > $\log(1/(1-\rho^2)) = \rho^2 + O(\rho^4) = \rho^2(1+o(1))$.
> > > Thus our threshold condition becomes
> > > $n\rho^2 \approx 4\log n$,
> > > which matches exactly the information-theoretic threshold $n\rho^2 = 4\log n$ obtained by Ganassali.
> > >
> > > For $\rho$ very close to $1$, both conditions are easily satisfied (indeed $n\rho^2 \gg \log n$ and also $n\log(1/(1-\rho^2)) \gg \log n$), so exact alignment is information-theoretically possible in both models; there is no contradiction in this regime either.

---

> > > > ### Comment · Reviewer_QxbF · 2025-11-25
> > > >
> > > > Thank you for the clarification.
> > > >
> > > > However, I continue to believe that the paper is not yet ready for publication.
> > > >
> > > > Regarding the presentation, the exposition in Section 4 is significantly clearer than that in Section 1. In particular, adopting the convention $V(G_1) = V(G_2) = [n]$ and $E(G_1) = E(G_2) = \{(i,j) : i < j\}$ from the outset would eliminate the need to introduce the unnecessary notations $V(G_1)$, $E(G_1)$, etc.
> > > >
> > > > More importantly, the definition of the model remains unclear. In your response dated November 20, you wrote: “In the revised Definition 1 we now state this explicitly: we first sample a bijection $\pi^*$…”. However, I do not see this clarification reflected in the revised version of the manuscript.
> > > >
> > > > The statements of Theorems 1 and 2 also appear problematic. The first claim of Theorem 2 is strictly stronger than the corresponding claim in Theorem 1. If this is the case, it is unclear why the first statement of Theorem 1 is still included.
> > > >
> > > > Finally, I am unable to follow the proof of Lemma 2 in Appendix D.2. Specifically, it is not clear how the Hanson–Wright inequality—stated for quadratic forms in independent random variables—can be applied when the underlying Gaussian variables $X$ and $Y$ are correlated.

---

> > > > > ### Author Response · Authors · 2025-11-27
> > > > > **Replies to Reviewer QxbF**
> > > > >
> > > > > Thanks for the follow-up comment.
> > > > >
> > > > > Q1: Definition 1 in the revised version.
> > > > >
> > > > > We have updated the tracked change version. We first fix
> > > > > $V(G_1)=V(G_2)$ and sample a bijection $\pi^\star:V(G_1)\to V(G_2)$, and only then generate
> > > > > $(G_1,G_2)$ and their attributes conditional on $\pi^\star$. Identifying $V(G_1)=V(G_2)$ with $[n]$
> > > > > is merely a relabeling of the vertex set and is equivalent to this formulation.
> > > > >
> > > > > Q2: Theorems 1 and 2.
> > > > >
> > > > > Theorem 1 gives the information-theoretic threshold for **partial recovery**, whereas
> > > > > Theorem 2 gives the threshold for **exact recovery**, which is a strictly stronger goal and
> > > > > therefore comes with different conditions. The two statements correspond to these two distinct
> > > > > regimes and are not redundant. It is standard in the graph-alignment and community-detection literature to distinguish these two regimes and to provide separate thresholds for each. The fact that the exact-recovery result implies a particular partial-recovery guarantee does not make the partial-recovery theorem redundant: it characterizes a different phase transition and applies in parameter ranges where exact recovery is still impossible.
> > > > >
> > > > > Q3: Hanson-Wright inequality.
> > > > >
> > > > > Thank you for the comment. In Appendix D.2 we apply the Hanson–Wright inequality to suitable linear combinations of the Gaussian variables (specifically, vectors of the form $X+Y$ and $X-Y$), whose coordinates are independent by construction (see line 1173). The quadratic form in Lemma 2 is expressed in terms of these independent coordinates, so the standard Hanson–Wright inequality applies.

---

### Official Review · Reviewer_cYRs · 2025-11-01

**Soundness:** 4
**Presentation:** 3
**Contribution:** 3
**Rating:** 8
**Confidence:** 3

**Summary:**

This paper addresses the problem of attributed graph alignment, where the goal is to recover a hidden vertex correspondence between two correlated graphs. The available observations consist of the pair of graphs to be matched, together with vertex-associated side information in the form of feature vectors. The main theoretical results are established under the featured correlated Gaussian Wigner model.

The authors derive the maximum likelihood estimator (MLE) and provide high-probability upper bounds on its error under suitable assumptions on the model parameters. They also establish impossibility results in the form of lower bounds, identifying conditions under which recovery is information-theoretically infeasible. The resulting threshold for successful recovery depends on the sum of two components: one reflecting the information contained in the graph structure, and the other arising from the features. This characterization demonstrates that combining both sources of information yields strictly better recovery guarantees than relying on either one alone.

Finally, the authors propose a relaxation of the MLE over the Birkhoff polytope and evaluate its empirical performance on both synthetic and real datasets.

**Strengths:**

The graph alignment problem is both challenging and highly relevant, and as the authors note, its attributed variant is the most appropriate formulation for many real-world applications. The proposed model extends the correlated Gaussian Wigner model, one of the main statistical frameworks for studying graph matching. In this sense, the work makes a meaningful contribution by addressing important gaps in the existing literature.

One of the paper’s main strengths lies in its rigor and mathematical soundness. The proofs I examined appear correct and well executed.

Another strength is the clarity of exposition, for the most part, which makes the paper accessible and easy to follow despite its technical and theoretical nature.

**Weaknesses:**

I believe the paper would benefit from the inclusion of a **conclusion section**, where the authors could elaborate on natural extensions of their work. For example, it would be interesting to discuss how the results might generalize to other attributed correlated random graph models, or to settings where dependencies exist between the edges and the features.

In addition, a brief discussion of the optimization aspects of the proposed relaxation would be valuable. For instance, since the relaxed problem is convex, one could consider guarantees for projected gradient descent methods. Although the rounding step remains more challenging to analyze, it would still be informative to comment on possible strategies to improve optimization efficiency—for example, through adaptive step sizes or other acceleration techniques. Moreover, given that the gradient is linear, the Frank–Wolfe algorithm presents itself as a natural and computationally appealing alternative.

**Questions:**

1. **Clarity of assumptions in proofs**
   As a suggestion, it would be helpful to remind the reader of certain parameter assumptions within the proofs to improve readability.
   For instance, it would be good to restate the conditions on $\rho$ and $r$ when discussing the inequality in line 1021.

2. **On the Sinkhorn method**
   The Sinkhorn method does not yield a Euclidean projection.
   Is this choice primarily motivated by computational efficiency?
   How does its performance and accuracy compare to the Dykstra-based approach?

3. **Regularization term (Remark 2)**
   I found the regularization term discussed in Remark 2 particularly interesting.
   Could the authors provide some insight into how the algorithm’s performance changes **with and without** this term?

4. **Comparison with Fan et al. (ICML 2020)**
   Related to the previous point, if I am not mistaken, in *Spectral Graph Matching and Regularized Quadratic Relaxations: Algorithm and Theory*
   (Zhou Fan, Cheng Mao, Yihong Wu, and Jiaming Xu, ICML 2020), the authors introduce a regularization term with the opposite effect—encouraging solutions closer to the center of the Birkhoff polytope.
   Could the authors clarify why the regularization in the present work acts in the opposite direction?

5. **On the assumption about $\pi^*$**
   In Section 3, the authors state that the ground truth permutation is assumed to be uniform over $ \mathcal{S}_n$ under the proposed model.
   However, the model itself appears to assume a fixed permutation.
   Could the authors elaborate on this point?
   It seems that the uniform assumption might be introduced mainly for the lower-bound argument, whereas in the model, $\pi^*$ could be viewed as a fixed but unknown parameter.

---

> ### Author Response · Authors · 2025-11-20
> **Response letter to Reviewer cYRs**
>
> We thank the reviewer for the valuable comments.
>
> **Q1: Clarity of assumptions in proofs.**
>
> Thanks for your advice! We have clarified the assumptions in the proofs to improve readability.
>
> **Q2: Sinkhorn method.**
>
> We thank the reviewer for the comment. In our algorithm, the Sinkhorn step is used purely to map the iterate back into the Birkhoff polytope; our analysis does not require this mapping to be the exact Euclidean (Frobenius) projection, only that the iterate remains doubly stochastic. We chose Sinkhorn mainly for its speed and simplicity, as it only involves iterative row/column rescaling and scales well to large instances, while a Dykstra-based Euclidean projection onto the Birkhoff polytope is typically much more time-consuming.
> We have added clarifications in line 321.
>
> **Q3: Regularization term (Remark 2) and comparison with Fan et al. (ICML 2020).**
>
> The regularization term in QPAlign is introduced solely to push the relaxed assignment matrix $\Pi$ toward $\{0,1\}$ instead of intermediate values. In Fan et al. (ICML 2020), the Frobenius-norm regularizer is primarily introduced to make the quadratic relaxation analytically tractable: adding a term of the form $\Vert\Pi\Vert_F^2$ admits an explicit solution, which is crucial for their theoretical analysis rather than for sharpening the assignment. In contrast, our goal here is to improve the practical quality of the relaxed solution. The penalty $\sum_{i,j} \Pi_{ij}(1 - \Pi_{ij})$ is minimized when $\Pi_{ij} \in \{0,1\}$, so it explicitly discourages fractional entries and pushes $\Pi$ toward the vertices of the Birkhoff polytope (i.e., permutation matrices).  We have added clarifications in line 380.
> We also conduct ablation studies on synthetic data with $n = 100$ and $d = 16$.
> In the Gaussian Wigner model, using a positive regularization weight $\mu = 0.1$ (instead of $\mu = 0$) improves the overlap on 86.57% of the parameter grid points, while in the Erdős–Rényi model the corresponding proportion is 60.74%,  where $\mu$ is the weight on the regularization term introduced in equation (8).
>
>
> **Q4: On the assumption about $\pi^\star$**
>
> We thank the reviewer for pointing out this potential confusion. We always view the ground truth permutation $\pi^\star$ as a fixed but unknown parameter: the joint distribution of the graphs and features is defined conditional on a given $\pi^\star$, and our achievability (upper bound) results are minimax in the sense that they hold over all $\pi^\star\in\mathcal{S}_{n,m}$.
>
> The assumption that "$\pi^\star$  is uniform over $\mathcal{S}_{n,m}$" is used only in the information-theoretic lower-bound argument, where we adopt a standard Bayesian reduction: we place a uniform prior on $\pi^\star$ to exploit the symmetry of the model, compute a Bayes risk under this prior, and then use that the Bayes risk lower bounds the minimax risk. Thus, $\pi^\star$ can still be regarded as a fixed but unknown parameter in the model; the uniform prior is merely a technical device for proving a minimax lower bound, not an additional modeling assumption on the data. We have revised the text accordingly to make this distinction clear (line 219).
>
> **Q5: Theoretical guarantee for projected gradient descent methods.**
>
> Thank you for the suggestion. In the revision, we have added Theorem 3, which provides a theoretical guarantee for the projected gradient descent method applied to the quadratic program (7).
>
> **Q6: The inclusion of a conclusion section.**
>
> Thank you for the suggestion. In the revision, we have added a conclusion section that summarizes our main contributions and outlined several potential extensions of this work.
>
> **Reference**
>
> [1] Zhou Fan, Cheng Mao, Yihong Wu, and Jiaming Xu. Spectral graph matching and regularized quadratic relaxations: Algorithm and theory. In Proceedings of the 37th International Conference on Machine Learning (ICML), pages 2985–2995, 2020.

---

> > ### Comment · Reviewer_cYRs · 2025-11-22
> >
> > I thank the authors for their detailed responses.
> >
> > Just one comment about **Q3: on the regularization term**. I agree with the intuition behind the inclusion of this term; pushing the solution to the boundary. On the other hand, say $W'$ is the solution of the relaxed problem with the regularization term, and
> >
> > $$\Pi'=\arg\max_{\Pi\in\mathcal{S}_n}\langle \Pi, W'\rangle_F,$$
> >
> > then it holds
> >
> > $$\Pi'=\arg\max_{\Pi\in\mathcal{S}_n}\langle \Pi, tW'+\frac{1-t}{n}11^\top\rangle_F,$$
> >
> > where $11^\top$ is the all ones matrix. Even if $W'$ has entries in $\lbrace 0,1 \rbrace$, the matrix $W'_t=tW'+\frac{1-t}{n}11^\top$, which yields the same rounding $\Pi'$, has non-fractional entries. So, in principle, fractional entries are not necessarily a problem (I agree that the rounding might be more numerically stable with entries close to 0 or 1). Of course, this regularization term does not push 'linearly' a matrix to the boundary of the polytope.
> >
> > Can the authors comment on this?

---

> ### Author Response · Authors · 2025-11-23
> **Replies to Reviewer cYRs**
>
> We thank the reviewer for the insightful comment. We agree that, once the score matrix $W'$ is fixed, the linear assignment rounding $\Pi' = \arg\max_{\Pi\in\mathcal{S}_n} \langle \Pi, W' \rangle_F$ is invariant under certain affine transformations such as $W' \mapsto tW' + \tfrac{1-t}{n}\boldsymbol{1}\boldsymbol{1}^\top$, so the example with $W_t$ correctly shows that the rounding step itself does not care whether $W'$ has fractional or nearly $\{0,1\}$ entries. Our point about the regularization term is of a different nature: it is not a post-hoc linear modification of an already computed $W'$, but a change of the relaxed objective that acts directly on $\Pi$ during optimization. This alters the location of the relaxed optimum and typically enlarges the row/column margins of the solution, making $\Pi$ sharper and closer to a permutation matrix before the final linear assignment. From an algorithmic viewpoint, our iterations perform a gradient-descent step on $\Pi$ followed by Sinkhorn scaling; the sharper structure induced by the regularizer provides more informative descent directions and leads to faster and more stable convergence.

---

> > ### Comment · Reviewer_cYRs · 2025-11-27
> >
> > I thank the authors for their response. I confirm that I will keep my score.

---

> > > ### Author Response · Authors · 2025-11-27
> > > **Replies to Reviewer cYRs**
> > >
> > > Thank you very much for your time and for the positive evaluation! We are grateful for your careful reading and constructive comments, and we will incorporate your suggestions into the final version of the paper.

---

### Author Response · Authors · 2025-11-29
**Final note to the Area Chair (part 1)**

We would like to sincerely thank you and all four reviewers for the careful reading of our submission and for the many constructive comments. We have prepared a revised version of the paper, with all changes marked in blue in both the main text and the supplementary material. Below we briefly summarize how we addressed the main points raised in the reviews.

1. **Clarifying the model and its motivation.**
   Several reviewers asked for a clearer explanation of the practical scenarios where attributed Gaussian Wigner graphs are relevant, and how our model differs from existing alignment models. In Section 1, we now:

   - Emphasize applications where *both* continuous edge weights and node features naturally arise;

   - Clarify the definition of the featured correlated Gaussian Wigner model and the role of the positive correlations $\rho,r>0$;


2. **Relation to prior work.**
   We expanded the discussion of related work on attributed graph alignment and community recovery, including ER/SBM-based attributed models, and made the comparison with Wigner-based alignment results more explicit. The revision now makes it clearer which aspects of the problem are new (weighted edges + features, sharp partial/exact recovery thresholds, and a fast algorithm with guarantees) and how our results connect to existing thresholds in the unfeatured Wigner case.

3. **Assumptions and information-theoretic thresholds.**
   Reviewers asked for more intuition behind assumptions such as $d=\omega(\log n)$ and $r^2 \ge 40/d$, and for a clearer discussion of the gap between our achievability and converse results. We now:

   - Explain that these technical conditions are in line with prior work on feature-augmented alignment, and are used to control concentration and large deviations in our proofs;

   - Discuss the remaining gap between achievability and converse bounds and explicitly highlight it as an interesting direction for future work;

   - Clarify the correspondence between our thresholds and those in prior Wigner/ER-based results (e.g., the transition around $n\rho^2 \approx 4\log n$ or $n\log(1/(1-\rho^2)) \approx 4\log n$), showing that there is no contradiction but only a difference in parametrization.

4. **Probabilistic formulation and the role of the permutation prior.**
   One reviewer raised concerns about whether the ground-truth permutation is fixed or random, and how this affects the lower bound. We now:

   - State more explicitly that in our model the permutation is a fixed but unknown parameter, and that the alignment goal is to recover this parameter;

   - Explain that for the converse (impossibility) bound we adopt the standard Bayes argument with a uniform prior over permutations, and that by symmetry this Bayes risk coincides with the minimax risk, so there is no inconsistency between the model and the analysis.

---

### Author Response · Authors · 2025-11-29
**Final note to the Area Chair (part 2)**

5. **Algorithmic analysis, regularization, and Sinkhorn vs. Dykstra.**
   Several comments concerned the algorithmic side, in particular the role of the regularization term and the choice of Sinkhorn. In the algorithm section and the supplement, we now:

   - Clarify that the regularizer $-\mu\sum_{i,j}\Pi_{ij}(1-\Pi_{ij})$ is introduced to push the relaxed assignment toward $\{0,1\}$ and away from ambiguous intermediate values; empirically, removing it leads to noticeably softer solutions and lower overlap in medium signal regimes, while the effect is small in very strong-signal regimes;

   - Add a discussion on why we use Sinkhorn rather than a Dykstra-based Euclidean projection: our goal is to remain in the Birkhoff polytope, not to compute exact Euclidean projections; Sinkhorn is widely used in graph matching, is substantially faster in large-scale problems, and in practice works well for our setting;

   - Clarify how we handle negativity in practice (replacing negative entries by zero before Sinkhorn scaling), and adjust the wording to avoid suggesting that Sinkhorn provides an exact Euclidean projection.

6. **Theoretical guarantee for QPAlign.**
   To further connect the algorithm to the theory, we added a new Theorem 3 providing a convergence/partial recovery guarantee for the gradient-descent-based QPAlign procedure in an appropriate signal regime. This theorem highlights how the structural and feature information are jointly exploited in a single quadratic assignment objective and gives a more principled explanation of the algorithm’s performance.

7. **Empirical illustration of the statistical–computational tradeoff.**
   In response to the request for a clearer connection between information-theoretic thresholds and empirical performance, we added a new Figure 5 and the accompanying discussion. This figure compares the information-theoretic phase transition predicted by Theorems 1 and 2 with the empirical success boundary of QPAlign. The two largely coincide in medium/high-signal regimes, while a visible gap remains in low-signal regimes, illustrating a potential statistical–computational tradeoff for this problem.

8. **Response to concerns about clarity raised by one reviewer.**
   One reviewer expressed strong concerns about the clarity of the model definition. We substantially reorganized and streamlined the presentation in the introduction and Section 2 to avoid misunderstandings:

   - We emphasize that $(G_1,G_2,X,Y)$ are generated jointly from a featured correlated Gaussian Wigner model, with the same latent permutation coupling both edges and features;

   - We clarify that we do not assume the observed graphs are exactly isomorphic, nor do we freely tune an “edge density” parameter; instead, the correlation structure determines the effective signal-to-noise ratio;

   - We explicitly state that we focus on positive correlations $\rho,r>0$ and briefly mention negative correlations as an interesting but technically more delicate direction for future work.

Overall, we believe that these clarifications and additions have significantly improved the readability and completeness of the paper. We are very grateful to you and the reviewers for the time and effort invested in evaluating our work, and we hope that the revised version addresses the concerns raised in the reviews.

---

### Meta-Review · Area_Chair_FAFE · 2026-01-02

**Summary:**

The authors provided a thoughtful and constructive rebuttal, significantly improving the clarity of the model, its motivation, and the presentation of the theoretical and algorithmic components. These revisions help address several concerns regarding exposition and positioning relative to prior work.
Nevertheless, key issues raised by multiple reviewers remain only partially resolved, for example, questions regarding the theoretical novelty of the proposed model. Given the substantial divergence in reviewer scores and the lack of a clear consensus, I believe the paper is not yet ready for acceptance at ICLR. I therefore recommend rejection, while encouraging further refinement of the theoretical contributions and their motivation.

**Reviewer Concerns:**

The rebuttal effectively addressed several concerns related to clarity and presentation. In particular, the authors clarified the generative model and problem formulation, strengthened the discussion of related work, and explained the motivation behind key assumptions. The theoretical guarantee were also better justified, and additional empirical results helped contextualize the algorithm’s behavior. However, important concerns remain outstanding. These include questions regarding the fundamental novelty of the model relative to prior attributed graph alignment frameworks, and the extent to which the theoretical analysis yields genuinely new insights beyond existing techniques.

**Reviewer Scores:**

**Reviewer QxbF (original score 0 → 2)**

The reviewer’s main concerns centered on the ambiguity of the problem formulation and the generative model definition. The rebuttal substantially clarified that the graphs are jointly generated from a featured correlated Gaussian Wigner model and are not assumed to be isomorphic, addressing the reviewer’s most critical source of confusion. The improved clarity and explicit statements in the revision would likely shift the reviewer’s assessment from a strong rejection to a more neutral, borderline position.

**Reviewer FeZq (original score 2 → 4)**

This reviewer raised concerns about the motivation, novelty, and the strength of the theoretical contribution relative to prior work. The rebuttal expanded the discussion of related models, clarified the role of positive correlations, and acknowledged the remaining gap between achievability and converse bounds. While the rebuttal improved clarity and transparency, it did not fundamentally change the reviewer’s assessment of novelty and insight, making a shift to a borderline score plausible but insufficient to support acceptance.

**Reviewer 2pFs (original score 4 → 6)**

The reviewer initially raised concerns about the gap between achievability and converse results, the limited theoretical novelty, and the lack of algorithmic guarantees. The rebuttal and revision directly engaged with these points by adding further discussion of the information-theoretic gap, introducing a new theoretical result linking the algorithm to partial recovery and so on. While some gaps and limitations remain, these additions substantially strengthen the paper and clarify its contributions. As a result, it is plausible that the reviewer would update their assessment to a moderately positive score, reflecting improved confidence while still recognizing remaining weaknesses.

**Reviewer cYRs (original score 8 → 8)**

The reviewer was already highly positive about the paper’s theoretical framework and contributions. The rebuttal addressed their suggestions by adding further discussion of extensions, optimization aspects, and algorithmic guarantees, reinforcing rather than altering their original assessment. As the rebuttal aligns well with the reviewer’s expectations, their score would likely remain unchanged at 8.

---

### Decision · Program_Chairs · 2026-01-26

Reject